# Manifold Constraint Reduces Exposure Bias in Accelerated Diffusion Sampling

**Yuzhe Yao**[1][*] **Jun Chen**[2,5], **Zeyi Huang**[3], **Haonan Lin**[1], **Mengmeng Wang**[4,5][†]
**Guang Dai**[5], **Jingdong Wang**[6]
[1]Xi'an Jiaotong University, [2]Zhejiang Normal University, [3]Huawei,
[4]Zhejiang University of Technology, [5] SGIT AI Lab, State Grid Corporration of China, [6] Baidu

{yuzheyao.22,junc.change,mengmewang,guang.gdai}@gmail.com
huangzeyi2@huawei.com,linhaonan@stu.xjtu.edu.cn
wangjingdong@outlook.com

## Abstract

Diffusion models have demonstrated significant potential for generating high-quality images, audio, and videos. However, their iterative inference process entails substantial computational costs, limiting practical applications. Recently, researchers have introduced accelerated sampling methods that enable diffusion models to generate samples with far fewer timesteps than those used during training. Nonetheless, as the number of sampling steps decreases, the prediction errors significantly degrade the quality of generated outputs. Additionally, the exposure bias in diffusion models further amplifies these errors. To address these challenges, we leverage a manifold hypothesis to explore the exposure bias problem in depth. Based on this geometric perspective, we propose a manifold constraint that effectively reduces exposure bias during accelerated sampling of diffusion models. Notably, our method involves no additional training and requires only minimal hyperparameter tuning. Extensive experiments demonstrate the effectiveness of our approach, achieving a FID score of 15.60 with 10-step SDXL on MS-COCO, surpassing the baseline by a reduction of 2.57 in FID.

## 1 Introduction

Diffusion models (DMs) Sohl-Dickstein et al. (2015); Ho et al. (2020); Song et al. (2021; 2020); Nichol & Dhariwal (2021) have emerged as powerful generative models for various applications, including image generation (Rombach et al., 2022; Li et al., 2024b), video synthesis (Ho et al., 2022), and audio generation (Kong et al., 2021). DMs are able to generate images of high fidelity and high diversity, when compared to generative adversarial networks (GANs) (Heusel et al., 2017; Goodfellow et al., 2014), variational autoencoders (VAEs) (Kingma & Welling, 2022; Vahdat & Kautz, 2020) and flow models (Lipman et al., 2023; Papamakarios et al., 2017; Kingma et al., 2016). However, DMs necessitate an iterative process to synthesize high-quality results (Ho et al., 2020).

In response to the computational demands of diffusion models, Song et al. (2021) propose an accelerated sampling method. Given a pretrained diffusion model, during generation, instead of sampling all latent variables on the sampling trajectory, Song et al. (2021) sample along a trajectory shorter than that used during training, significantly accelerating the generation. Nevertheless, Kim et al. (2024) find that during the accelerated generation, the error introduced to the sample at every timestep is proportional to the step size. Therefore, reducing the

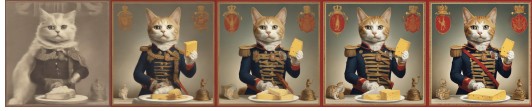

Figure 1: Samples generated with 5, 10, 20, 50, and 1k steps (from left to right). Prompt: *"A propaganda poster depicting a cat dressed as French emperor Napoleon holding a piece of cheese"*.

---

[*]This work was partly completed during the internship at SGIT AI Lab, State Grid Corporation of China.
[†]Corresponding Author.

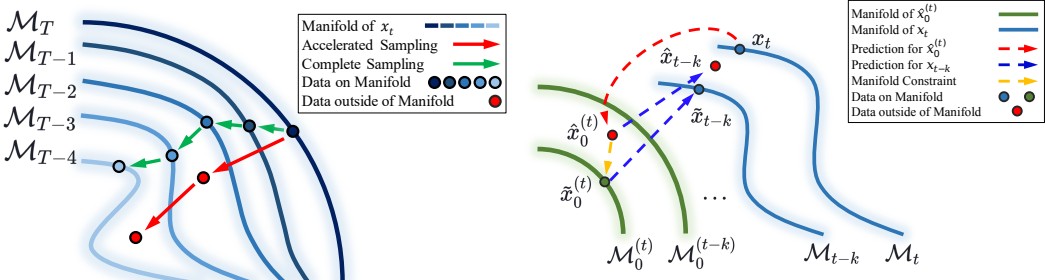

(a) Exposure bias in accelerated sampling      (b) Manifold Constraint on Denoising Observation

Figure 2: In both (a) and (b), the blue manifolds represent noisy data manifolds. In (a), small step sizes allow diffusion models to transport $x_t$ accurately between manifolds. However, with large step intervals (accelerated sampling), the noisy prediction transports data away from the manifolds, reducing accuracy in the next step. In (b), the green manifolds are manifolds of denoising observations (Equation 6). The yellow arrow represents the manifold constraint, which projects $\hat{x}_0^{'(t)}$ toward $\mathcal{M}_0^{(t)}$, allowing error correction in noise estimation and bringing $x_{t-k}$ closer to $\mathcal{M}_{t-k}$.

number of sampling steps, while improving efficiency, can introduce substantial noise and degrade the sample quality (Rombach et al., 2022). To illustrate how the prediction errors affect sample quality, we utilized Stable Diffusion XL (Podell et al., 2024) to generate images with varying sampling steps. The results, shown in Figure 1, demonstrate that as the number of sampling steps decreases, the generated images exhibit a noticeable loss of detail and increased distortion of objects.

We propose that another issue contributes to the degradation of sample quality during accelerated generation: exposure bias (Schmidt, 2019). This training-inference discrepancy, commonly observed in auto-regressive models, also affects diffusion models (Ning et al., 2023; Li & van der Schaar, 2024; Li et al., 2024a; Ning et al., 2024; Ren et al., 2024). During training, diffusion models are only exposed to ground truth inputs; however, when the input is significantly corrupted (e.g., during accelerated sampling), the models may generate increasingly inaccurate predictions. This cumulative effect is illustrated in Figure 2a.

Li et al. (2024a) evaluate exposure bias using the concept of *pixel variance* within a sample and propose a simple yet effective method based on it. Since pixel variance differs significantly from the *sample variance* used in prior works (Ho et al., 2020; Song et al., 2021; Nichol & Dhariwal, 2021; Dhariwal & Nichol, 2021), we are particularly interested in its implications, especially in accelerated sampling.

To explore this, we collected the pixel variance for different sampling step settings in the generation of Figure 1 and visualized it in Figure 3. The results show that with larger step sizes, the pixel variance deviates significantly from that observed with smaller step sizes. Inspired by prior works (Song et al., 2021; Chung et al.,

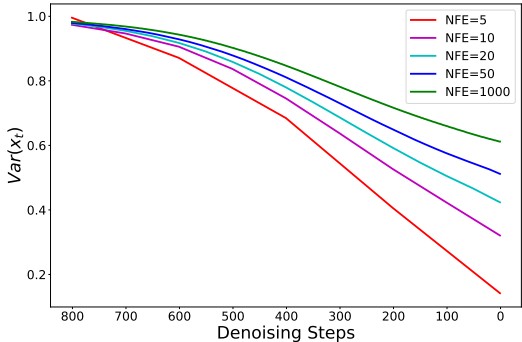

Figure 3: Variation in pixel variance for a single sample across sampling settings: 5, 10, 20, 50, and 1000 steps.

2022; Chen et al., 2023; Yu et al., 2023; Humayun et al., 2024; Su et al., 2024), we aim to explore the relation between exposure bias and pixel variance of a sample under the manifold hypothesis.

In this work, we focus on addressing exposure bias in the accelerated sampling process of diffusion models. Our contributions are summarized as follows:

- We identify exposure bias as a key challenge in accelerated sampling for diffusion models and introduce the manifold hypothesis, offering a geometric perspective on this issue.

- Based on the manifold hypothesis, we propose a manifold constraint on the denoising observation in diffusion models. This method is training-free, nearly hyperparameter-free, and effectively reduces exposure bias by correcting prediction errors during accelerated sampling.

- Extensive experiments across various image generation tasks and base models demonstrate consistent and substantial improvements with our approach.

## 2 RELATED WORK

**Diffusion Models** represent a significant advancement in generative modeling. Initially introduced by Sohl-Dickstein et al. (2015), these models iteratively refine data through denoising. Key enhancements were made by Ho et al. (2020), who incorporated a weighted variational bound in training. Song et al. (2021) introduced a more efficient class of iterative implicit probabilistic models with the same training objective as DDPMs. Further developments include Nichol & Dhariwal (2021), who optimized model architecture. Additionally, Ho & Salimans (2022) offers a method to jointly train a conditional and an unconditional diffusion model. Additionally, works on accelerating sampling methods (Song et al., 2023; Zhou et al., 2023; Chen et al., 2024; Lu et al., 2022a;b; Zhang & Chen, 2023; Liu et al., 2022) have contributed to their practical use in various domains.

**Exposure Bias** refers to the discrepancy between inputs during training and inference in sequential prediction models (Schmidt, 2019; Wang & Sennrich, 2020; Zhang et al., 2024). This issue also arises in diffusion models, where ground truth inputs are used during training, but previous predictions are fed into the model during inference (Li & van der Schaar, 2024; Ning et al., 2024; 2023; Ren et al., 2024; Yao et al., 2024; Everaert et al., 2024). This discrepancy can lead to cumulative errors in the generation process. Various strategies have been proposed to mitigate exposure bias in diffusion models. Ning et al. (2023) suggest perturbing training data to reflect potential input variations during inference, while Li & van der Schaar (2024) analyze error propagation and recommend using the upper bound of cumulative error as a regularization term during training. Although these approaches enhance robustness, they necessitate model retraining. In contrast, Li et al. (2024a) observe that the pixel variance of noisy samples is not always aligned with the schedule during inference, prompting them to adjust the timestep accordingly. Additionally, Ning et al. (2024) identify noise prediction error as a key source of exposure bias and propose scaling noise estimation to address magnitude discrepancies, although direction errors remain unaddressed.

## 3 BACKGROUND

**Diffusion Models** In the forward diffusion process, Gaussian noise is added to samples from the data distribution $q(x_0)$, over a series of time steps $t \in \{0, 1, \ldots, T\}$, resulting in increasingly noisy versions of the data:

$$q(\boldsymbol{x}_t|\boldsymbol{x}_{t-1}) = \mathcal{N}(\boldsymbol{x}_t; \sqrt{1 - \beta_t}\boldsymbol{x}_{t-1}, \beta_t\mathbf{I}), \tag{1}$$

where $\beta_t$ is a variance schedule that controls the amount of noise added at each step. The overall forward process is described as:

$$q(\boldsymbol{x}_{1:T}|\boldsymbol{x}_0) = \prod_{t=1}^{T} q(\boldsymbol{x}_t|\boldsymbol{x}_{t-1}). \tag{2}$$

In the reverse process, diffusion models denoise $\boldsymbol{x}_T$ back to the original data distribution:

$$p_\theta(\boldsymbol{x}_{t-1}|\boldsymbol{x}_t) = \mathcal{N}(\boldsymbol{x}_{t-1}; \boldsymbol{\mu}_\theta(\boldsymbol{x}_t, t), \boldsymbol{\Sigma}), \tag{3}$$

where $\boldsymbol{\mu}_\theta$ is the mean predicted by a neural network with parameters $\theta$, and $\boldsymbol{\Sigma}$ is the predicted variance. Ho et al. (2020) found that in practice, using a fixed variance schedule $\boldsymbol{\Sigma}_t$ leads to simpler and more stable training. Diffusion models can be trained by minimizing the Kullback-Leibler divergence:

$$\mathcal{L} = \mathbb{E}_{q(\boldsymbol{x}_{0:T})} \left[ \sum_{t=1}^{T} \mathrm{KL}(q(\boldsymbol{x}_{t-1}|\boldsymbol{x}_t, \boldsymbol{x}_0) \| p_\theta(\boldsymbol{x}_{t-1}|\boldsymbol{x}_t)) \right]. \tag{4}$$

An alternative, often simpler, loss function involves predicting the added noise directly:

$$\mathcal{L}_{simple} = \mathbb{E}_{t,\boldsymbol{x}_0,\boldsymbol{\epsilon}_t} \left[ \|\boldsymbol{\epsilon}_t - \boldsymbol{\epsilon}_\theta^{(t)}(\boldsymbol{x}_t)\|^2 \right], \tag{5}$$

where $\boldsymbol{\epsilon}_\theta$ denotes the noise estimation network in diffusion model.

After training, diffusion models can generate data by sampling from the standard normal prior $p(\mathbf{x}_T)$, iteratively refining these samples through the reverse process defined in Equation 3. When employing the DDIM sampler (Song et al., 2021), the model proceeds as follows at each timestep $t$.

First, the model predicts the *denoising observation* using noise estimation:

$$\hat{\boldsymbol{x}}_0^{(t)} = \frac{\boldsymbol{x}_t - \sqrt{1 - \bar{\alpha}_t}\,\boldsymbol{\epsilon}_\theta^{(t)}(\boldsymbol{x}_t)}{\sqrt{\bar{\alpha}_t}}, \tag{6}$$

where $\hat{\boldsymbol{x}}_0^{(t)}$ represents the estimated clean image at timestep $t$, $\boldsymbol{x}_t$ is the noisy image at timestep $t$, and $\boldsymbol{\epsilon}_\theta^{(t)}(\boldsymbol{x}_t)$ is the model's prediction of noise.

Next, the latent variable for timestep $t-1$ is predicted:

$$\hat{\boldsymbol{x}}_{t-1} = \sqrt{\bar{\alpha}_{t-1}}\hat{\boldsymbol{x}}_0^{(t)} + \sqrt{1 - \bar{\alpha}_{t-1} - \sigma_t^2}\,\boldsymbol{\epsilon}_\theta^{(t)}(\mathbf{x}_t) + \sigma_t\boldsymbol{\epsilon}_t', \quad \boldsymbol{\epsilon}_t' \sim \mathcal{N}(\mathbf{0}, \mathbf{I}), \tag{7}$$

where $\alpha_t := 1 - \beta_t$, $\bar{\alpha}_t := \prod_{s=1}^t \alpha_s$, and $\sigma_t$ controls the stochasticity of the generation. The term $\boldsymbol{\epsilon}_t'$ is sampled from a standard normal distribution, introducing stochasticity into the generation process.

**Prediction Error in Accelerated Sampling** Given a diffusion model trained with a forward process of $T$ steps, during generation, instead of sampling all latent variables $\{\boldsymbol{x}_T, \boldsymbol{x}_{T-1}, \ldots, \boldsymbol{x}_1\}$ iteratively, Song et al. (2021) propose to sample along a trajectory of length $T'$: $\{\boldsymbol{x}_{\tau_{T'}}, \boldsymbol{x}_{\tau_{T'-1}}, \ldots, \boldsymbol{x}_{\tau_1}\}$, where $\tau$ is defined as an increasing subsequence of $\{0, 1, \ldots, T\}$. Since $T' < T$, the proposed method significantly accelerates the sampling.

However, the prediction error tends to increase as the sampling interval grows. Kim et al. (2024) derive that the error introduced to the latent variable $\boldsymbol{x}_t$ during the transition from time $t$ to $t - \Delta t$:

$$\boldsymbol{e}_t = \frac{\Delta t}{t}\mathcal{O}(\Delta t), \tag{8}$$

where $\Delta t$ denotes the timestep interval. Equation 8 shows an inherent trade-off between accuracy and efficiency in the sampling process in diffusion models.

**Exposure Bias in Diffusion Models** During training, diffusion models are trained on noisy data sampled from the real data distribution $\boldsymbol{x}_0$:

$$q(\boldsymbol{x}_t|\boldsymbol{x}_0) = \mathcal{N}(\boldsymbol{x}_t; \sqrt{\bar{\alpha}_t}\boldsymbol{x}_0, (1 - \bar{\alpha}_t)\mathbf{I}). \tag{9}$$

On the contrary, models are only exposed to their previously prediction $\hat{\boldsymbol{x}}_t$ during inference. The input discrepancy, $\boldsymbol{x}_t - \hat{\boldsymbol{x}}_t$, disrupts the model's prediction in subsequent timesteps.

The exposure bias is exacerbated in accelerated sampling, where errors at each timestep are significantly amplified, as shown in Equation 8. Therefore, we identify the combined effect of exposure bias and enlarged prediction error as the main challenge in accelerated diffusion model sampling.

## 4 METHOD

Inspired by previous work (Song & Ermon, 2019; Chung et al., 2022; Chen et al., 2024; He et al., 2024), we are interested in understanding the exposure bias problem under the geometric view of diffusion models. In this section, we first provide the geometric view of the exposure bias problem. Then, based on the geometrical analysis, we introduce *Manifold Constraint on Denoising Observation* (MCDO), a training-free and almost hyperparameter tuning-free method that mitigates the exposure bias in the accelerated sampling of diffusion models.

### 4.1 GEOMETRIC VIEW OF EXPOSURE BIAS

To better understand the training-inference discrepancy in diffusion models, we borrow a geometric view of the data manifold.

**Assumption 1** *(Chung et al., 2022) Let $\mathcal{M}_0 \subset \mathbb{R}^n$ be the clean data manifold. The distribution of the ground truth noisy data $p(\boldsymbol{x}_t)$ is concentrated on an $(n-1)$-dimensional manifold $\mathcal{M}_t$.*

Under Assumption 1, for a diffusion model trained with $T$ sampling steps, its ground truth noisy samples obtained with Equation 9 form a series of manifolds $\mathcal{M}_1, \mathcal{M}_2, \ldots, \mathcal{M}_T$. The model is trained to smoothly map the data from $\mathcal{M}_t$ to $\mathcal{M}_{t-1}$ (Chung et al., 2022). During the inference, pure noises from $\mathcal{M}_T$ are gradually mapped towards the target manifold $\mathcal{M}_0$. The perfect generation process can be formulated as follows:

$$\mathcal{M}_T \xrightarrow{\Lambda_T} \cdots \xrightarrow{\Lambda_3} \mathcal{M}_2 \xrightarrow{\Lambda_2} \mathcal{M}_1 \xrightarrow{\Lambda_1} \mathcal{M}_0, \tag{10}$$

where $\Lambda_i$, $i \in \{1, 2, \ldots, T\}$, denotes a series of smooth maps.

However, the perfect mapping is only applicable to data concentrated on the manifold.

**Assumption 2** *(Extended from He et al. (2024)) Define the distance from a data point to a given manifold as: $d(\boldsymbol{x}, \nu, \mathcal{M}) \coloneqq \inf_{\boldsymbol{z} \in \mathcal{M}} \|\boldsymbol{x} - \nu \boldsymbol{z}\|$, for $\nu > 0$. For a given timestep $t$, diffusion models only make accurate noise predictions for data concentrated on the manifold: $\mathbb{X}_t \coloneqq \{\boldsymbol{x}_t \in \mathbb{R}^n \mid d(\boldsymbol{x}_t, 1, \mathcal{M}_t) < r_t\}$, where $r_t > 0$ is the radius.*

While prediction errors are inevitable, it is possible that, for a given timestep $t+1$, data from $\mathcal{M}_{t+1}$ is mapped to points away from $\mathcal{M}_t$ such that $d(\boldsymbol{x}_t, 1, \mathcal{M}_t) > r_t$. With Assumption 2, models' noise estimation for data $\hat{\boldsymbol{x}}_t \notin \mathbb{X}_t$ will degrade. Based on Equation 8, we argue that the large deviation from the manifold is the main reason for exposure bias in the accelerated sampling of diffusion models. We illustrate this problem in Figure 2a.

### 4.2 RELATION BETWEEN DISTANCE TO MANIFOLD AND THE PIXEL VARIANCE OF DATA

Under Assumptions 1 and 2, we relate the pixel variance of a sample $\hat{\boldsymbol{x}}_t$, to its $L^2$ distance to $\mathcal{M}_t$.

For a prediction $\hat{\boldsymbol{x}}_t$, let $\boldsymbol{x}_t^* = \operatorname{argmin}_{\boldsymbol{z}_t \in \mathcal{M}_t} \|\hat{\boldsymbol{x}}_t - \boldsymbol{z}_t\|$. The distance between $\hat{\boldsymbol{x}}_t$ and $\mathcal{M}_t$ satisfies:

$$d(\hat{\boldsymbol{x}}_t, 1, \mathcal{M}_t) = \|\hat{\boldsymbol{x}}_t - \boldsymbol{x}_t^*\| \geq \left| \|\boldsymbol{x}_t^*\| - \|\hat{\boldsymbol{x}}_t\| \right|. \tag{11}$$

Given that $\boldsymbol{x}_T \sim \mathcal{N}(\boldsymbol{0}, \boldsymbol{I})$, all pixels in $\boldsymbol{x}_T$ follows a normal distribution, such that $\mathbb{E}[\frac{1}{n}\Sigma_{i=0}^n \boldsymbol{x}_T] = 0$. According to Equation 4, we have $\mathbb{E}[\frac{1}{n}\Sigma_{i=0}^n \boldsymbol{\epsilon}_t] = 0$, when the diffusion models are well-trained. As a consequence, the expectation of pixel mean of a single noisy sample approaches 0 as well: $\mathbb{E}[\frac{1}{n}\Sigma_{i=0}^n \boldsymbol{x}_t] = 0$, according to Equations 6 and 7 (more details in Appendix A.1). Then the $L^2$ norm of $\hat{\boldsymbol{x}}_t$ and its pixel variance satisfy:

$$\operatorname{Var}(\hat{\boldsymbol{x}}_t) = \frac{1}{n}\Sigma_{i=1}^n (\hat{\boldsymbol{x}}_{t_i} - \frac{1}{n}\Sigma_{j=1}^n \hat{\boldsymbol{x}}_{t_j})^2 \approx \frac{1}{n}\Sigma_{i=1}^n \hat{\boldsymbol{x}}_{t_i}^2 = \frac{1}{n}\|\hat{\boldsymbol{x}}_t\|^2, \tag{12}$$

where $n$ is the data dimension. We show the evolution of $L^2$ norm of a sample as well as its pixel variance in Figure 4. It is shown that the curve of the $L^2$ norm and pixel variance are highly overlapped for most of the time. Therefore, we use Equation 12 to approximate the $L^2$ norm of noisy data with its pixel variance.

Plugging Equations 9 and 12 into Equation 11, we have:

$$d(\hat{\boldsymbol{x}}_t, 1, \mathcal{M}_t) \gtrapprox \left| \|\sqrt{\bar{\alpha}_t}\boldsymbol{x}_0 + \sqrt{1 - \bar{\alpha}_t}\boldsymbol{\epsilon}_t\| - \sqrt{n\operatorname{Var}(\hat{\boldsymbol{x}}_t)} \right|. \tag{13}$$

According to Wegner (2024), for data $\boldsymbol{\epsilon} \sim \mathcal{N}(\boldsymbol{0}, \boldsymbol{I})$, $\boldsymbol{\epsilon} \in \mathbb{R}^n$, one has:

$$\left| \mathbb{E}[\|\boldsymbol{\epsilon}\|] - \sqrt{n} \right| \leq \frac{1}{\sqrt{n}}. \tag{14}$$

Given that the dimension $n$ of the latent space is usually large for diffusion models (e.g., $n = 12288$ for LDM-4 trained on CelebA-HQ), with high probability, $\|\boldsymbol{\epsilon}_t\| \approx \sqrt{n}$.

In addition, since $\epsilon_t$ is a random noise independent of $\boldsymbol{x}_0$, when $n$ is large, $\epsilon_t$ and $\boldsymbol{x}_0$ are almost orthogonal with high probability (Wegner, 2024). Therefore, we have:

$$\mathbb{E}_{\boldsymbol{x}_0 \sim q(\boldsymbol{x}_0), \epsilon_t \sim \mathcal{N}(\boldsymbol{0}, \boldsymbol{I})}[\|\sqrt{\bar{\alpha}_t}\boldsymbol{x}_0 + \sqrt{1 - \bar{\alpha}_t}\epsilon_t\|^2] = \mathbb{E}_{\boldsymbol{x}_0 \sim q(\boldsymbol{x}_0), \epsilon_t \sim \mathcal{N}(\boldsymbol{0}, \boldsymbol{I})}[\|\sqrt{\bar{\alpha}_t}\boldsymbol{x}_0\|^2 + \|\sqrt{1 - \bar{\alpha}_t}\epsilon_t\|^2]. \tag{15}$$

Plugging Equations 12, 14, and 15 into Equation 13, the following inequality holds with high probability:

$$d(\hat{\boldsymbol{x}}_t, 1, \mathcal{M}_t) \gtrapprox \left| \sqrt{\bar{\alpha}_t \|\boldsymbol{x}_0\|^2 + n(1 - \bar{\alpha}_t)} - \sqrt{n\mathrm{Var}(\hat{\boldsymbol{x}}_t)} \right|, \tag{16}$$

where equality holds if and only if $\hat{\boldsymbol{x}}_t$, $\boldsymbol{x}_t^*$ and the origin are collinear.

Equation 16 indicates that the deviation in $\mathrm{Var}(\hat{\boldsymbol{x}}_t)$ could potentially make $d(\hat{\boldsymbol{x}}_t, 1, \mathcal{M}_t) > r_t$, resulting in exposure bias problem under Assumption 2. This is consistent with Figures 1 and 3.

From this perspective, the timestep-shifting strategy proposed by Li et al. (2024a), which determines timestep $t_s$ within range $w$: $t_s = \mathrm{argmin}_{t'} \left( \sqrt{1 - \bar{\alpha}_{t'}} - \sqrt{\mathrm{Var}(\hat{\boldsymbol{x}}_t)} \right), t' \in \{t - w/2, \ldots, t + w/2\}$, can be interpreted as locating the manifold $\mathcal{M}_i, i \in \{t - w/2, \ldots, t + w/2\}$, where the lower bound of samples-to-manifold distance is potentially reduced according to Equation 16.

As this strategy provides valuable insights, the inaccessibility of the $L^2$ norm of the clean data $\boldsymbol{x}_0$ in Equation 16 presents an opportunity for further refinement of the approach.

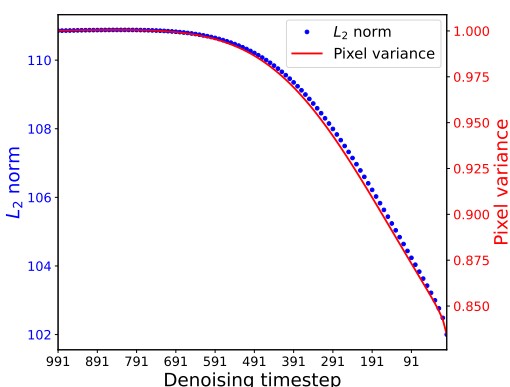

Figure 4: Evolution of $L^2$ norm and pixel variance of the noisy data. We plot the average statistics of 6,400 samples in 100-steps generation on CelebA-HQ.

Inspired by prior studies (Bao et al., 2022; Chen et al., 2023; 2024), we delve into the model's predictions for noisy data and the corresponding denoising observations. Specifically, when the model is given the ground truth input at $t + 1$, the analytical form of $\hat{\boldsymbol{x}}_0^{(t)}$ and $\hat{\boldsymbol{x}}_t$ can be expressed as (see Appendix A.2):

$$\hat{\boldsymbol{x}}_0^{(t)} = \boldsymbol{x}_0 + d_t \boldsymbol{e}_\theta^{(t)}, \hat{\boldsymbol{x}}_t = \boldsymbol{x}_t + n_t \boldsymbol{e}_\theta^{(t)}, \tag{17}$$

Here, $\boldsymbol{x}_0$ denotes the clean data, $\boldsymbol{x}_t$ the ground truth noisy data, $\boldsymbol{e}_\theta^{(t)}$ the noise estimation error. The scaling factors $d_t$ and $n_t$ vary with $t$. We find that while $n_t$ remains within a small range near 0, $d_t$ is hundreds or even thousands of times larger than $n_t$ when $t$ is large (see Figure 5).

This substantial magnitude of $d_t$ suggests that $\hat{\boldsymbol{x}}_0^{(t)}$ is predominantly influenced by the scaled noise prediction error term $d_t \boldsymbol{e}_\theta^{(t)}$ at most timesteps. As a consequence, we propose to rectify the noise estimation error $\boldsymbol{e}_\theta^{(t)}$ by incorporating information from the denoising observation $\hat{\boldsymbol{x}}_0^{(t)}$.

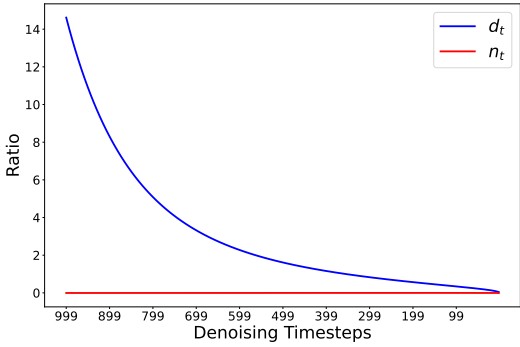

Figure 5: Comparison between error scale on denoising observation ($d_t$) and on noisy data ($n_t$).

### 4.3 MANIFOLD
### CONSTRAINT ON DENOISING OBSERVATION

Chung et al. (2022) view the diffusion process as an interpolation between $\mathcal{M}_0$ and $\mathcal{M}_T$. Inspired by this, we substitute $\hat{\epsilon}_\theta^{(t)}(\boldsymbol{x})$ in Equation 7 with Equation 6:

$$(\frac{\sqrt{1-\bar{\alpha}_{t-1}-\sigma_t^2}\sqrt{\bar{\alpha}_t}}{\sqrt{1-\bar{\alpha}_t}} - \sqrt{\bar{\alpha}_{t-1}})\hat{\boldsymbol{x}}_0^{(t)} = \frac{\sqrt{1-\bar{\alpha}_{t-1}-\sigma_t^2}}{\sqrt{1-\bar{\alpha}_t}}\boldsymbol{x}_t - \hat{\boldsymbol{x}}_{t-1} + \sigma_t\epsilon_t', \epsilon_t' \sim \mathcal{N}\left(\boldsymbol{0}, \boldsymbol{I}\right). \quad (18)$$

Equation 18 suggests that $\hat{\boldsymbol{x}}_0^{(t)}$ can be interpreted as interpolating between two noisy data. Based on this, we assume that the denoising observation $\hat{\boldsymbol{x}}_0^{(t)}$ also concentrates on a series of manifolds.

**Assumption 3** *(Extended from Chung et al. (2022)) Denote $\mathbb{X}_0^{(t)}$ as the set of denoising observations of $\boldsymbol{x}_t$, $\boldsymbol{x}_t \in \mathbb{X}_t$. There exists a series of manifolds: $\mathcal{M}_0^{(T)}, \mathcal{M}_0^{(T-1)}, \ldots, \mathcal{M}_0^{(1)}, \mathcal{M}_0^{(0)}$, where $\mathbb{X}_0^{(t)}$, for $t \in \{T, T-1, \ldots, 0\}$, concentrates on: $d(\hat{\boldsymbol{x}}_0^{(t)}, 1, \mathcal{M}_0^{(t)}) < r_0^{(t)}, \forall \hat{\boldsymbol{x}}_0^{(t)} \in \mathbb{X}_0^{(t)}$. Where $r_0^{(t)} > 0$ is the radius.*

Under Assumption 3, when a noisy data $\hat{\boldsymbol{x}}_t$ is distant from $\mathcal{M}_t$, its denoising observation, calculated using Equation 6, could potentially be mapped away from $\mathcal{M}_0^{(t)}$. For clarity, we denote denoising observations close to $\mathcal{M}_0^{(t)}$ as $\hat{\boldsymbol{x}}_0^{(t)}$, and those distant from $\mathcal{M}_0^{(t)}$ as $\hat{\boldsymbol{x}}_0^{'(t)}$.

For a denoising observation $\hat{\boldsymbol{x}}_0^{'(t)} \notin \mathbb{X}_0^{(t)}$. Let $\hat{\boldsymbol{x}}_0^{*(t)} := argmin_{\boldsymbol{z}_0^{(t)} \in \mathcal{M}_0^{(t)}} \|\hat{\boldsymbol{x}}_0^{'(t)} - \boldsymbol{z}_0^{(t)}\|$.

As $\hat{\boldsymbol{x}}_0^{'(t)}$ is a linear combination of $\boldsymbol{x}_t$ and $\boldsymbol{\epsilon}_\theta^{(t)}$ according to Equation 6, the expectation of pixel mean for both $\hat{\boldsymbol{x}}_0^{'(t)}$ and $\boldsymbol{x}_0^{*(t)}$ is 0. We substitute $\hat{x}_t$ and $x_t^*$ in Equation 11 with $\hat{\boldsymbol{x}}_0^{'(t)}$ and $\boldsymbol{x}_0^{*(t)}$:

$$d(\hat{\boldsymbol{x}}_0^{'(t)}, 1, \mathcal{M}_0^{(t)}) \geq |\|\hat{\boldsymbol{x}}_0^{'(t)}\| - \|\hat{\boldsymbol{x}}_0^{*(t)}\|| \approx \sqrt{n}\left|\sqrt{\mathrm{Var}(\hat{\boldsymbol{x}}_0^{'(t)})} - \sqrt{\mathrm{Var}(\hat{\boldsymbol{x}}_0^{*(t)})}\right|. \quad (19)$$

Equation 19 indicates that the deviated $\hat{\boldsymbol{x}}_0^{'(t)}$ can be constrained to approach $\mathcal{M}_0^{(t)}$, when its pixel variance is scaled to $\mathrm{Var}(\hat{\boldsymbol{x}}_0^{*(t)})$.

Given that $\hat{\boldsymbol{x}}_0^{*(t)}$ is inaccessible in generation process, we introduce the *reference pixel variance*: $v_t = \mathbb{E}_{\boldsymbol{x}_T \sim \mathcal{N}(\boldsymbol{0}, \boldsymbol{I}), y \sim \mathbb{Y}} \mathrm{Var}(\hat{\boldsymbol{x}}_0^{(t)}|\boldsymbol{x}_T, y)$, where $\mathbb{Y}$ denotes the set of conditions. We approximate: $\mathrm{Var}(\hat{\boldsymbol{x}}_0^{*(t)}) \approx v_t, \forall \hat{\boldsymbol{x}}_0^{*(t)} \in \mathcal{M}_0^{(t)}$. In practice, the pixel variance used for estimating $v_t$ is collected during generation with large sampling steps (e.g., $T = 1000$), such that the denoising observations are close to their corresponding manifolds. The pre-computation process is detailed in Algorithm 1.

During accelerated sampling, we first align the pixel variance of $\hat{\boldsymbol{x}}_0^{'(t)}$ to approach $v_t$:

$$c_t = \mathrm{Var}(\hat{\boldsymbol{x}}_0^{'(t)}), \quad \tilde{\boldsymbol{x}}_0^{(t)} = \sqrt{\frac{v_t}{c_t}}\left[\hat{\boldsymbol{x}}_0^{'(t)} - \mathrm{Mean}\left(\hat{\boldsymbol{x}}_0^{'(t)}\right)\right] + \mathrm{Mean}\left(\hat{\boldsymbol{x}}_0^{'(t)}\right). \quad (20)$$

Then, we correct the noise estimation, by inverting Equation 6:

$$\tilde{\boldsymbol{\epsilon}}_\theta^{(t)}\left(\boldsymbol{x}_t\right) = \frac{\boldsymbol{x}_t - \sqrt{\bar{\alpha}_t}\tilde{\boldsymbol{x}}_0^{(t)}}{\sqrt{1-\bar{\alpha}_t}}. \quad (21)$$

By substituting Equations 20 and 21 into Equation 7, we obtain a refined prediction for $\boldsymbol{x}_{t-1}$:

$$\tilde{\boldsymbol{x}}_{t-1} = \sqrt{\bar{\alpha}_{t-1}}\tilde{\boldsymbol{x}}_0^{(t)} + \sqrt{1-\bar{\alpha}_{t-1}-\sigma_t^2}\tilde{\boldsymbol{\epsilon}}_\theta^{(t)}\left(\boldsymbol{x}_t\right) + \sigma_t\epsilon_t', \epsilon_t' \sim \mathcal{N}\left(\boldsymbol{0}, \boldsymbol{I}\right). \quad (22)$$

The correction is applied for $t \in \{T, \ldots, t_{thre}\}$, where $t_{thre}$ is a timestep threshold. The proposed method is illustrated in Figure 2b and detailed in Algorithm 2.

We refer to the proposed method as *Manifold Constraint on Denoising Observation* (MCDO), as its key idea is to constrain the denoising observation to approach its corresponding manifold.

A distinction between reducing the lower bound of $d(\hat{x}_t, 1, \mathcal{M}_t)$ by scaling $\text{Var}(\hat{x}_t)$ and that of $d(\hat{x}_0^{'(t)}, 1, \mathcal{M}_0^{(t)})$ by correcting $\text{Var}(x_0^{'(t)})$ is that the former modifies $x_t$, while the latter refines the noise estimation when $x_t$ is preserved (see Appendix A.3).

Estimating $v_t$ requires few samples (e.g., $N = 20$ for SDXL on MS-COCO). Once computed, $v_t$ can be directly used to correct $\text{Var}(\hat{x}_0^{'(t)})$ during accelerated sampling *without additional computational overhead*. Since $v_t$ depends only on the pre-trained model, when accurately estimated, the only tunable hyperparameter is $t_{thre}$, which often works well at $t_{thre} = 0$. Thus, DDIM-MCDO requires minimal hyperparameter tuning.

---

**Algorithm 1** Pre-Computation for $v_t$

1: Initialize $n_t = \text{list}()$
2: **for** $i = 1, 2, \ldots, N$ **do**
3: $\quad x_T \sim \mathcal{N}(\mathbf{0}, \mathbf{I}), y \sim \mathbb{Y}$
4: $\quad$ **for** $t = T, \ldots, 2, 1$ **do**
5: $\qquad \hat{x}_0^{(t)} = \frac{x_t - \sqrt{1-\bar{\alpha}_t}\epsilon_\theta^{(t)}(x_t, y)}{\sqrt{\bar{\alpha}_t}}$
6: $\qquad n_t.append\left(\text{Var}\left(\hat{x}_0^{(t)}\right)\right)$
7: $\qquad \overrightarrow{x_t} = \sqrt{1 - \bar{\alpha}_{t-1} - \sigma_t^2}\epsilon_\theta^{(t)}(x_t, y)$
8: $\qquad \epsilon_t' \sim \mathcal{N}(\mathbf{0}, \mathbf{I})$
9: $\qquad x_{t-1} = \sqrt{\bar{\alpha}_{t-1}}\hat{x}_0^{(t)} + \overrightarrow{x_t} + \sigma_t\epsilon_t'$
10: $\quad$ **end for**
11: **end for**
12: **for** $t = T, \ldots, 2, 1$ **do**
13: $\quad v_t = \text{Mean}(n_t)$
14: **end for**

---

## 5 EXPERIMENTS

### 5.1 EXPERIMENTS SETUP

We conduct experiments on high-resolution datasets across various tasks. Specifically, we evaluate MS-COCO (Lin et al., 2014) with SDXL (Podell et al., 2024), and CelebA-HQ (Karras et al., 2018), ImageNet-256 (Russakovsky et al., 2015), and LSUN-Bedroom ($256 \times 256$) with LDM-4 (Rombach et al., 2022).

For text-to-image (T2I) generation, we generate 30k samples using MS-COCO prompts to compute FID (Heusel et al., 2017) and CLIP scores (Radford et al., 2021). For other tasks, 50k images are generated to evaluate FID, with sFID and IS reported for ImageNet. We compare our method with state-of-the-art training-free approaches (Ning et al., 2024; Li et al., 2024a), following Li et al. (2024a) to implement TS-DDIM and conducting a hyperparameter search based on their range and qualitative results. For Ning et al. (2024), a coarse-to-fine search is used for the uniform schedule $\lambda$. We visualize exposure bias reduction using Ning

---

**Algorithm 2** MCDO Sampling

1: $x_T \sim \mathcal{N}(\mathbf{0}, \mathbf{I}), y \sim \mathbb{Y}$
2: **for** $i = T', T' - 1, \ldots, 1$ **do**
3: $\quad \hat{x}_0^{'(\tau_i)} = \frac{x_{\tau_i} - \sqrt{1-\bar{\alpha}_t}\epsilon_\theta^{(\tau_i)}(x_{\tau_i}, y)}{\sqrt{\bar{\alpha}_{\tau_i}}}$
4: $\quad$ **if** $\tau_i \geq t_{thre}$ **then**
5: $\qquad \mu_{\tau_i} = \text{Mean}\left(\hat{x}_0^{'(\tau_i)}\right)$
6: $\qquad c_{\tau_i} = \text{Var}(\hat{x}_0^{'(\tau_i)})$
7: $\qquad \tilde{x}_0^{(\tau_i)} = \sqrt{\frac{v_{\tau_i}}{c_{\tau_i}}}\left(\hat{x}_0^{'(\tau_i)} - \mu_{\tau_i}\right) + \mu_{\tau_i}$
8: $\qquad \tilde{\epsilon}_\theta^{(\tau_i)} = \frac{x_{\tau_i} - \sqrt{\bar{\alpha}_{\tau_i}}\tilde{x}_0^{(\tau_i)}}{\sqrt{1-\bar{\alpha}_{\tau_i}}}$
9: $\quad$ **else**
10: $\qquad \tilde{\epsilon}_\theta^{(\tau_i)} = \hat{\epsilon}_\theta^{(\tau_i)}(x_{\tau_i}, y), \tilde{x}_0^{(\tau_i)} = \hat{x}_0^{(\tau_i)}$
11: $\quad$ **end if**
12: $\quad \overrightarrow{x_{\tau_i}} = \sqrt{1 - \bar{\alpha}_{\tau_{i-1}} - \sigma_\tau^2}\tilde{\epsilon}_\theta^{(\tau_i)}$
13: $\quad \epsilon_\tau' \sim \mathcal{N}(\mathbf{0}, \mathbf{I})$
14: $\quad \tilde{x}_{\tau_{i-1}} = \sqrt{\bar{\alpha}_{\tau_{i-1}}}\tilde{x}_0^{(\tau_i)} + \overrightarrow{x_{\tau_i}} + \sigma_\tau\epsilon_\tau'$
15: **end for**

---

et al. (2024)'s metric (see Appendix A.4). Results with DPM-Solver++ (Lu et al., 2022b) and Stable Diffusion 3 (Esser et al., 2024) are in Appendices A.12 and A.13.

### 5.2 MAIN RESULTS ON TEXT-TO-IMAGE GENERATION

Results of T2I generation (Table 1, Figure 6) show that with few sampling steps, image details and object shapes degrade significantly, suggesting deviation from the data manifold. The proposed manifold constraint improves image quality, indicating more accurate inference.

To implement MCDO, we use $N = 20, T = 1000$ for pre-computation. Samples are conditioned on different prompts from MS-COCO. While a larger $N$ could improve $v_t$ estimation, we keep it fixed for simplicity. For TS-DDIM (Li et al., 2024a) we use $t_c = 300, w = 4$ for cutoff timestep and window size. A uniform epsilon scale: $\lambda = 1.008$ is applied for DDIM-ES (Ning et al., 2024).

Table 1: Results on Text-to-Image generation ($1024 \times 1024$) on MS-COCO with SDXL.

| Model | Method | Steps | FID↓ | CLIP Score↑ | $t_{thre}$ |
|---|---|---|---|---|---|
| | DDIM | 10 | 18.17 | 31.58 | – |
| SDXL | DDIM-ES Ning et al. (2024) | 10 | 18.91 | 31.59 | – |
| | TS-DDIM Li et al. (2024a) | 10 | 23.82 | 31.31 | – |
| | DDIM-MCDO | 10 | **15.60** | **31.75** | 0 |

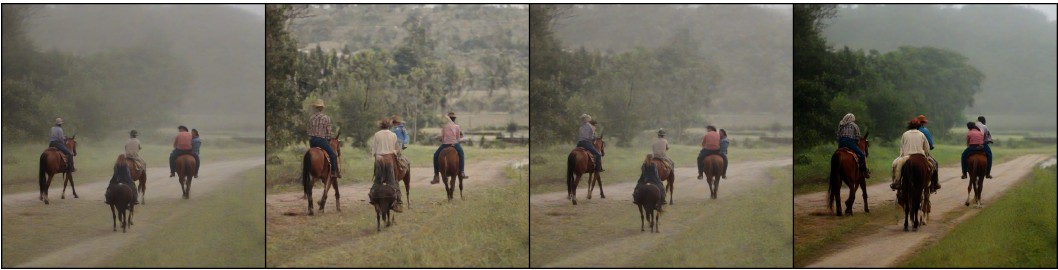

(a) *"Three people on horse back at a rural road intersection."*

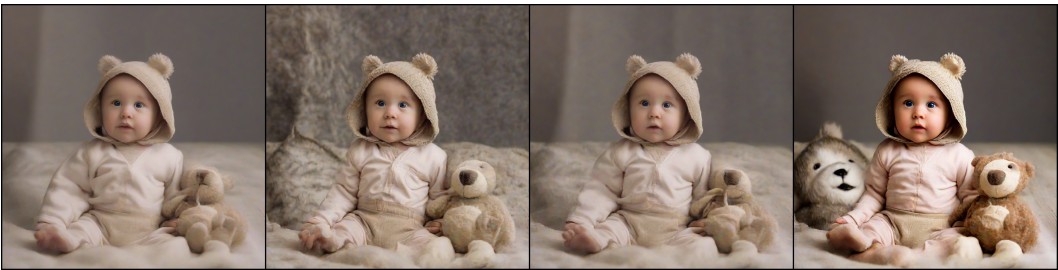

(b) *"A baby next to a stuffed bear of some sort."*

Figure 6: Images generated using 10 steps SDXL with DDIM sampler. From left to right: SDXL, TS-DDIM (Li et al., 2024a), LDM-ES (Ning et al., 2024), and DDIM-MCDO (ours).

## 5.3 RESULTS ON UNCONDITIONAL GENERATION

Considering that both LDM-ES (Ning et al., 2024) and DDIM-MCDO aim to reduce noise estimation error, we investigate their orthogonality. We conduct experiments combining DDIM-MCDO with LDM-ES using the same hyperparameters, denoted as DDIM-MCDO†.[1]

We compare DDIM-MCDO and DDIM-MCDO†with state-of-the-art methods (Ning et al., 2024; Li et al., 2024a) on the CelebA-HQ dataset (Karras et al., 2018) using LDM-4. Results under the recommended setting ($\eta = 0.0$) (Rombach et al., 2022) are shown in Table 2, with $\lambda$ representing the *epsilon scaling factor* from Ning et al. (2024). For TS-DDIM (Li et al., 2024a), we set $t_c = 100$, $w = 60$ for 20 and 10 steps sampling, and $t_c = 200$, $w = 30$ for 5 steps. Qualitative results are in Appendix A.6.

Table 2: Results on CelebA-HQ $256 \times 256$ with DDIM Sampler. $\eta = 0.0$. Bold indicates the best results, and underlined indicates the second-best.

| Method | $T'$ | FID↓ | $\lambda$ |
|---|---|---|---|
| DDIM | 20 | 10.59 | – |
| TS-DDIM (Li et al., 2024a) | 20 | **8.29** | – |
| LDM-ES (Ning et al., 2024) | 20 | 18.39 | 1.001 |
| LDM-ES (Ning et al., 2024) | 20 | 11.73 | 0.999 |
| DDIM-MCDO | 20 | 9.49 | – |
| DDIM-MCDO† | 20 | 10.59 | 1.001 |
| DDIM-MCDO† | 20 | 10.53 | 0.999 |
| DDIM | 10 | 21.08 | – |
| TS-DDIM (Li et al., 2024a) | 10 | 15.71 | – |
| LDM-ES (Ning et al., 2024) | 10 | 18.56 | 1.002 |
| DDIM-MCDO | 10 | 15.63 | – |
| DDIM-MCDO† | 10 | **13.36** | 1.002 |
| DDIM | 5 | 54.99 | – |
| TS-DDIM (Li et al., 2024a) | 5 | 50.69 | – |
| LDM-ES (Ning et al., 2024) | 5 | 49.40 | 1.005 |
| DDIM-MCDO | 5 | 30.76 | – |
| DDIM-MCDO† | 5 | **27.39** | 1.005 |

DDIM-MCDO significantly enhances performance across all three accelerated generation settings, attributed to its correction of both magnitude

---

[1] †DDIM-MCDO combined with LDM-ES using the same hyperparameters.

and direction errors in noise estimation. It is worth noting that as $T'$ decreases, the DDIM-MCDO yields more improvement, compared to other methods. Notably, DDIM-MCDO†outperforms either component method alone, both exceeding the baseline. Additionally, at $T' = 20$, LDM-ES (Ning et al., 2024) shows sensitivity to hyperparameters (e.g., a large FID difference when $\lambda$ varies from 0.999 to 1.001), but performance stabilizes when combined with DDIM-MCDO.

To maintain consistency with (Ning et al., 2024), we report results under the DDPM setting ($\eta = 1.0$) in Table 8 (see Appendix A.8). Additionally, we provide quantitative results for LSUN-Bedroom $256 \times 256$ (Yu et al., 2015) in Table 3, with corresponding qualitative results available in Appendix A.9.

For implementing DDIM-MCDO, variance statistics from 64 samples are collected during 500 steps of DDIM sampling. In all experiments on CelebA-HQ and LSUN-Bedroom, we set the threshold timestep $t_{thre}$ to 0. Given that DDIM-MCDO consistently improves performance across all experiments, we consider it a hyperparameter tuning-free.

Table 3: Results on LSUN-Bedroom $256 \times 256$, with LDM-4 and DDIM sampler, $\eta = 0.0$, $t_{thre} = 0$.

| Method | $T'$ | FID↓ |
|---|---|---|
| DDIM | 20 | 4.26 |
| DDIM-MCDO | 20 | **4.20** |
| DDIM | 10 | 9.33 |
| DDIM-MCDO | 10 | **6.25** |
| DDIM | 6 | 25.94 |
| DDIM-MCDO | 6 | **12.34** |
| DDIM | 5 | 44.91 |
| DDIM-MCDO | 5 | **19.31** |
| DDIM | 4 | 78.70 |
| DDIM-MCDO | 4 | **33.79** |

### 5.4 RESULTS ON CLASS-CONDITIONAL GENERATION

We conduct class-conditional image generation on ImageNet-256 (Russakovsky et al., 2015). To show the hyperparameter tuning-free feature of DDIM-MCDO, we employ the same hyperparameters in Section 5.3: $N = 64$, $t_{thre} = 0$. The pixel variance of 64 samples from different classes (1 sample per class) is collected during a 500-step DDIM sampling. Quantitative and qualitative results are shown in Table 4 and Appendix A.10, respectively.

Table 4: Results on ImageNet-256, with LDM-4 and DDIM sampler. $\eta = 0.0$, guidance scale $s = 3.0$, $t_{thre} = 0$.

| Method | $T'$ | FID↓ | sFID↓ | IS↑ |
|---|---|---|---|---|
| DDIM | 20 | **11.01** | 7.69 | 69.44 |
| DDIM-MCDO | 20 | 11.29 | **6.03** | **69.47** |
| DDIM | 10 | **9.78** | 12.35 | 66.30 |
| DDIM-MCDO | 10 | 10.22 | **7.13** | **67.95** |
| DDIM | 5 | 16.81 | 36.07 | 49.92 |
| DDIM-MCDO | 5 | **8.30** | **10.89** | **62.15** |

### 5.5 ABLATION STUDY

In this section, we investigate how the number of constraints affects image quality by varying the hyperparameter $t_{thre}$ from 1000 to 0. We evaluate the FID, sFID, and IS scores of 50,000 images generated with each $t_{thre}$ on ImageNet-256 (Russakovsky et al., 2015) using LDM-4 and 5-step DDIM samplers (Song et al., 2021).

Results in Table 5 show consistent improvements as $t_{thre}$ decreases. While increasing discrepancy in $x_0^{(t)}|_y$ at lower $t$ may introduce er-

Table 5: Result on ImageNet-256 with LDM-4. $T = 5$, $\eta = 0.0$, $s = 3.0$

| Method | FID↓ | sFID↓ | IS↑ | $t_{thre}$ |
|---|---|---|---|---|
| | 16.81 | 36.07 | 49.92 | 1000 |
| | 13.45 | 26.72 | 54.64 | 800 |
| DDIM-MCDO | 10.67 | 19.14 | 58.29 | 600 |
| | 9.38 | 15.32 | 60.29 | 400 |
| | 8.48 | 11.76 | 61.82 | 200 |
| | **8.30** | **10.89** | **62.15** | 0 |

rors in pixel variance correction, the benefits of DDIM-MCDO in later stages outweigh this issue, leading to better results than its early-stopping version (e.g., $t_{thre} = 200$).

## 6 CONCLUSION

In this paper, we address exposure bias in accelerated sampling of diffusion models. By analyzing this bias through the manifold hypothesis, we propose a manifold constraint that aligns deviated denoising observations with the manifold, enhancing both magnitude and direction accuracy in noise estimation. Our experiments on several large-resolution datasets demonstrate significant effectiveness. Furthermore, the method is training-free and requires minimal hyperparameter tuning, making it a simple plug-and-play module for other sampling methods.

ACKNOWLEDGMENTS

This work is funded by the National Natural Science Foundation of China (62403429), Zhejiang Provincial Natural Science Foundation of China under Grant (LQN25F030018, LQN25F030008).

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

# A    APPENDIX

## A.1    ZERO MEAN ASSUMPTION AND ERROR ANALYSIS

The noise added to clean data during the training phase follows normal distribution: $\epsilon_t \sim \mathcal{N}(\mathbf{0}, \mathbf{I})$, which implies $\mathbb{E}[\epsilon_t] = 0$. According to Equation 5, assuming $\mathbb{E}[\epsilon_\theta^t] = 0$ is reasonable for well-trained diffusion models. According to Equation 7, the predicted noisy data at timestep $t-1$ can be expressed as $\hat{\boldsymbol{x}}_{t-1} = \sqrt{\bar{\alpha}_t} \frac{\hat{\boldsymbol{x}}_t - \sqrt{1-\bar{\alpha}_t}\epsilon_\theta^{(t)}(\boldsymbol{x}_t)}{\sqrt{\bar{\alpha}_t}} + \sqrt{1 - \bar{\alpha}_{t-1} - \sigma_t^2}\epsilon_\theta^{(t)}(\boldsymbol{x}_t) + \sigma_t \epsilon_t', \epsilon_t' \sim \mathcal{N}(\mathbf{0}, \mathbf{I})$. Then as long as $\mathbb{E}[\frac{1}{n}\Sigma_{i=0}^n \boldsymbol{x}_{t_i}] = 0$, one has $\mathbb{E}[\frac{1}{n}\Sigma_{i=0}^n \hat{\boldsymbol{x}}_{t-1_i}] = 0, \forall t \in [1, T]$.

We illustrate the evolution of the average absolute pixel mean ($\mathbb{E}[|\Sigma_{i=0}^n \frac{1}{n} \cdot |]$) for the noisy data prediction, denoising observation, and noise estimation in Figure 9. The statistics were collected from 6,400 samples generated using a 100-step LDM on CelebA-HQ. It can be observed that the average absolute pixel mean values for $\hat{\boldsymbol{x}}_t$, $\hat{\boldsymbol{x}}_0^{(t)}$ and $\epsilon_\theta^{(t)}$ all remain close to zero throughout the generation process. This suggests that the pixel mean for all variables exhibits only minor deviations from 0 across timesteps.

In Equations 12 and 19, the square root of the pixel variance for $\hat{\boldsymbol{x}}_t$ and $\hat{\boldsymbol{x}}_0^{(t)}$ is used to approximate their $L^2$ norm: $\sqrt{n\mathrm{Var}(\hat{\boldsymbol{x}}_t)} \to \|\hat{\boldsymbol{x}}_t\|$, $\sqrt{n\mathrm{Var}(\hat{\boldsymbol{x}}_0^{(t)})} \to \|\hat{\boldsymbol{x}}_0^{(t)}\|$. We present the evolution of the $L^2$ norm alongside the pixel variance for the denoising observation in Figure 7. The relative error of this approximation is presented in Figure 8. The results demonstrate that throughout the entire generation process, the approximation error remains very small (less than 0.045). This indicates that using the pixel variance of $\hat{\boldsymbol{x}}_t$ or $\hat{\boldsymbol{x}}_0^{(t)}$ as an approximation for their $L^2$ norm introduces only minimal error, which is acceptable for practical applications.

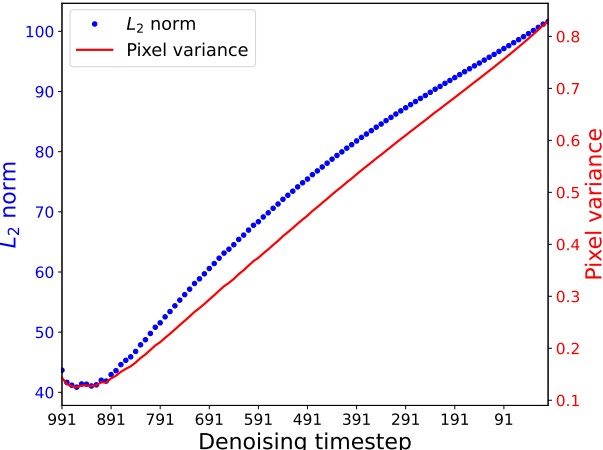

Figure 7: Evolution of the $L^2$ norm and pixel variance for the denoising observation, based on average statistics from 6,400 samples generated using 100-step LDM on CelebA-HQ.

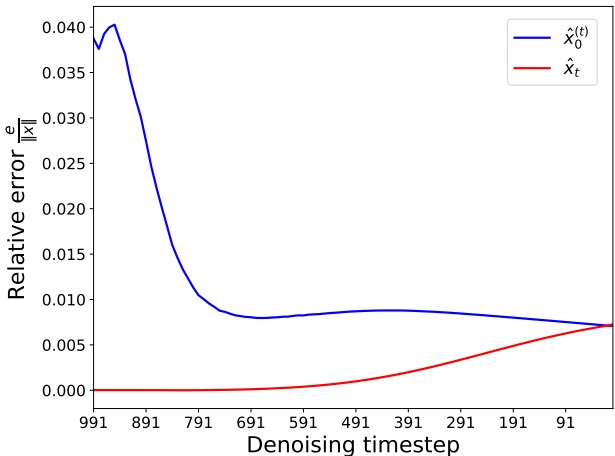

Figure 8: Relative error of using square root of pixel variance to approximate the $L^2$ norm of $\hat{\boldsymbol{x}}_t$ and $\hat{\boldsymbol{x}}_0^{(t)}$. Statistics are collected from 6,400 samples generated using 100-step LDM on CelebA-HQ.

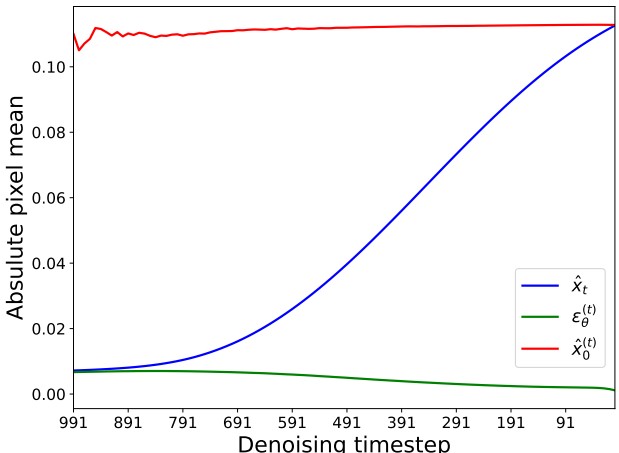

Figure 9: Evolution of the average absolute pixel mean for the noisy data prediction, denoising observation, and noise estimation. Statistics are collected from 6,400 samples generated using 100-step LDM on CelebA-HQ.

## A.2 DERIVATION OF NOISY RATIO IN A SINGLE STEP FOR DDIM

In the forward process of DDIM, we have:

$$\sqrt{\bar{\alpha}_t}\boldsymbol{x}_0 + \sqrt{1-\bar{\alpha}_t}\boldsymbol{\epsilon} = \boldsymbol{x}_t, \boldsymbol{\epsilon} \in \mathcal{N}(\boldsymbol{0}, \boldsymbol{I}). \tag{23}$$

In the reverse process, when given the ground truth input $\boldsymbol{x}_{t+1}$, the model's prediction of $\boldsymbol{x}_t$ is:

$$\hat{\boldsymbol{x}}_t = \sqrt{\bar{\alpha}_t}\hat{\boldsymbol{x}}_0^{(t+1)} + \sqrt{1-\bar{\alpha}_t-\sigma_{t+1}^2}\boldsymbol{\epsilon}_\theta^{(t+1)}(\boldsymbol{x}_{t+1}) + \sigma_{t+1}\boldsymbol{\epsilon}_{t+1}', \boldsymbol{\epsilon}_{t+1}' \sim \mathcal{N}(\boldsymbol{0}, \boldsymbol{I}). \tag{24}$$

When $\sigma_t = 0$ for all $t$, the sampling process become deterministic, and Equation 24 is simplified as:

$$\hat{\boldsymbol{x}}_t = \sqrt{\bar{\alpha}_t}\hat{\boldsymbol{x}}_0^{(t+1)} + \sqrt{1-\bar{\alpha}_t}\boldsymbol{\epsilon}_\theta^{(t+1)}(\boldsymbol{x}_{t+1}). \tag{25}$$

For timestep $t$, by denoting the prediction error on the noise as $\boldsymbol{e}_\theta^{(t)}$, we have:

$$\boldsymbol{\epsilon}_\theta^{(t)}(\boldsymbol{x}_t) = \boldsymbol{\epsilon}_t + \boldsymbol{e}_\theta^{(t)}. \tag{26}$$

We now derive the error on denoising observation with respect to the error on the noise estimation. By plugging Eq 26 into Equation 6, we have:

$$\begin{aligned}
\hat{\boldsymbol{x}}_0^{(t+1)} &= \frac{\boldsymbol{x}_{t+1} - \sqrt{1-\bar{\alpha}_{t+1}}\boldsymbol{\epsilon}_\theta^{(t+1)}(\boldsymbol{x}_{t+1})}{\sqrt{\bar{\alpha}_{t+1}}} \\
&= \frac{\boldsymbol{x}_{t+1} - \sqrt{1-\bar{\alpha}_{t+1}}\boldsymbol{\epsilon}_{t+1}}{\sqrt{\bar{\alpha}_{t+1}}} - \frac{\sqrt{1-\bar{\alpha}_{t+1}}}{\sqrt{\bar{\alpha}_{t+1}}}\boldsymbol{e}_\theta^{(t+1)} \\
&= \boldsymbol{x}_0 - \frac{\sqrt{1-\bar{\alpha}_{t+1}}}{\sqrt{\bar{\alpha}_{t+1}}}\boldsymbol{e}_\theta^{(t+1)}.
\end{aligned} \tag{27}$$

It is worth noting that in the early stage of generation, e.g., when $t = T$, $\bar{\alpha}_t$ approaches $0$. Therefore, according to Equation 27, the error coefficient on the denoising observation: $d_t = \frac{\sqrt{1-\bar{\alpha}_{t+1}}}{\sqrt{\bar{\alpha}_t}}$ is non-negligible.

As $\hat{\boldsymbol{x}}_t^{(0)}$ is notably influenced by the amplified noise prediction error term $d_t\boldsymbol{e}_\theta^{(t)}$ at most timesteps, it is possible to correct the noise estimation error by correcting the denoising observation.

With the same manner, we derive the error on the noisy data prediction with respect to the noise estimation error. By plugging X Equations 26 and 27 into Equation 25, we have:

$$\begin{aligned}
\hat{\boldsymbol{x}}_t &= \sqrt{\bar{\alpha}_t}\left(\boldsymbol{x}_0 - \frac{\sqrt{1-\bar{\alpha}_{t+1}}}{\sqrt{\bar{\alpha}_{t+1}}}\boldsymbol{e}_\theta^{(t+1)}\right) + \sqrt{1-\bar{\alpha}_t}\boldsymbol{\epsilon}_\theta^{(t+1)}(\boldsymbol{x}_{t+1}) \\
&= \sqrt{\bar{\alpha}_t}\boldsymbol{x}_0 - \sqrt{\bar{\alpha}_t}\frac{\sqrt{1-\bar{\alpha}_{t+1}}}{\sqrt{\bar{\alpha}_{t+1}}}\boldsymbol{e}_\theta^{(t+1)} + \sqrt{1-\bar{\alpha}_t}\boldsymbol{\epsilon}_{t+1} + \sqrt{1-\bar{\alpha}_t}\boldsymbol{e}_\theta^{(t+1)} \\
&= \left(\sqrt{\bar{\alpha}_t}\boldsymbol{x}_0 + \sqrt{1-\bar{\alpha}_t}\boldsymbol{\epsilon}_{t+1}\right) + \left(\sqrt{1-\bar{\alpha}_t} - \sqrt{\bar{\alpha}_t}\frac{\sqrt{1-\bar{\alpha}_{t+1}}}{\sqrt{\bar{\alpha}_{t+1}}}\right)\boldsymbol{e}_\theta^{(t+1)} \\
&= \boldsymbol{x}_t + \left(\sqrt{1-\bar{\alpha}_t} - \sqrt{\bar{\alpha}_t}\frac{\sqrt{1-\bar{\alpha}_{t+1}}}{\sqrt{\bar{\alpha}_{t+1}}}\right)\boldsymbol{e}_\theta^{(t+1)}.
\end{aligned} \tag{28}$$

We denote the scaling factor for $\boldsymbol{x}_t$ as $n_t$, $n_t = \sqrt{1-\bar{\alpha}_t} - \sqrt{\bar{\alpha}_t}\frac{\sqrt{1-\bar{\alpha}_{t+1}}}{\sqrt{\bar{\alpha}_{t+1}}}$. The value of both $d_t$ and $n_t$ during denoising process are shown in Figures 10 and 11. It is illustrated that while $\boldsymbol{e}_\theta^{(t)}$ is scaled down for $\boldsymbol{x}_t$, the same error is scaled up for $\boldsymbol{x}_0^{(t)}$.

Correcting the pixel variance of $\hat{\boldsymbol{x}}_t$ using a scale factor (directly correcting the pixel variance of the noisy data prediction) will eventually scale the entire term $\boldsymbol{x}_t + n_t\boldsymbol{\epsilon}_\theta^{(t)}$, which leads to distortion of the $\boldsymbol{x}_t$ component in $\hat{\boldsymbol{x}}_t$. When correcting the pixel variance of $\hat{\boldsymbol{x}}_0^{(t)}$ using the proposed method, one corrects $\boldsymbol{x}_0 + d_t\boldsymbol{e}_\theta^{(t)}$. While the $\boldsymbol{x}_0^{(t)}$ term is also influenced, it is important to note that by recalculating the noise estimation with the corrected denoising observation in Equation 6, the prediction error is reduced while the original $\boldsymbol{x}_t$ is preserved.

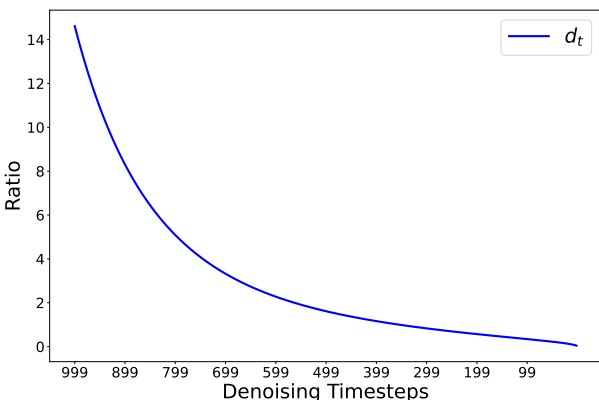

Figure 10: Error scale for denoising observation

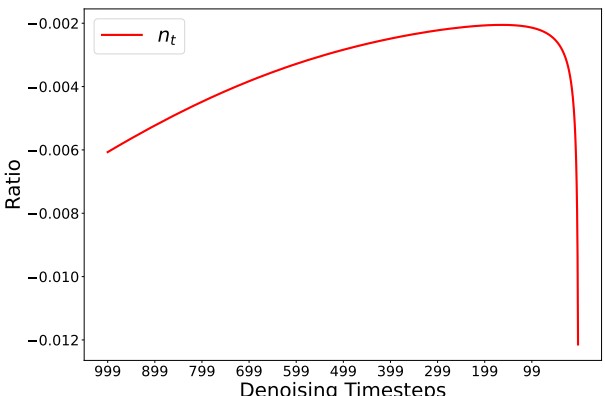

Figure 11: Error scale for noisy data

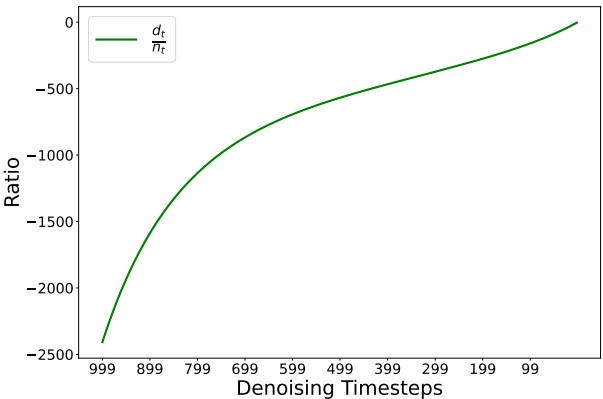

Figure 12: Ratio of error scale

## A.3 COMPARISON BETWEEN NOISY DATA AND DENOISING OBSERVATION CORRECTION

According to Equation. 16, adjusting the pixel variance of noisy data $\hat{x}_t$ during the accelerated sampling process to variance collected from sampling process with large enough $T$, could potentially reduce the exposure bias and improve the generation results. For clarity, we refer to this method as Noisy Data Variance Adjustment (NDVA). We provide the results of NDVA in Tables 6 and 7.

In Table 6, it can be observed that NDVA outperforms the baseline method, and exhibits performance comparable to TS-DDIM when $T = 20$. However, it is notably worse than MCDO as the number of timesteps further decreases.

Table 6: Results on CelebA-HQ dataset.

| Method | Steps | FID↓ |
|---|---|---|
| DDIM | 20 | 10.59 |
| TS-DDIM | 20 | **8.29** |
| DDIM-NVDA | 20 | 9.21 |
| DDIM-MCDO | 20 | 9.49 |
| DDIM | 10 | 21.08 |
| TS-DDIM | 10 | 15.71 |
| DDIM-NVDA | 10 | 16.45 |
| DDIM-MCDO | 10 | **15.63** |
| DDIM | 5 | 54.99 |
| TS-DDIM | 5 | 50.69 |
| DDIM-NVDA | 5 | 47.52 |
| DDIM-MCDO | 5 | **30.76** |

Table 7: Results on MS-COCO.

| Method | Steps | FID↓ |
|---|---|---|
| DPM-Solver++(2M) | 10 | 15.10 |
| DPM-Solver++(2M) + NVDA | 10 | 15.76 |
| DPM-Solver++(2M) + MCDO | 10 | **14.88** |
| DPM-Solver++(2M) | 8 | 15.57 |
| DPM-Solver++(2M) + NVDA | 8 | 15.76 |
| DPM-Solver++(2M) + MCDO | 8 | **15.32** |
| DPM-Solver++(2M) | 6 | 17.05 |
| DPM-Solver++(2M) + NVDA | 6 | 17.13 |
| DPM-Solver++(2M) + MCDO | 6 | **16.39** |
| DPM-Solver++(2M) | 5 | 19.16 |
| DPM-Solver++(2M) + NVDA | 5 | 19.02 |
| DPM-Solver++(2M) + MCDO | 5 | **18.51** |

## A.4 EXPOSURE BIAS MEASUREMENT

In this section, we employ the exposure bias metric proposed by Ning et al. (2024) to evaluate the single step variance discrepancy reduction when using our method. The statistics are collected during the accelerated diffusion sampling (5, 10, 20 steps) on Stable Diffusion XL. The variance deviation with respect to the fixed schedule $1 - \bar{\alpha}_t$, as well as the standard deviation are illustrated in Figure 13. Although our approach is based on the manifold hypothesis and the concept of pixel variance, the results indicate that the proposed manifold constraint still reduces sample variance error.

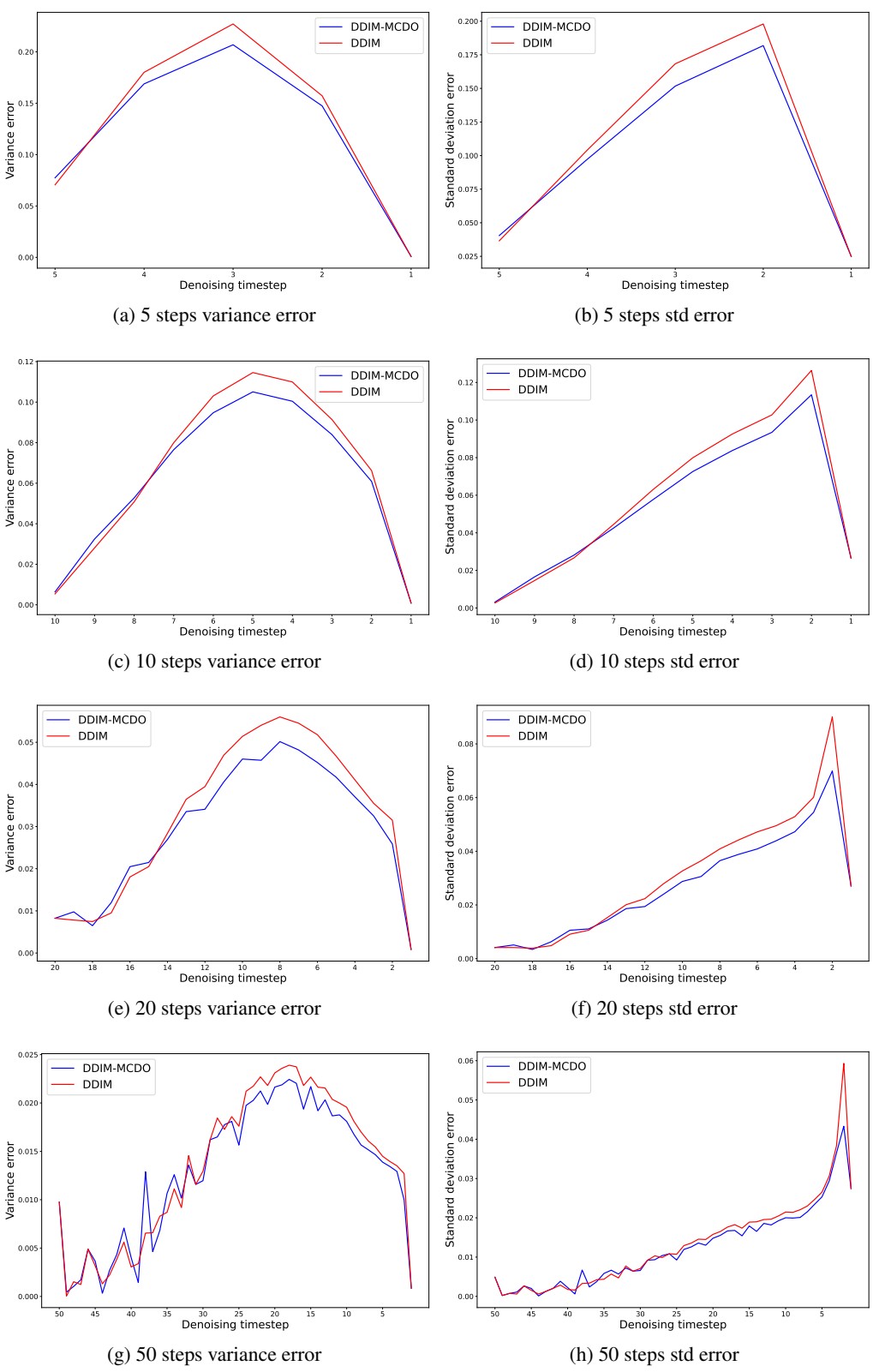

Figure 13: Exposure bias measurement. We plot the variance error and standard deviation error in single-step samplings.

### A.5    MORE QUALITATIVE RESULTS ON MS-COCO USING SDXL

We provide more qualitative results on text-to-image generation in Figure 14.

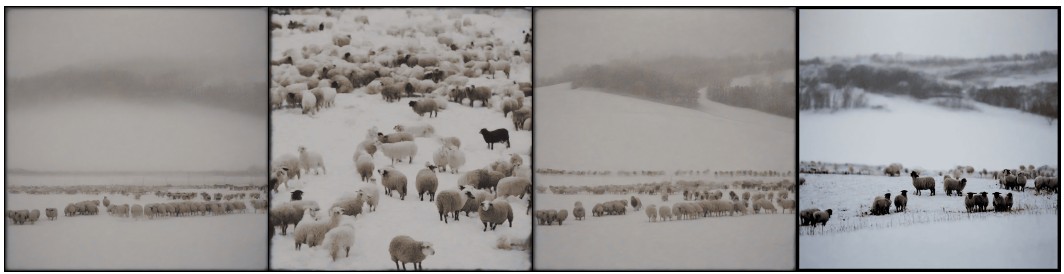

(a) *"A bunch of sheep are standing in a snowy field."*

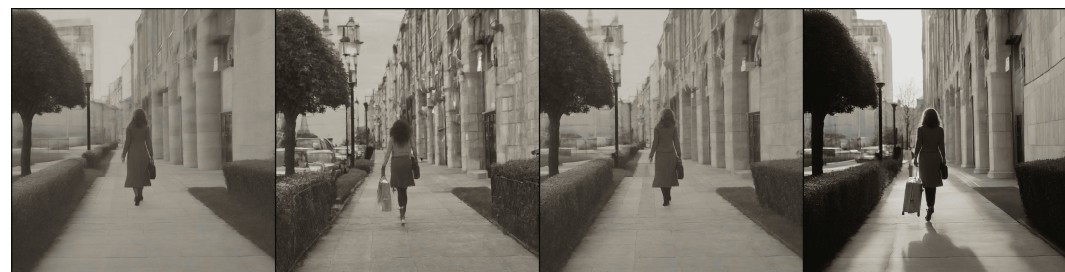

(b) *"a woman walking along a sidewalk while holding a suitcase"*

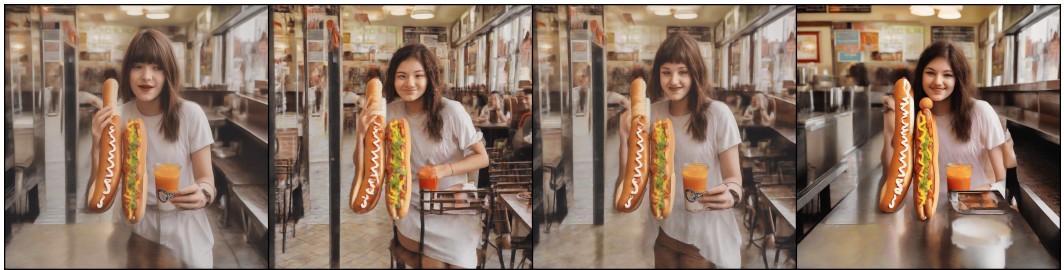

(c) *"A girl holding a foot long hot dog in a restaurant"*

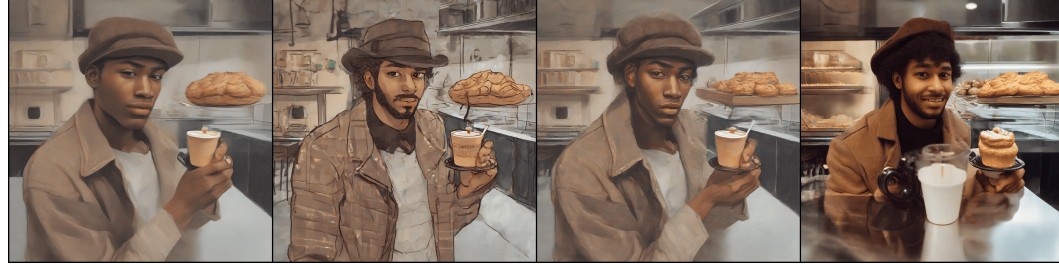

(d) *"A young man in a hat holding a cup of coffee and a pastry."*

Figure 14: Images generated using 10 steps SDXL with DDIM sampler. From left to right: SDXL, TS-DDIM (Li et al., 2024a), LDM-ES (Ning et al., 2024), DDIM-MCDO (ours).

### A.6    QUALITATIVE RESULTS ON CELEBA-HQ

We present qualitative results for unconditional generation on CelebA-HQ using LDM-4 in Figures 15, 16, and 17, with sampling steps of 5, 10, and 20, respectively. For the sampling settings, we use the recommended value of $\eta = 0.0$ when the number of steps is low (Rombach et al., 2022).

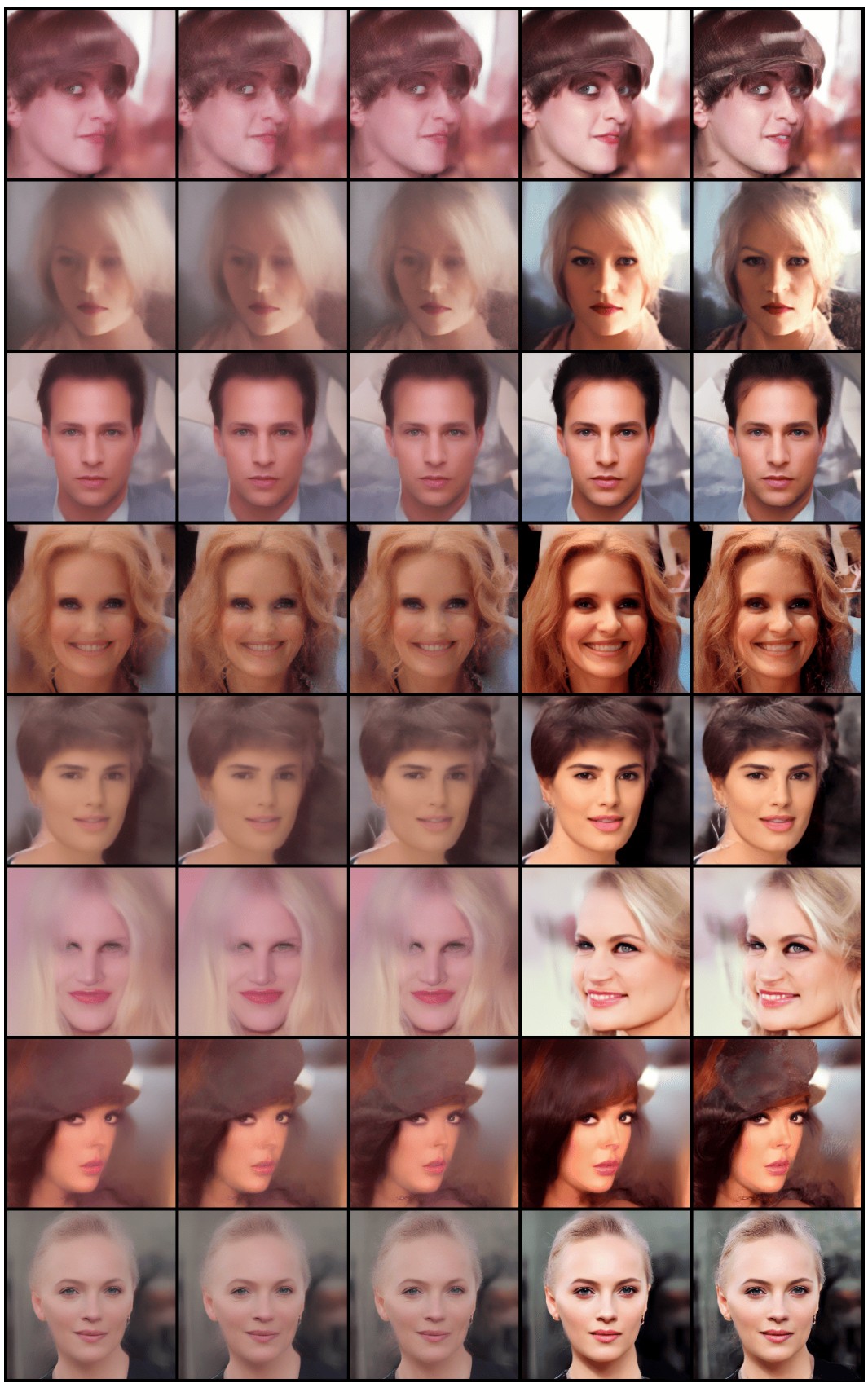

Figure 15: Samples on CelebA-HQ $256 \times 256$ using 5 steps LDM-4. From left to right: DDIM, TS-DDIM (Li et al., 2024a), LDM-ES (Ning et al., 2024) with $\lambda = 1.005$, DDIM-MCDO, DDIM-MCDO†with $\lambda = 1.005$.

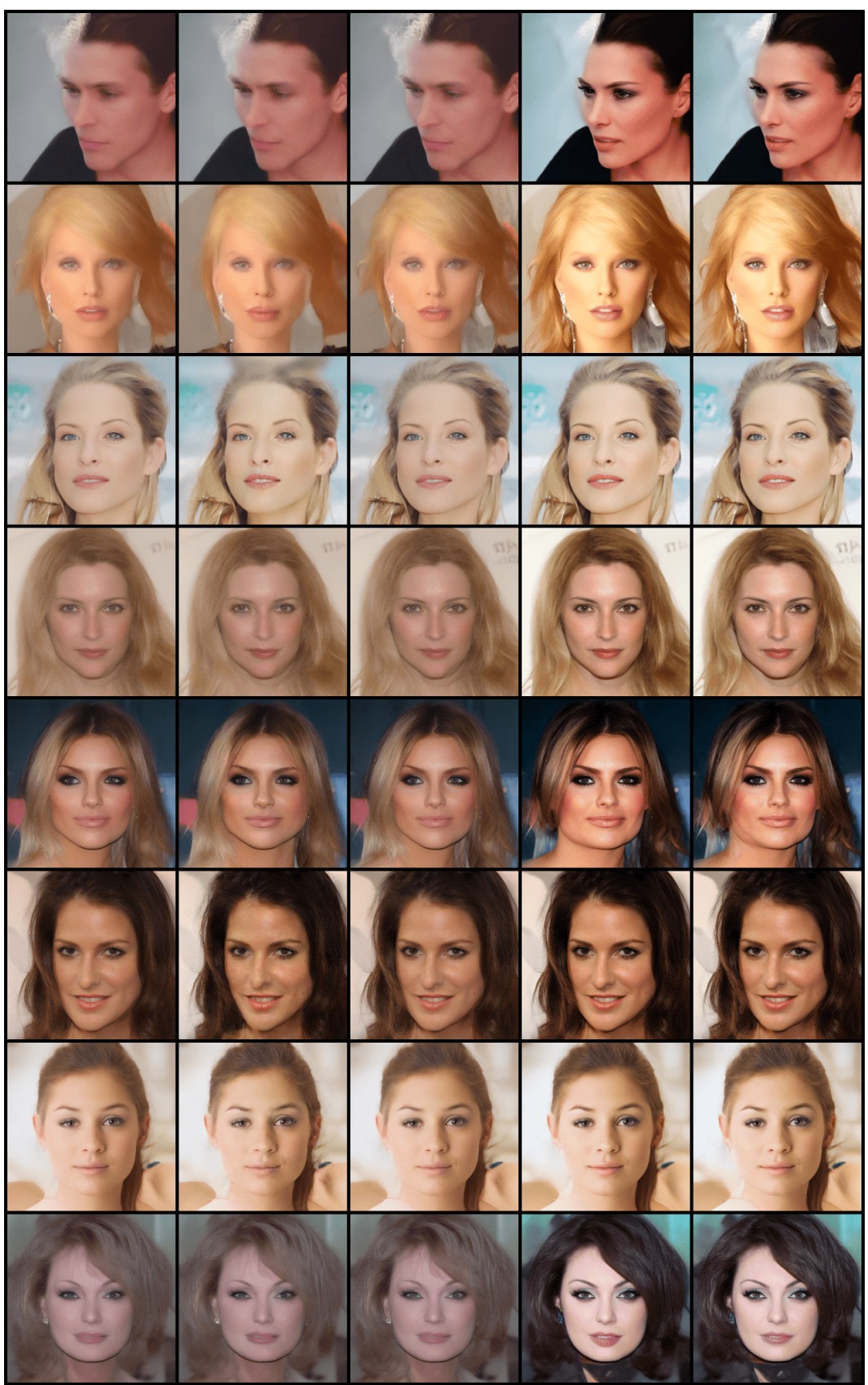

Figure 16: Samples on CelebA-HQ $256 \times 256$ using 10 steps LDM-4. From left to right: DDIM, TS-DDIM (Li et al., 2024a), LDM-ES (Ning et al., 2024) with $\lambda = 1.002$, DDIM-MCDO, DDIM-MCDO†with $\lambda = 1.002$.

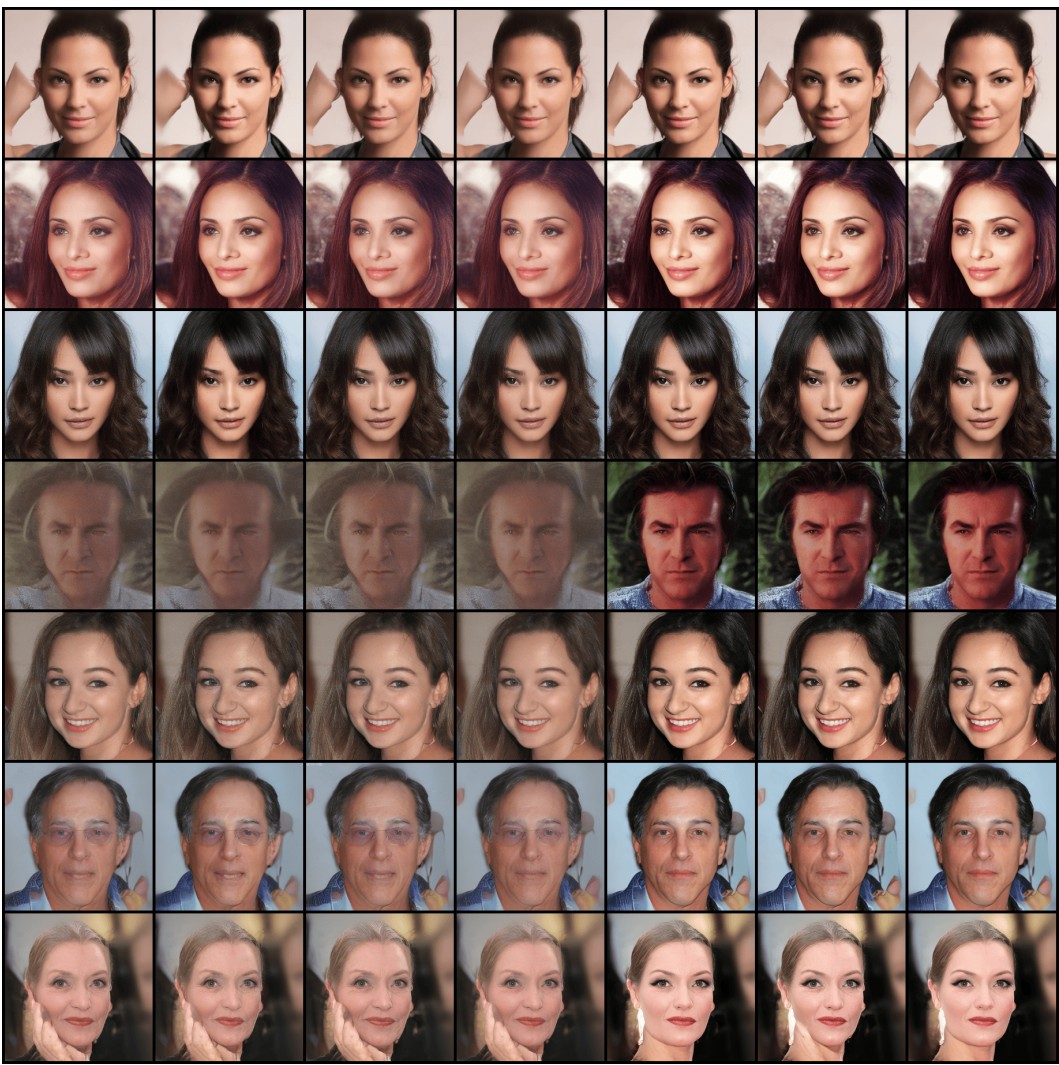

Figure 17: Samples on CelebA-HQ $256 \times 256$ using 20 steps LDM-4. From left to right: DDIM, TS-DDIM (Li et al., 2024a), LDM-ES (Ning et al., 2024) with $\lambda = 0.999$, LDM-ES (Ning et al., 2024) with $\lambda = 1.001$, DDIM-MCDO, DDIM-MCDO†with $\lambda = 0.999$, DDIM-MCDO with $\lambda = 1.001$.

## A.7 PIXEL-LEVEL CONSTRAINT FOR UNCONDITIONAL GENERATION

Considering that in unconditional generation, the similarity between samples within an inference batch is higher than that in Text-to-Image generation, we propose a more fine-grained correction approach: to estimate the pixel-level variance of the denoising observation:$\boldsymbol{v}_{t,c,h,w} = \mathrm{Var}(\hat{\boldsymbol{x}}_{0_{i,c,h,w}}^{(t)})$, $\boldsymbol{v}_t \in \mathbb{R}^{C,H,W}$, $\hat{\boldsymbol{x}}_0^{(t)} \in \mathbb{R}^{B,C,H,W}$. Where $B$ is the batch size, $C$ is the channel number, $H$ and $W$ are the latent's height and weight, respectively. We denote this variant as DDIM-MCDO*. When DDIM-MCDO* is coupled with LDM-ES (Ning et al., 2024), it is denoted as DDIM-MCDO‡[2] The batch size $N$ used for implementing DDIM-MCDO* is 64 on CelebA-HQ.

## A.8 QUANTITATIVE RESULTS ON CELEBA-HQ WITH DDPM SAMPLER

We provide results on CelebA-HQ using LDM-4 and DDPM sampler, where $\eta = 1.0$, in Table 8. We notice that contrast to LDM-ES (Ning et al., 2024), our method yields more performance improvement for DDIM than DDPM. Considering the DDIM setting is recommended under accelerated generation (Rombach et al., 2022), results in Table 8 suggests that the proposed methods might be easily combined with other fast sampling method in scenarios where $T'$ is extremely low.

Table 8: Results on CelebA-HQ $256 \times 256$ with LDM-4 and DDPM sampler, $\eta = 1.0$.

| Method | $T'$ | FID↓ | $\lambda$ |
|---|---|---|---|
| DDPM | 20 | 29.61 | – |
| LDM-ES (Ning et al., 2024) | 20 | **15.68** | 1.01 |
| DDPM-MCDO | 20 | 26.83 | – |
| DDPM-MCDO* | 20 | 23.84 | – |
| DDPM-MCDO† | 20 | 16.47 | 1.01 |
| DDPM-MCDO‡ | 20 | 19.21 | 1.01 |
| DDPM | 10 | 56.22 | – |
| LDM-ES (Ning et al., 2024) | 10 | 33.36 | 1.03 |
| DDPM-MCDO | 10 | 47.29 | – |
| DDPM-MCDO* | 10 | 29.76 | – |
| DDPM-MCDO† | 10 | 19.87 | 1.03 |
| DDPM-MCDO‡ | 10 | **16.92** | 1.03 |
| DDPM | 5 | 102.1 | – |
| LDM-ES (Ning et al., 2024) | 5 | 73.49 | 1.04 |
| DDPM-MCDO | 5 | 78.12 | – |
| DDPM-MCDO* | 5 | 54.56 | – |
| DDPM-MCDO† | 5 | 56.40 | 1.04 |
| DDPM-MCDO‡ | 5 | **51.76** | 1.04 |

---

[2]‡DDIM-MCDO* combined with LDM-ES using the same hyperparameter.

### A.9 QUALITATIVE RESULTS ON LSUN-BEDROOM

We present qualitative results for unconditional generation on LSUN-Bedroom using LDM-4 in Figures 18, 19, and 20, with sampling steps of 4, 5, and 10, respectively. For the sampling settings, we use the recommended value of $\eta = 0.0$ when the number of steps is low (Rombach et al., 2022).

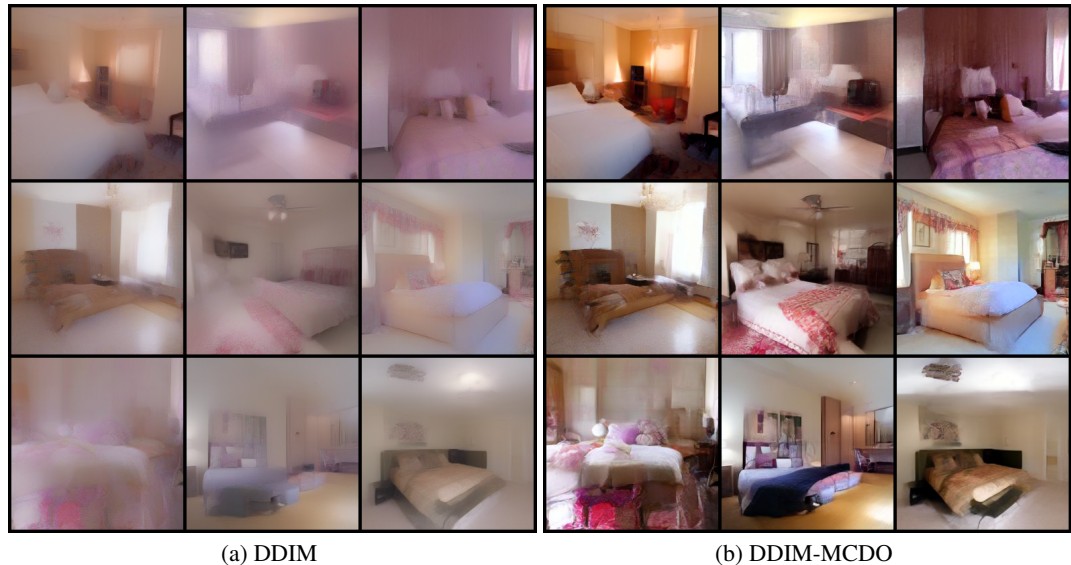

(a) DDIM          (b) DDIM-MCDO

Figure 18: Samples on LSUN-bedroom $256 \times 256$ using 4 steps LDM-4.

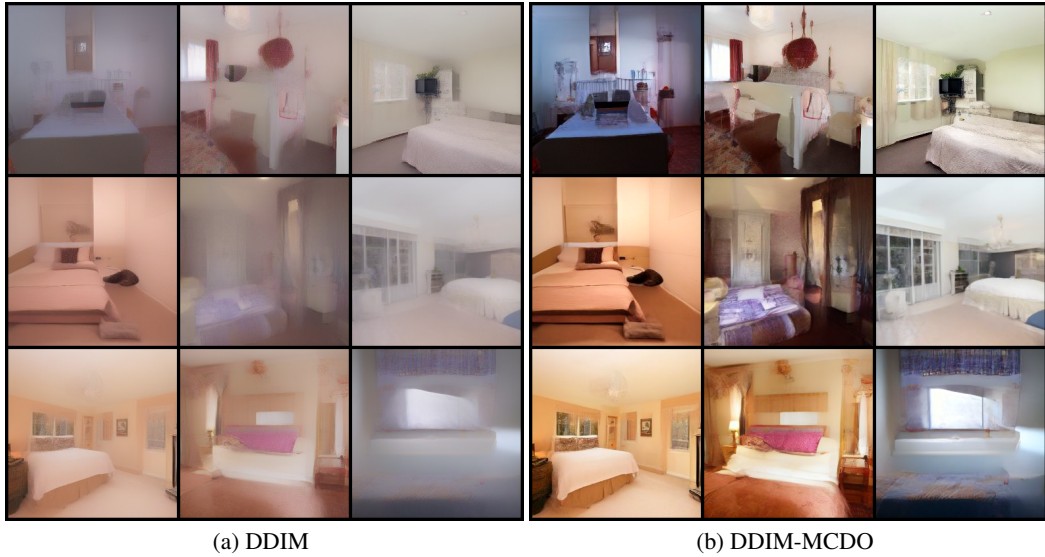

(a) DDIM          (b) DDIM-MCDO

Figure 19: Samples on LSUN-bedroom $256 \times 256$ using 5 steps LDM-4.

### A.10 QUALITATIVE RESULTS ON IMAGENET-256

We present qualitative results of class-conditional generation on ImageNet-256 using LDM-4 in Figures 21, 22, and 23, with sampling steps of 5, 10, and 20, respectively. For the sampling settings, we use the recommended value of $\eta = 0.0$ when the number of steps is low (Rombach et al., 2022).

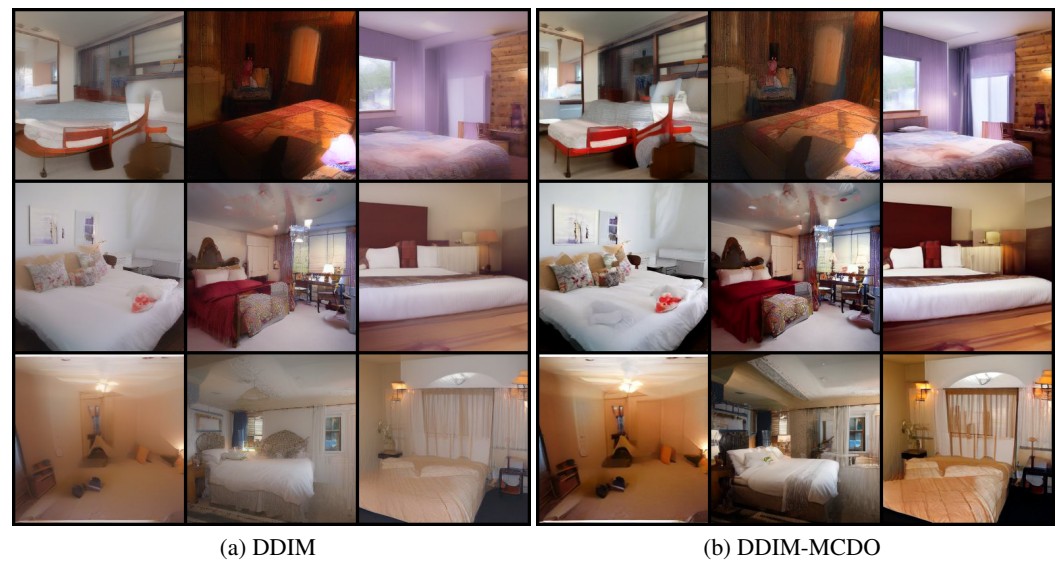

(a) DDIM                                        (b) DDIM-MCDO

Figure 20: Samples on LSUN-bedroom $256 \times 256$ using 10 steps LDM-4.

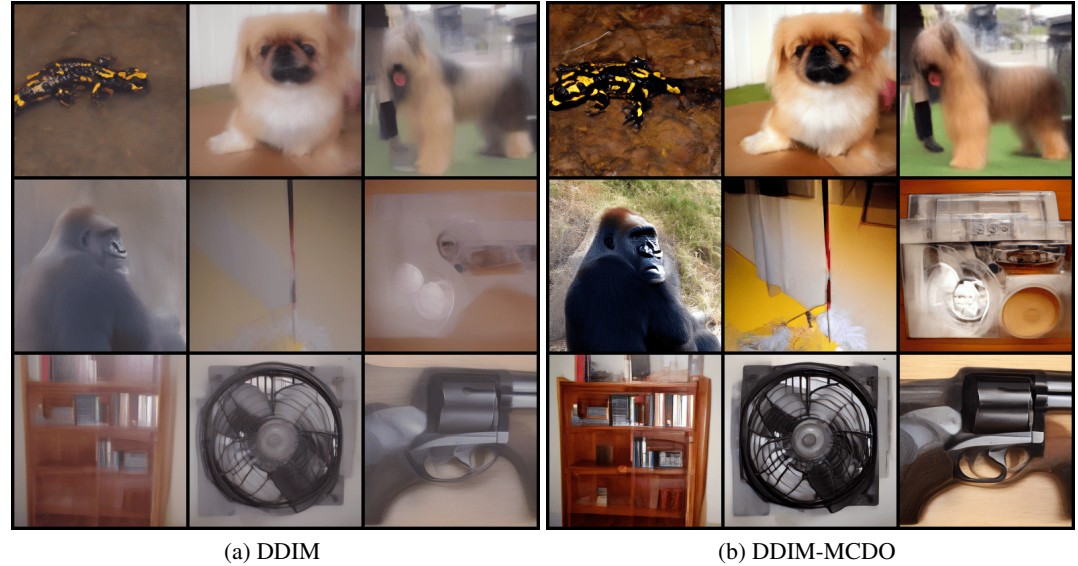

(a) DDIM                                        (b) DDIM-MCDO

Figure 21: Samples on ImageNet $256 \times 256$ using 5 steps LDM-4.

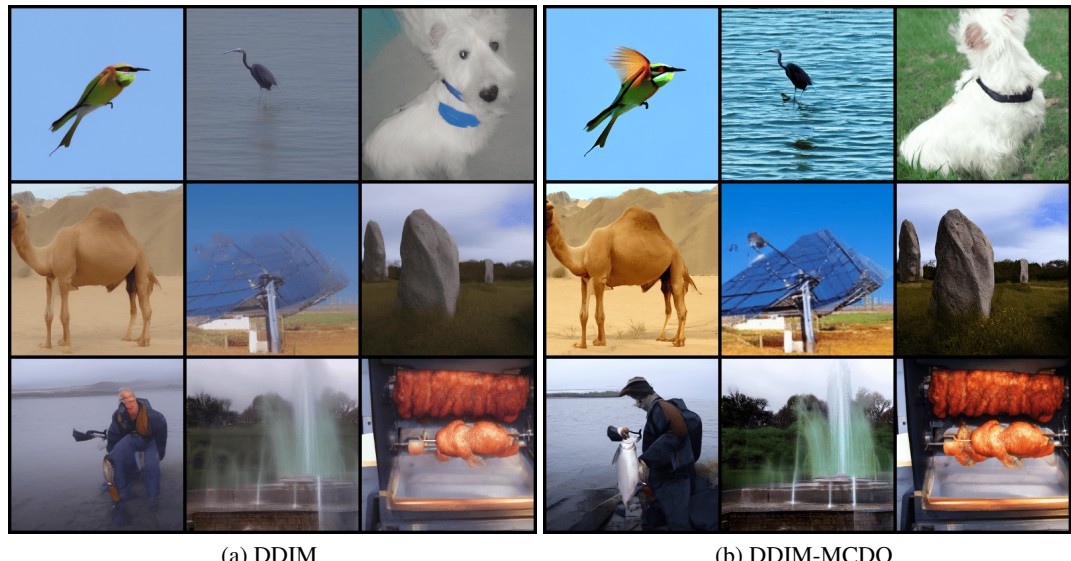

(a) DDIM                                        (b) DDIM-MCDO

Figure 22: Samples on ImageNet $256 \times 256$ using 10 steps LDM-4.

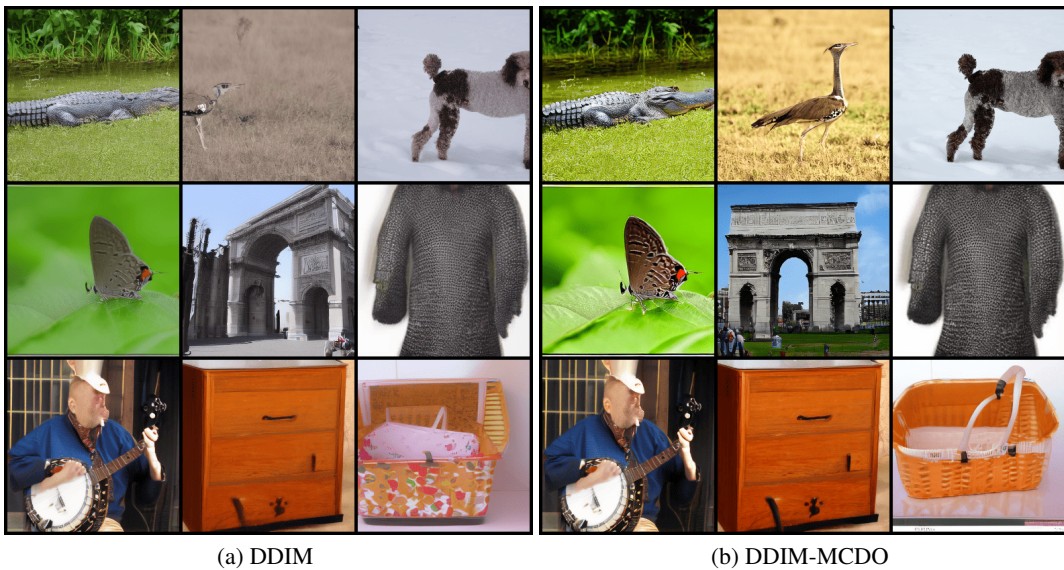

(a) DDIM                                        (b) DDIM-MCDO

Figure 23: Samples on ImageNet $256 \times 256$ using 20 steps LDM-4.

### A.11 A More Fine-Grained Manifold Constraint

In this section, based on the manifold analysis in Section 4.1, we provide a more fine-grained version of MCOD. Which is denoted as DDIM-MCDO+. The statistics collection process of DDIM-MCDO+ and DDIM-MCDO are shared, so DDIM-MCDO+ can be implemented by adding one more line of code.

During inference, we first correct the pixel variance of the denosing observation $\hat{x}_0^{(t)}$ like in DDIM-MCDO:

$$\tilde{x}_0^{(t)}|_{y_2, x_{T_2}} = \sqrt{c_t}\left[\hat{x}_0^{(t)}|_{y_2, x_{T_2}} - \text{Mean}\left(\hat{x}_0^{(t)}|_{y_2, x_{T_2}}\right)\right] + \text{Mean}\left(\hat{x}_0^{(t)}|_{y_2, x_{T_2}}\right). \tag{29}$$

Then, we calculate the transition coefficient $d_{t-k}$ between $\mathcal{M}_0^{(t)}$ and $\mathcal{M}_0^{(t-k)}$:

$$d_{t-k} = \frac{\text{Var}(x_0^{\star(t-k)}|_{y_2, x_{T_2}})}{\text{Var}(x_0^{\star(t)}|_{y_2, x_{T_2}})}. \tag{30}$$

With $d_{t-k}$, we can further correct the denoising observation, which will be used in predicting $x_{t-k}$.

$$\tilde{x}_0^{(t-k)}|_{y_2, x_{T_2}} = \sqrt{d_{t-k}}\tilde{x}_0^{(t)}|_{y_2, x_{T_2}}. \tag{31}$$

The final expression of $\tilde{x}_{t-k}$:

$$\tilde{x}_{t-k} = \sqrt{\bar{\alpha}_{t-k}}\tilde{x}_0^{(t-k)} + \sqrt{1 - \bar{\alpha}_{t-k} - \sigma_{t-k}^2}\tilde{\epsilon}_\theta\left(x_t\right) + \sigma_{t-k}\epsilon_{t-k}', \epsilon_{t-k}' \sim \mathcal{N}\left(\mathbf{0}, \mathbf{I}\right). \tag{32}$$

The result on MS-COCO (Lin et al., 2014) dataset with Stable Diffusion XL (Podell et al., 2024) are provided in Figure 24 and Table 9. We find that although DDIM-MCDO+ does not outperform DDIM-MCDO on metrics like FID and CLIP score, the quantitative results of DDIM-MCDO+ might be better.

Table 9: Results on Text-to-Image generation on MS-COCO val2014 with SDXL and DDIM sampler. Image Resolution is $1024 \times 1024$.

| Model | Method | Steps | FID↓ | CLIP Score↑ | $t_{thre}$ |
|-------|--------|-------|------|-------------|-----------|
| SDXL | DDIM | 10 | 18.17 | 31.58 | – |
| | DDIM-MCDO | 10 | **15.60 (-2.75)** | **31.75 (+0.18)** | 0 |
| | DDIM-MCDO+ | 10 | 15.79 (-2.38) | 31.72 (+0.14) | 100 |
| | DDIM-MCDO+ | 10 | 15.98 (-2.19) | 31.71 (+0.13) | 0 |

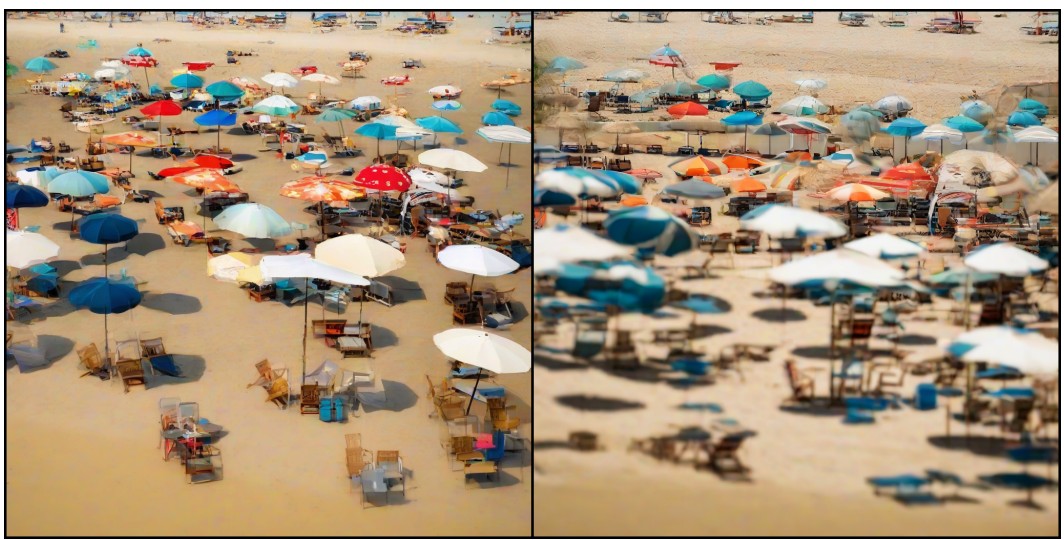

(a) *"A beach area with several chairs and umbrellas."*

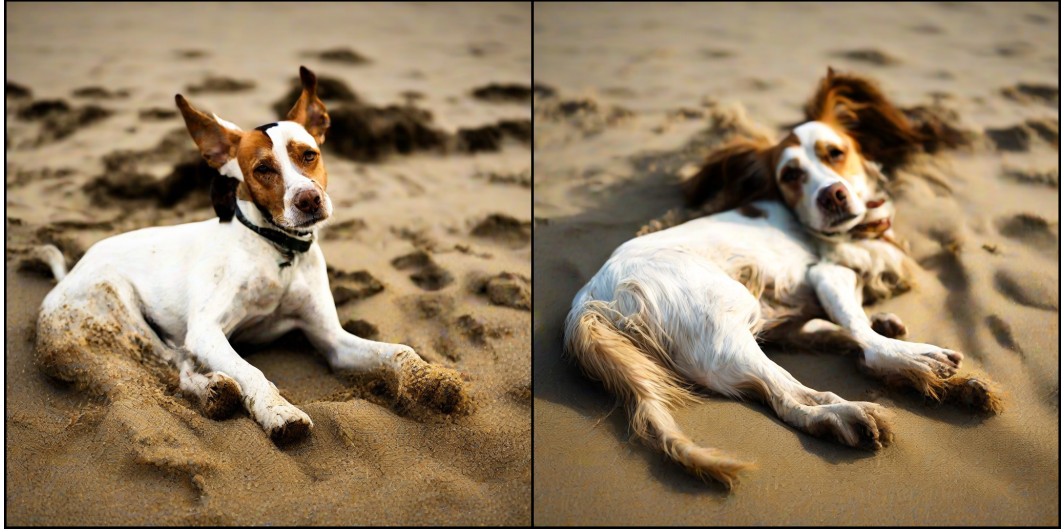

(b) *"A dog laying on the beach in the sand."*

Figure 24: Comparison between DDIM-MCDO+ (left) DDIM-MCDO (right) using 10 steps SDXL.

## A.12 RESULTS WITH DPM-SOLVER++

To further investigate the compatibility of MCOD with high-order solvers, we evaluate the proposed method with DPM-Solver++ (Lu et al., 2022b). The pixel variance of 60 samples, collected from the 1000 steps SDE-DPM-Solver++(2M) sampling process, were used for $v_t$ estimation. We generated 10k images for each evaluation in Table 10. The corresponding qualitative results are provided in Figures 25, 26, 27 and 28.

Table 10: Results on Text-to-Image generation on MS-COCO val2014 with SDXL and DPM-Solver++. Image Resolution is 1024 × 1024

| Method | Steps | FID↓ | CLIP Score↑ | $t_{thre}$ |
|---|---|---|---|---|
| DPM-Solver++(2M) | 10 | 15.10 | 31.74 | – |
| DPM-Solver++(2M)-MCDO | 10 | **14.88** | **31.85** | 100 |
| DPM-Solver++(2M) | 8 | 15.57 | 31.74 | – |
| DPM-Solver++(2M)-MCDO | 8 | **15.32** | **31.79** | 100 |
| DPM-Solver++(2M) | 6 | 17.05 | 31.65 | – |
| DPM-Solver++(2M)-MCDO | 6 | **16.39** | **31.72** | 100 |
| DPM-Solver++(2M) | 5 | 19.16 | 31.52 | – |
| DPM-Solver++(2M)-MCDO | 5 | **18.51** | **31.62** | 100 |

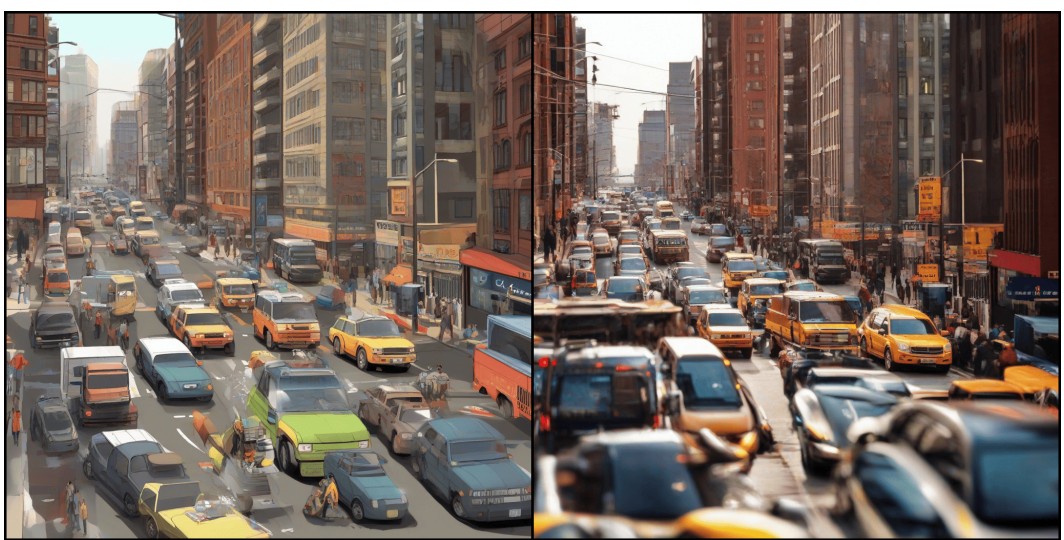

(a) *"A busy city street with many different vehicles."*

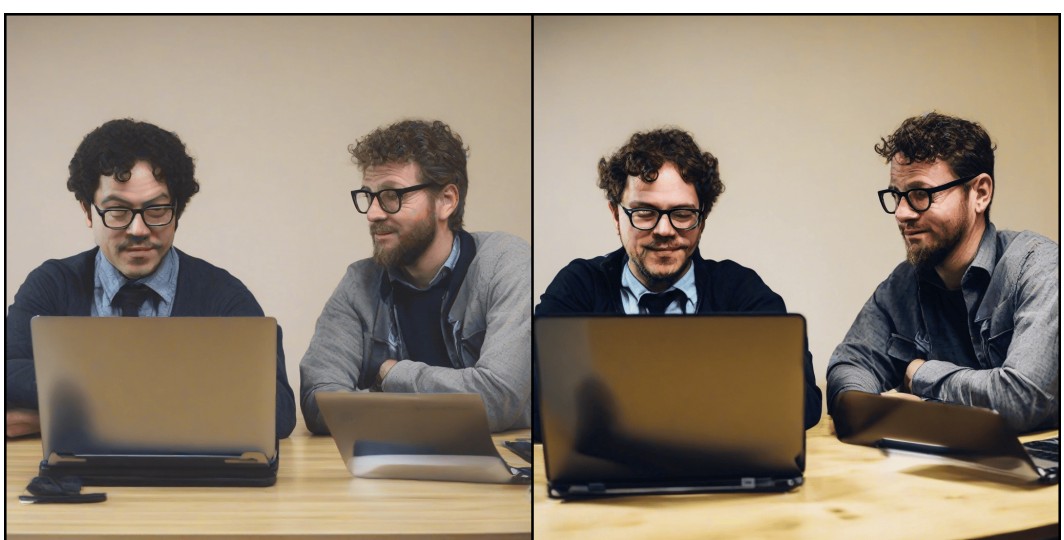

(b) *"Two men with glasses sitting at a table using laptops"*

Figure 25: Comparison between DPM-Solver++(2M) (Lu et al., 2022b) (left) DPM-Solver++(2M)-MCDO (right) using 10 steps SDXL.

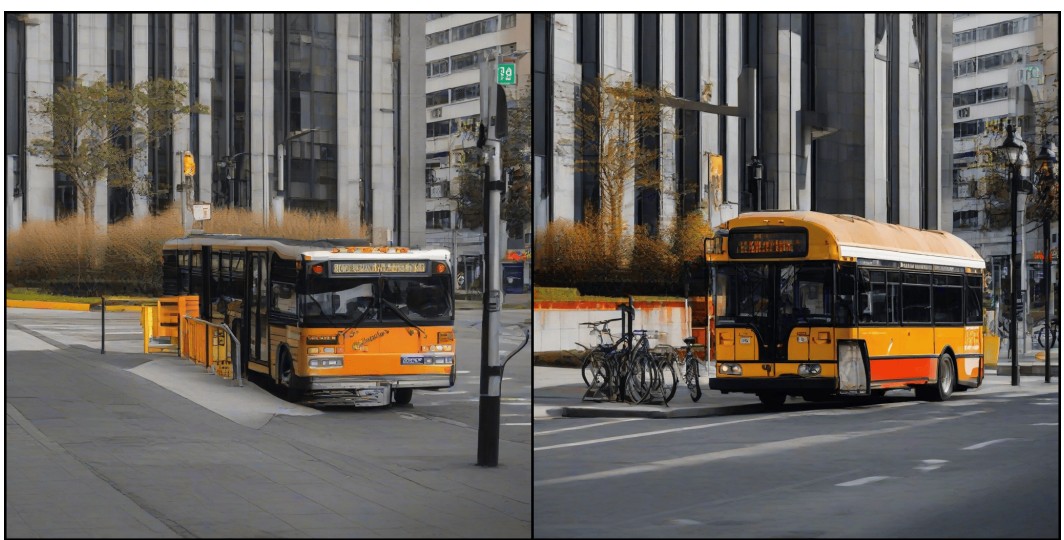

(a) *"A public transportation bus near a curb with a bicycle rack."*

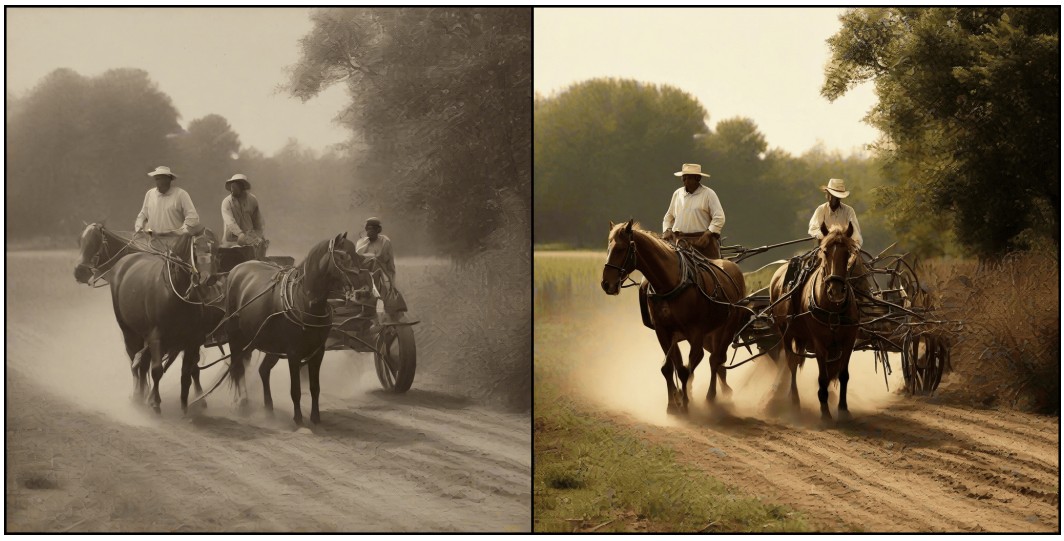

(b) *"Two horses plowing the land as a man directs them."*

Figure 26: Comparison between DPM-Solver++(2M) (Lu et al., 2022b) (left) DPM-Solver++(2M)-MCDO (right) using 8 steps SDXL.

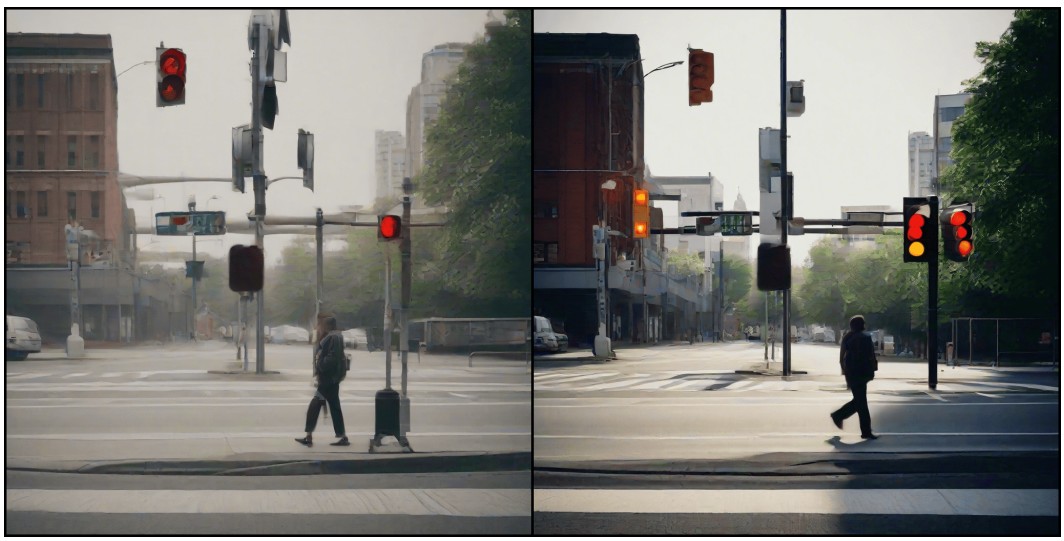

(a) *""A person walking across the street with a traffic light above."*

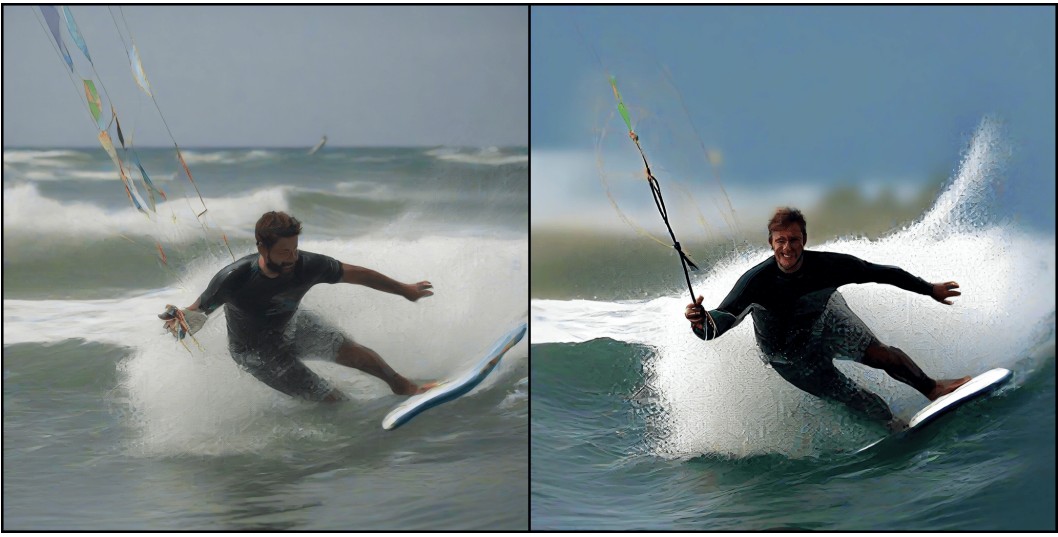

(b) *A man in the ocean surfing holding on to a kite."*

Figure 27: Comparison between DPM-Solver++(2M) (Lu et al., 2022b) (left) DPM-Solver++(2M)-MCDO (right) using 6 steps SDXL.

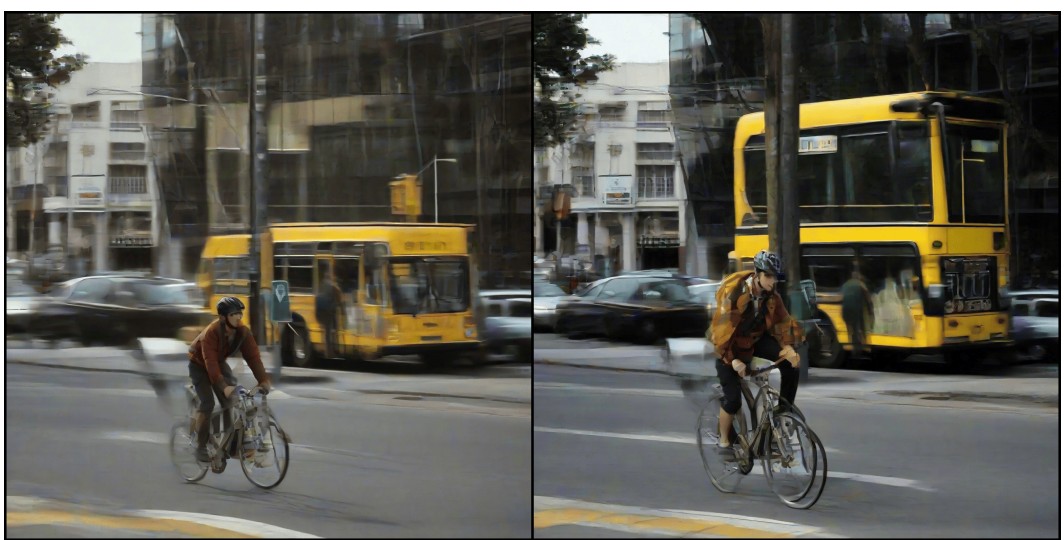

(a) *"A bicyclist is riding past a bus that is parked."*

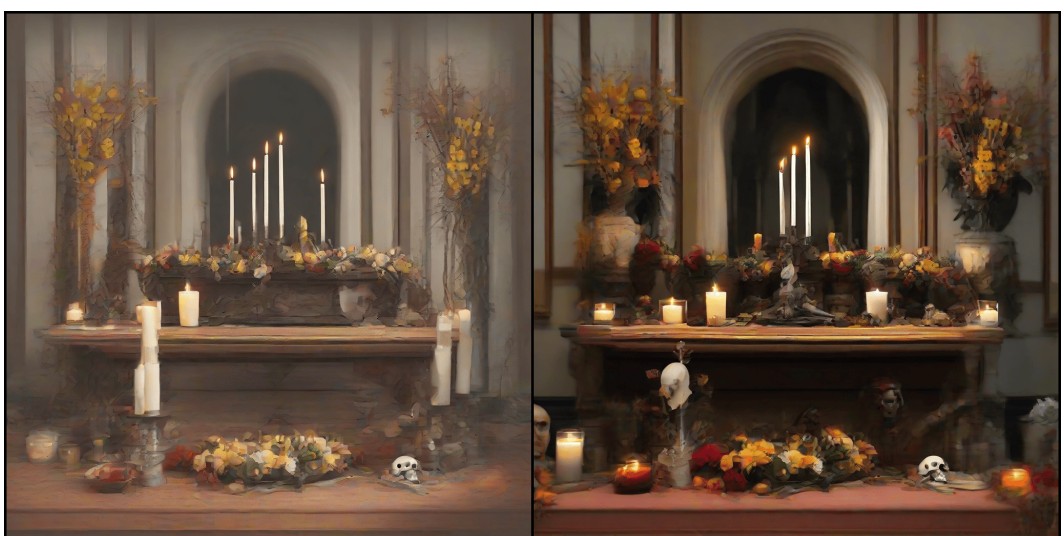

(b) *"A picture of an alter holding bananas, skeletons, candles and flowers."*

Figure 28: Comparison between DPM-Solver++(2M) (Lu et al., 2022b) (left) DPM-Solver++(2M)-MCDO (right) using 5 steps SDXL.

## A.13 Results on Stable Diffusion 3

We further evaluate the scalability on rectified flow model: Stable Diffusion 3 (Esser et al., 2024). Quantitative results, computed using 10k images, are provided as follows.

Table 11: Results on Text-to-Image generation on MS-COCO val2014 with Stable Diffusion 3 (Esser et al., 2024). Image Resolution is $1024 \times 1024$.

| Model | Method | Steps | FID↓ | CLIP Score↑ | $t_{thre}$ |
|---|---|---|---|---|---|
| | DDIM | 6 | 33.31 | 30.56 | – |
| | DDIM-MCDO | 6 | **27.57** | **31.18** | 0 |
| Stable Diffusion 3 | DDIM | 9 | 22.71 | 31.44 | – |
| | DDIM-MCDO | 9 | **22.26** | **31.70** | 0 |

## A.14 Results with Different Classifier-Free Guidance Scale

Considering that large classifier-free guidance (CFG) scales might cause exposure bias (Chung et al., 2025), we evaluate the proposed method, as the CFG scale varies. Quantitative results, computed using 10k images, are provided in Table 12.

Table 12: Results on Text-to-Image generation on MS-COCO val2014 with SDXL (Podell et al., 2024). Image Resolution is $1024 \times 1024$.

| Method | CFG Scale | FID↓ | CLIP Score↑ | $t_{thre}$ |
|---|---|---|---|---|
| DDIM | 8 | 18.51 | 31.81 | – |
| DDIM-MCDO | 8 | **17.38** | **32.03** | 100 |
| DDIM | 10 | 18.99 | 31.82 | – |
| DDIM-MCDO | 10 | **17.49** | **32.13** | 100 |
| DDIM | 12 | 20.62 | 31.70 | – |
| DDIM-MCDO | 12 | **17.80** | **32.18** | 100 |

The results show that incorporating the proposed method into sampling significantly enhances performance across varying CFG scales. Specifically, while higher CFG scales (e.g., 12) typically exacerbate exposure bias, the DDIM-MCDO method effectively mitigates this issue, reducing the FID score by 2.82.

## A.15   QUANTITATIVE COMPARISON WITH 1,000 STEPS RESULTS

We provide quantitative comparison between images generated using 10 steps DDIM, 10 steps DDIM-MCDO, and 1,000 steps DDIM in Figure 29.

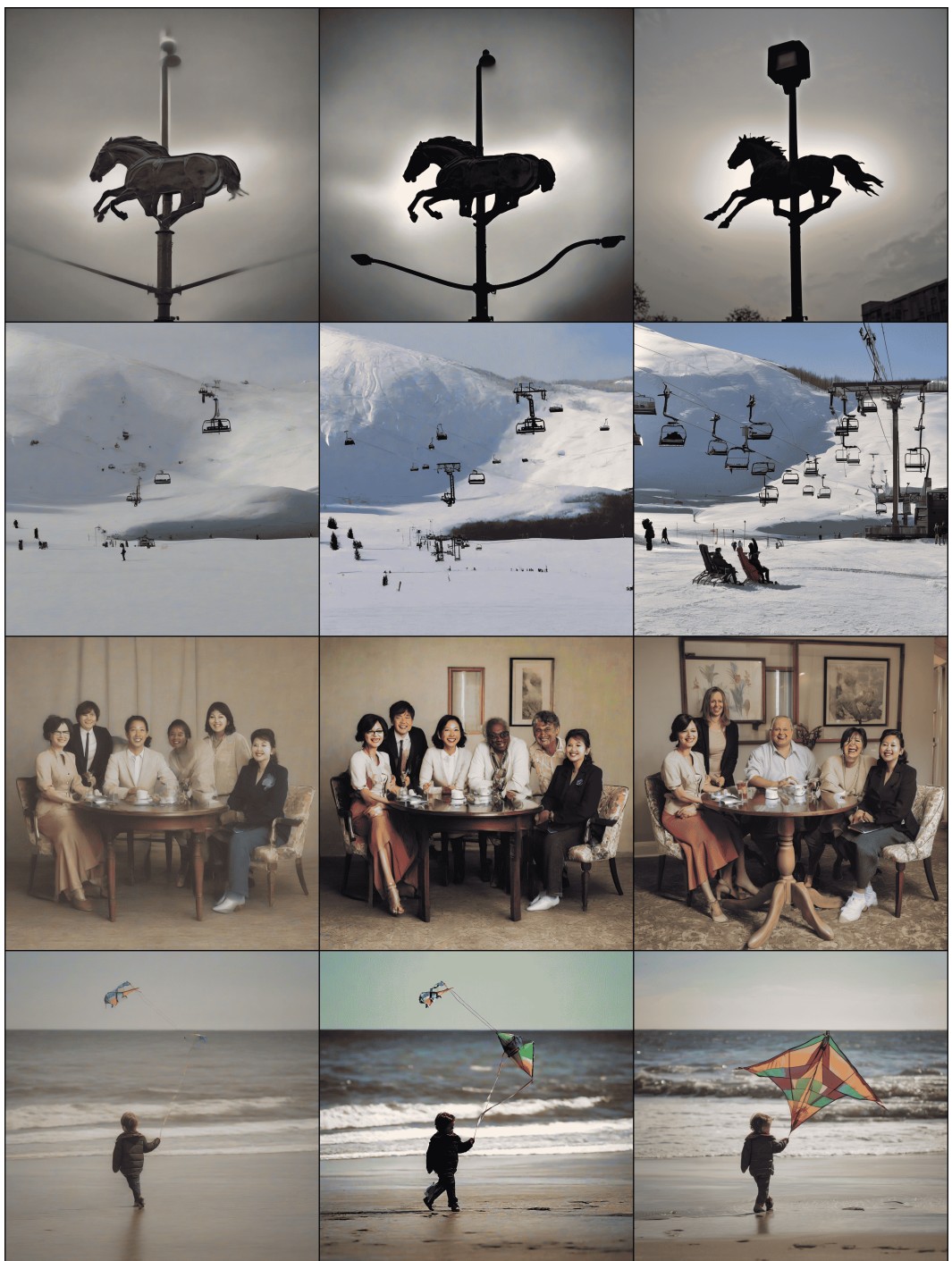

Figure 29: Comparison between 10 steps DDIM (left), DDIM-MCDO (middle) and 1000 steps DDIM (right) with SDXL. Prompts used (from top to bottom): *"A street light that shows, horse crossing on it"*, *"There are people on the snow bank and snow lift chairs are above."*, *"Four people gather around a table and smile for the picture."*, *"A small child with a kite walking on a beach."*

