# OpenReview forum: "Manifold Constraint Reduces Exposure Bias in Accelerated Diffusion Sampling"
_ICLR.cc/2025/Conference — ICLR 2025 Poster_

### Official Review · Reviewer_6dod · 2024-10-25

**Soundness:** 4
**Presentation:** 3
**Contribution:** 3
**Rating:** 6
**Confidence:** 4

**Summary:**

The paper studies the exposure bias of accelerated diffusion model sampling from geometry perspective. The authors extend the previous manifold constraint theory with more detailed description of pixel variance, claiming that both exposure bias and truncation error account for the performance degradation. To this end, the authors propose to pre-calculate the reference pixel variance, which serves as a correction during inference. Such method achieves a training-free and easy-to-implement solution to performance degradation. Comprehensive experiments confirm the efficacy of the proposed algorithm.

**Strengths:**

- The paper is well structured and easy to follow. Most of the derivation is clear.

- The paper extends the previous manifold constraint strategy with deeper study of the pixel variance and the consequent exposure bias, providing a novel perspective of geometric technique in diffusion models.

- The proposed algorithm is overall both efficient and effective, inference time cost barely increases.

- Quantitative experiments are convincing and extensive.

**Weaknesses:**

- There seem some theoretical flaws in the draft, harming the soundness:
  1. Eq. (15) is wrong, when $x$ and $y$ are orthogonal, one can only deduce that $|x+y|^2=|x|^2+|y|^2$. Besides, what is the definition of $x_0$ and $\epsilon_t$? $\epsilon_t$ and $x_0$ may not be orthogonal if no further assumption are made.
  2. I cannot understand the relation between Eq. (18) and analytical form of $Var(x^{(t)}_0)$ and $Var(x_t)$, $Var(\cdot)$ is supposed to be the pixel variance as claimed in Eq. (12) which is a scalar.
  3. Eq. (16) and Eq. (19) are similar, but the authors conclude differently. If minimizing right hand side in Eq. (19) could lead to the distance reduction, then so could minimizing right hand side in Eq. (16) be. They are all the **lower bounds** of distance of samples to manifold.
  4. If the authors insist that nonzero $|x_0|$ affects the derivation, then (1) since $\hat{x_t}\in\mathcal{M}_t$, one can simply choose $x_0=0$ (which is reasonable since the authors have already assumed zero mean in L346), or (2) move the term with $|x_0|$ outside the absolute value using $|a+b-c|\geqslant|b-c|-|a|$, which is similar to the form of Eq. (19). The authors should make further clarification.
  5. Why assume zero mean in L346? $\hat{x}_0^{(t)}$ is the denoise observation at timestep $t$, somewhat a data sample with no noise. Then why is the case? The authors could calculate the mean to confirm the reasonability.

- Figs. 4 and 6 only employ 64 samples, which seems inconvincing.

- Visualization in Fig. 7 fails to be photorealistic with obvious color shift artifact, which is also the case in Fig. 12.

**Questions:**

- It is intuitive that applying MCDO with larger NFEs or better sampler will achieve weaker improvements. I am curious about the comparison on better sampler like Heun or DPM-Solver. There is also no discussion about applicability on high-order samplers.

- MCDO is proposed for manifold constraint to relieve exposure bias, how will the efficacy vary if different CFG scales are set? Larger CFG scale may also lead to severe exposure bias.

---

> ### Author Response · Authors · 2024-11-24
> **Response to Reviewer 6dod (part 1)**
>
> Thank you for your insightful comments and questions. Please find our response below:
>
> >### **W1: Theoretical details in the paper**
>
> >#### **W1.1 Definition of $\\boldsymbol{x}\_0$ and $\\boldsymbol{\\epsilon}\_t$**
>
> **A1** Thanks for pointing this out. $x_0$ is the clean image sampled from data distribution, while $\epsilon_t$ is the noise sampled from normal distribution. Since $e_t$ is independent of $x_0$, with a high enough latent dimension $n$ [1], one has: $\mathbb{E}[\| \boldsymbol{x}_0 + \boldsymbol{\epsilon}_t\|^2 ] = \mathbb{E}[\| \boldsymbol{x}_0\|^2] + \mathbb{E}[\|\boldsymbol{\epsilon}_t\|^2]$. This typo is corrected in the revised manuscript.
>
> [1] Sven-Ake Wegner. Lecture notes on high-dimensional data, 2024.
>
> >#### **W1.2: Relation between the Analytical Forms of $\\boldsymbol{\\hat{x}}_t$ and $\\boldsymbol{\\hat{x}}_0^{(t)}$ and Their Pixel Variance**
>
> **A2** Thank you for pointing this out. We would like to provide further clarification. The analytical forms of $\\hat{x}\_0^{(t)}$ and $\\hat{x}\_t$ given in Equation 18 are presented to demonstrate the following:
>
> 1. During the generation process, the noisy data prediction $\\hat{\\boldsymbol{x}}\_t$ contains a component of scaled-down prediction error, denoted as $n\_t \\cdot \\boldsymbol{e}\_\\theta^{(t)}$, where $n\_t$ is the scale factor. Conversely, the denoising observation $\\hat{\\boldsymbol{x}}\_0^{(t)}$ incorporates a scaled-up prediction error, expressed as $d\_t \\cdot \\boldsymbol{e}\_\\theta^{(t)}$, where $d\_t$ is the scale factor.
> 2. The large magnitude of $d\_t$ suggests that **$\\hat{x}\_0^{(t)}$ is predominantly influenced by the amplified noise prediction error term $d\_t \\cdot e\_\\theta^{(t)}$ at most timesteps.** Thus, it is possible to correct the noise estimation error $\\boldsymbol{e}\_\\theta^{(t)}$ by incorporating information from the denoising observation $\\hat{x}_0^{(t)}$.
>
> In other words, **while the noisy data prediction $\\hat{x}\_t$ primarily reflects information about its ground truth noisy data $\\boldsymbol{x}\_t$, the denoising observation $\\hat{x}\_0^{(t)}$ conveys information about the prediction noise $e_\\theta^{(t)}$. This discrepancy leads to the following results**:
>
> 1. **Correcting the pixel variance of $\\hat{x}\_t$ using a scale factor will eventually scales the entire term**, $\\boldsymbol{x}\_t + n\_t \\boldsymbol{e}\_\\theta^{(t)}$. Consequently, this **leads to distortion of the $\\boldsymbol{x}\_t$ component in $\\hat{x}\_t$.**
> 2. When correcting the pixel variance of $\\hat{x}\_0^{(t)}$ using the proposed method, one is correcting $\\boldsymbol{x}\_0 + d\_t \\boldsymbol{e}\_\\theta^{(t)}$. While errors in $\\boldsymbol{e}\_\\theta^{(t)}$ also influence the $\\boldsymbol{x}_0$ term, **it is important to note that by recalculating the noise estimation with the corrected denoising observation (Equation 21)** : $\\tilde{\\boldsymbol{\\epsilon}}^{(t)}\_\\theta = \\frac{x\_{t} - \\sqrt{\\bar{\\alpha}\_{t}} \\tilde{x}\_0^{(t)}}{\\sqrt{1 - \\bar{\\alpha}\_{t}}}$ **the prediction error is reduced while the original $\\boldsymbol{x}\_t$ is preserved.**
>
> We hope this clarification addresses your concerns. Please feel free to reach out if further explanation is needed.
>
>
> >#### **W1.3 Comparison between Equation 16 and Equation 19 regarding their role in reducing data-manifold distance.**
>
> **A3:** **We respectfully disagree with your statement that "the authors conclude differently."** It appears there may be some misunderstanding, and we would like to clarify this point further.
>
> 1. **As clearly stated in Section 4.2, Equation (16) offers a manifold-based perspective to understand the effectiveness of the previous method, TS-DDIM [1]**, which relies on the assumption of "using pixel variance to estimate the sample distribution variance."
> 2. We agree with your observation that "minimizing the right-hand side in Eq. (16) could also lead to the reduction of the distance." As we view this approach as an effective explanation of the timestep-shifting strategy in TS-DDIM [1], from the manifold perspective. **We acknowledge the simplicity and effectiveness of their proposed method, which inspired us to explore the concept of pixel variance further, and to leverage Equation 19 to develop the proposed method.**
>
> [1] Li et al. Alleviating exposure bias in diffusion models through sampling with shifted time steps. ICLR 2024.

---

> ### Author Response · Authors · 2024-11-24
> **Response to Reviewer 6dod (part 2)**
>
> >#### **W1.4 Different derivation with/without $\\| \\boldsymbol{x}_0 \\|$**
>
> **A4** Thank you for your thorough and thoughtful review. We agree that the method you propose is effect, and we would like to offer further clarification to address some potential misunderstandings regarding our method:
>
> 1. **We did not claim that it is essential to consider $\\boldsymbol{x}_0$.**
> 2. **Our proposal introduces an alternative approach that incorporates $\\boldsymbol{x}_0$ into the process.**
> 3. **We view both approaches (with and without $\\| \\boldsymbol{x}_0 \\|$) as practical** (see Sections 4.2 and 4.3). The approach without $\\boldsymbol{x}_0$ can be interpreted as the objective of TS-DDIM [1].
> 4. However, on large resolution dataset (e.g., $1024\\times 1024$ for SDXL), we observe that our proposed method, which incorporates $\\boldsymbol{x}_0$, leads to improved results.
>
> Hope this clarificaiton addresses your concerns. Please do not hesitate to let us know if further explanation is required.
>
> [1] Li et al. Alleviating exposure bias in diffusion models through sampling with shifted time steps. ICLR 2024.
>
> >#### **W1.5 Assumption on pixel mean of denoising observation $\\boldsymbol{\\hat{x}}\_0$**
> **A5** Thanks for point this out. We would like to explain further.
> 1. Given that the diffusion model is trained to mininmize $\\| \\boldsymbol{\\epsilon}\_t - \\boldsymbol{\\epsilon}\_\theta^{(t)} \\|, \\boldsymbol{\\epsilon}\_t \\sim \\mathcal{N}(\\mathbf{0},\\boldsymbol{I})$. With a well-trained model, one has: $\\mathbb{E}[\\boldsymbol{\\epsilon}\_\\theta^{(t)}] \\rightarrow \\mathbb{E}[\\boldsymbol{\\epsilon}\_t] = 0$. For the first timestep in denoising process ($t=T$), $\\boldsymbol{\\hat{x}}\_t \\sim  \\mathcal{N}(\\mathbf{0},\\boldsymbol{I})$. The expectation of pixel mean of $\\hat{\\boldsymbol{x}}\_0^{(t)}$ is $0$ as it is a linear composition of $\\boldsymbol{\\hat{x}}\_t$ and $\\boldsymbol{\\epsilon}\_\\theta^{(t)}$: $\\hat{\\boldsymbol{x}}\_0^{(t)} = \\frac{\\hat{\\boldsymbol{x}}\_t - \\sqrt{1 - \\bar{\\alpha}\_t} \, \\boldsymbol{\\epsilon}^{(t)}\_\\theta\\left(\\boldsymbol{x}\_t\\right)}{\\sqrt{\\bar{\\alpha}\_t}}$.
> 2. Considering that the noisy data prediction for $t-1$ timestep can be expressed as $\\hat{\\boldsymbol{x}}\_{t-1} = \\sqrt{\\bar{\\alpha}\_{t-1}} \\hat{\\boldsymbol{x}}\_0^{(t)} + \\sqrt{1 - \\bar{\\alpha}\_{t-1} - \\sigma\_t^2} \, \\boldsymbol{\\epsilon}\_\\theta^{(t)}\\left(\\mathbf{x}\_t\\right) + \\sigma\_t \\boldsymbol{\\epsilon}'\_t,\\quad \\boldsymbol{\\epsilon}'\_t \\sim \\mathcal{N}\\left(\\mathbf{0}, \\mathbf{I}\\right)$, we have: $\\mathbb{E}[\\boldsymbol{\\hat{x}}\_{t-1}] = 0$. Then according to the definition of denoising observation, the expectation of pixel mean for denoising observation at timestep $t-1$: $\\mathbb{E}[\\frac{1}{n}\\Sigma\_{i=0}^{n}\\hat{\\boldsymbol{x}}\_0^{(t-1)}] = 0$
>
> >### **W3: Visualization with photorealistic and obvious color shift**
>
> **A6** We appreciate your observation and agree that further refinement of the correction strategy could potentially improve the sample quality. We acknowledge that some aspects related to the visualizatino presented in Figures 7 and 12 require clarification and would like to address your concerns in detail:
>
> 1. **Higher saturation alone does not equate to higher image quality, and it should not be interpreted as a flaw unique to our method.** While some color shifts (compared to the baseline which sometimes generates blurred and un-recognizable images) are noticeable, the higher saturation arises from our method's effectiveness in reducing models' prediciton error, therefore **producing image with finer quality (sharper textures and richer structural details), as can be observed in all visualizations presented in paper.**
> 2. As the sampling steps decreases, blurriness are observed in many baseline samples. As a consequence, it is reasonable for solutions which improve image quality make the generated results look "relatively saturated". Which is observed in many previous published works [2][3][4].
> 4. **In Appendix A.13 of the revised manuscript, we include the visualization results generated using 1,000 steps** which we believe will serve as a fair reference.
>
> [2] Chen et al. On the Trajectory Regularity of ODE-based Diffusion Sampling, ICML 2024.
> [3] Si et al. FreeU: Free Lunch in Diffusion U-Net. CVPR 2024 Oral.
> [4] Lin et al. Common diffusion noise schedules and sample steps are flawed. WACV 2024.

---

> ### Author Response · Authors · 2024-11-24
> **Response to Reviewer 6dod (part 3)**
>
> We sincerely thank you for your insightful reviews and valuable questions regarding the applicability of MCDO to high-order samplers and its effectiveness under large CFG scales. Below, we present detailed experimental results to address these points.
>
> >### **Q1**: Applicability to High-Order Sampler
>
> **A7** To evaluate the performance of MCDO with a high-order sampler, we conducted experiments using **DPM-Solver++** on **SDXL** with 5, 6, 10 steps. 60 samples were used for statistics estimation. The results, based on 10k generated images, are summarized below:
> | **Timesteps** | **Method**              | **FID**   | CLIP Score | **$t\_{thre}$** |
> | ----------------------- | ----------------------- | --------- | ---------- | -------------- |
> | 5                       | DPM-Solver++(2M)        | 19.16     | 31.52      | --             |
> | 5                       | DPM-Solver++(2M) + MCDO | **18.51** | **31.62**  | 100            |
> | 6                       | DPM-Solver++(2M)        | 17.05     | 31.65      | --             |
> | 6                       | DPM-Solver++(2M) + MCDO | **16.39** | **31.72**  | 100            |
> | 8                       | DPM-Solver++(2M)        | 15.57     | 31.74      | --             |
> | 8                       | DPM-Solver++(2M) + MCDO | **15.32** | **31.79**  | 100            |
> | 10                      | DPM-Solver++(2M)        | 15.10     | 31.74      | --             |
> | 10                      | DPM-Solver++(2M) + MCDO | **14.88** | **31.85**  | 100            |
>
> The results demonstrate that **the proposed method consistently enhances performance when combined with DPM-Solver++**. While the improvement of MCDO is more pronounced with DDIM compared to DPM-Solver++, this is consistent with the expectation that high-order solvers, such as DPM-Solver++, inherently reduce prediction errors. The results are also presented in Appendix A.10 of the revised manuscript.
>
>
>
> >### **Q2: Efficacy of MCDO with combined with larger CFG Scales**
>
> **A8** To assess the efficacy of MCDO under varying CFG scales, we performed additional experiments using DDIM sampler on SDXL with larger CFG scales (8, 10 ,12) using 10 sampling steps.  The number of samples used for statistics estimation is the same as that in the paper. The results are summarized below:
>
> | **CFG Scale** | **Method**      | **FID** | **CLIP Score** | **$t\_{thre}$** |
> | --------- | --------------- | ------- | ------- | -------------- |
> | 8    | DDIM            | 18.51   | 31.81  |--             |
> | 8     | DDIM + MCDO     | **17.38**  | **32.03**  | 100            |
> | 10 | DDIM | 18.99     | 31.82 | -- |
> | 10 | DDIM + MCDO | **17.49** | **32.13** | 100 |
> | 12 | DDIM | 20.62 | 31.70 | -- |
> | 12 | DDIM + MCDO | **17.80** | **32.18** | 100 |
>
> The findings demonstrate that **MCDO significantly improves performance under large CFG scales, achieving a notable reduction in FID.** This suggests that **MCDO effectively alleviates the increased exposure bias introduced by higher CFG scales.** The results are also presented in Appendix A.12 of the revised manuscript.

---

> ### Comment · Reviewer_6dod · 2024-11-25
>
> I highly appreciate the efforts of the authors in the rebuttal. And below is my feedback.
>
> - Figs. 4 and 6 only employ 64 samples, which seems inconvincing. The authors seemed to miss this one.
>
> - The clarification in W1.4 is not that convincing. I hope the authors could provide some experimental or theoretical results beyond the current response.
>
> - Response in W1.5 is still wrong.
>   1. First, there is a gap in the original Diffusion Model paper that, $\mathbf{x}_T$ in the forward diffusing process only **approximates** a Gaussian distribution with zero mean, *i.e.*, $\mathbf{x}_T\approx\mathcal{N}(\alpha_T\mathbf{x}_0,\sigma_T^2\mathbf{I})$. Both theoretically and empirically, $\alpha_T$ is extremely small but still nonzero. Therefore, the conclusion of zero mean by induction at each $t<T$ is wrong in your rebuttal.
>   2. Second, if the underlying data follows a distribution with extremely large mean, from my perspective, the zero-mean assumption is unreasonable and may lead to strong error in your pipeline. I hope the authors could conduct experiments on toy data to convince me, *e.g.*, following setting of Reviewer Tp1G, the authors could apply your method on a Dirac Delta distribution $\delta(\mathbf{x}_0)$, in which $\mathbf{x}_0$ is extremely far away from the origin. Such distribution has large mean and zero variance, and its score function has analytical form.

---

> > ### Author Response · Authors · 2024-11-26
> >
> > Dear Reviewer 6dod,
> >
> > We sincerely appreciate your valuable suggestions and the contributions you’ve made to enhancing our work.
> >
> > We will conduct the additional experiments you suggested, addressing your concerns in detail. Once we have the results, we will provide a more comprehensive response.
> >
> > Best regards,
> >
> > The authors

---

> > > ### Author Response · Authors · 2024-11-28
> > > **Response to further comments (part 3)**
> > >
> > > Thank you for the insightful comments. We believe that the differences in our understanding may stem from the gap between the theoretical foundations of diffusion models and their practical applications. We would like to offer further clarification regarding the concerns raised.
> > >
> > > > ### In the forward diffusing process, $x\_T$ only **approximates** a Gaussian distribution with zero mean, *i.e.*, $x\_T \\approx \\mathcal{N}(\\alpha\_T x\_0, \\sigma\_T^2 I)$. Both theoretically and empirically, $\\alpha\_T$ is extremely small but still nonzero.
> > >
> > > **A5**
> > > 1. We agree with your point regarding the theoretical form of the mean of $x\_T$ during training.
> > > 2. At the same time, we would like to mention **the gap between the theoretical and the practical aspects of diffusion models**. In practice, during the sampling process of diffusion models, the initial noise is typically sampled from $\\mathcal{N}(\\mathbf{0}, \\mathbf{I})$.
> > > 3. Therefore, the **assumption $\\boldsymbol{x}_T \\sim \\mathcal{N}(\\mathbf{0},\\mathbf{I})$ used in our explanation for W1.5 reflects the accurate distribution of $x\_T$ in the practical sampling process.**
> > >
> > > Based on these points, we believe the assumptions we made are within an acceptable margin of error.
> > >
> > > > ### If the underlying data follows a distribution with an extremely large mean, the zero-mean assumption seems unreasonable and may introduce significant error in your pipeline.
> > >
> > > **A6**
> > > 1. We agree with your opinion that when the data follows a distribution with an extremely large mean, the zero-mean assumption may indeed lead to errors in the pixel variance-L2 norm approximation.
> > > 2. However, we would also like to highlight that **this is not a very common scenario in many widely used diffusion models. (*e.g.* LDM[1], SD3[2], DiT[3] and SDXL[4])**. For example, in LDM[1], the distribution of $x\_0$ in the latent space is regularized to approach a standard normal distribution:
> > >
> > > > "To avoid arbitrarily high-variance latent spaces, we experiment with two different kinds of regularizations. The first variant, KL-reg., imposes a slight KL penalty towards a standard normal on the learned latent, similar to a VAE." [1]
> > >
> > > This regularization encourages the latent variables to follow a distribution with **a mean close to zero, preventing the scenario of extremely large means**. Therefore, in well-trained diffusion models, it is not typical for $x_0$ to have an extremely large mean, and thus the zero-mean assumption holds reasonably well.
> > >
> > > We hope this clarification addresses your concerns. To summarize, we would like to:
> > >
> > > 1. acknowledge your insightful point regarding the **limitations of our assumptions in extreme scenarios**, which we agree are an important consideration.
> > > 2. highlight that our method is **empirically supported in typical use cases**, as demonstrated by a range of performance metrics (FID, CLIP Score, sFID, IS, FID-DINO, KD) across various datasets, models (LDM [1], SD3 [3], SDXL [4]), and conditions (e.g., different CFG scales).
> > > 3. emphasize that all the presented **assumptions are supported by statistical evidence**, which falls within an acceptable margin of error.
> > > 4. Finally, given the simplicity of our approach—requiring no training and minimal hyperparameter tuning—we believe that these approximations are reasonable and acceptable for some practical applications.
> > >
> > > We are happy to provide further clarification in response to any additional concerns.
> > >
> > > [1] Rombach et al. High-Resolution Image Synthesis with Latent Diffusion Models. CVPR 2022.
> > > [2] Esser et al. Scaling Rectified Flow Transformers for High-Resolution Image Synthesis. ICML 2024.
> > > [3] Peebles and Xie. Scalable Diffusion Models with Transformers. ICCV 2022 Oral.
> > > [4] Podell et al.SDXL: Improving Latent Diffusion Models for High-Resolution Image Synthesis. ICLR 2024 Spotlight.

---

> > > > ### Comment · Reviewer_6dod · 2024-11-30
> > > >
> > > > Thanks for the comprehensive explanations and clarifications. I highly appreciate the great efforts and dedication of the authors. I think all my concerns except for W1.4 are well addressed.
> > > >
> > > > I am willing to raise my score if W1.4 is also clearly clarified and the authors update both the analysis and the reasonability of the zero mean assumption in the revised version.

---

> > > > > ### Author Response · Authors · 2024-12-02
> > > > >
> > > > > We sincerely appreciate your engagement with our work and your insightful comment regarding the differences between the methods developed from Eq. (16) and Eq. (19). In response, we provide additional quantitative results to support our previous explanation.
> > > > >
> > > > > For clarity, we refer to the Noisy Data Variance Adjustment (NDVA) method as being derived from Eq. (16) **with $\boldsymbol{x}_0$ excluded from the lower bound expression**, and the pixel variance of $\hat{\boldsymbol{x}}_t$ is directly adjusted across timesteps.
> > > > >
> > > > > >### Comparison with TS-DDIM
> > > > >
> > > > > Below, we compare the performance of TS-DDIM[1], NDVA, and MCDO on the CelebA-HQ dataset. Each FID evaluation involves generating 50k images.
> > > > >
> > > > > | Method                        | 20 Steps         | 10 Steps         | 5 Steps         |
> > > > > | ----------------------------- | ---------------- | ---------------- | --------------- |
> > > > > | DDIM                          | 10.59            | 21.08            | 54.99           |
> > > > > | TS-DDIM                       | **8.29**         | 15.71 | 50.69           |
> > > > > | DDIM + NDVA                   | 9.21 | 16.45            | 47.52 |
> > > > > | DDIM + MCDO (ours)            | 9.49             | **15.63**        | **30.76**       |
> > > > >
> > > > > From the table, it can be observed that the NDVA method:
> > > > >
> > > > > 1. **Outperforms the baseline method**, supporting your perspective and our earlier response:
> > > > > >We view both approaches (with and without $\\|\boldsymbol{x}_0\\|$) as practical.
> > > > > 2. **Exhibits performance comparable to TS-DDIM**. It also outperforms MCDO slightly at 20 timesteps.
> > > > > 3. **Performs notably worse than MCDO as the number of timesteps decreases**, with a clear gap evident at 10 and 5 steps.
> > > > >
> > > > > >### Results on large resolution dataset
> > > > >
> > > > > We evaluate NDVA on the MS-COCO dataset using SDXL and DPM-Solver at an image resolution of $1024 \\times 1024$. In this evaluation, we generate 10k images, employing the same settings as in our previous experiments on SDXL (e.g., the number of samples for statistical estimation and $t_{\text{thre}}$ for MCDO).
> > > > >
> > > > > | Method                   | 10 Steps  | 8 Steps   | 6 Steps   | 5 Steps   |
> > > > > | ------------------------ | --------- | --------- | --------- | --------- |
> > > > > | DDIM                     | 15.10     | 15.57     | 17.05     | 19.16     |
> > > > > | DDIM + NDVA              | 15.76     | 15.76     | 17.13     | 19.02     |
> > > > > | DDIM + MCDO (ours)       | **14.88** | **15.32** | **16.39** | **18.51** |
> > > > >
> > > > > The results demonstrate that, on larger resolution datasets, MCDO consistently outperforms NDVA across various low NFE experiments. It also supports our earlier statement:
> > > > > > on large resolution dataset (e.g., for SDXL), we observe that our proposed method, which incorporates $\\|\\boldsymbol{x}_0\\|$, leads to improved results.
> > > > >
> > > > > We hope these experimental results clarify the distinction between the methods (with and without $\\|\\boldsymbol{x}_0\\|$) and adequately address your concerns. We sincerely appreciate your interest in our work and your engagement in the discussion. Thank you again for your time and effort in providing such insightful feedback.
> > > > >
> > > > >
> > > > > [1] Li et al. Alleviating exposure bias in diffusion models through sampling with shifted time steps. ICLR 2024.

---

> > > > > > ### Comment · Reviewer_6dod · 2024-12-02
> > > > > >
> > > > > > Thanks for the further clarifications and extensive quantitative results. I at present believe the whole story is both theoretically and empirically solid, and the work will encourage future works on manifold constraints.
> > > > > >
> > > > > > Therefore, I raise my score accordingly. I hope that the authors could revise the draft according to the rebuttal, especially the further clarification of the pipeline and refinement of the theory part.

---

> > > > > > > ### Author Response · Authors · 2024-12-02
> > > > > > >
> > > > > > > Dear reviewer 6dod,
> > > > > > >
> > > > > > > Thank you for your engagement and for raising your score. We sincerely appreciate your recognition of this work. We also want to thank you for being active and constructive during the whole discussion. In the revised version, we will incorporate your suggestions, including further clarification of the pipeline and additional empirical results supporting the theoretical analysis.
> > > > > > >
> > > > > > > Best regards,
> > > > > > >
> > > > > > > The Authors

---

> > ### Author Response · Authors · 2024-11-28
> > **Response to further comments (part 1)**
> >
> > Dear reviewer,
> >
> > Thank you once again for your thoughtful engagement in the discussion, as well as your valuable feedback. We sincerely appreciate your acknowledgment of our effort. Below, we address each of your concerns in detail.
> >
> > > ### Zero mean assumption
> >
> > **A1**
> > First of all, we agree that the variables mentioned—$\\hat{\\boldsymbol{x}}_t$, $\\hat{\\boldsymbol{x}}\_0^{(t)}$, $\\boldsymbol{\\epsilon}\_\theta^{(t)}$, and $\\boldsymbol{x}\_T$—**do not strictly exhibit zero mean in all cases**. However, in diffusion models, these variables possess certain characteristics that **keep their pixel means within an acceptable range around zero**. As a result, the assumption in our work—that their pixel mean is approximately zero—introduces only negligible errors. Given this, we believe that this assumption is viable for some applications.
> >
> > > ### Figs. 4 and 6
> >
> > **A2**
> > Thank you for your suggestion regarding the use of additional data to enhance the generalizability of the observed variance-norm relation. In response, we have replaced Figure 4 with a **new curve based on statistics from 6,400 samples, which demonstrates a similar relationship to that observed with 64 samples**. Following thorough discussions among all authors, we have decided to remove Figure 6, as it is not critical for illustrating the core variance-norm relationship.
> >
> > > ### Error analysis for assuming zero pixel mean for $\\hat{\\boldsymbol{x}}\_0^{(t)}$ and  $\\hat{\\boldsymbol{x}}\_t$
> >
> > **A3**
> > Again, we agree that more refined assumptions could potentially lead to better results. However, the assumptions we make simplify the relationship between exposure bias in diffusion models and pixel variance, allowing us to propose a straightforward yet effective method.
> >
> > In response to your concern regarding the pixel mean assumption, we provide experimental results below to assess the error introduced by this assumption. Considering that the objective of assuming the zero mean for $\\hat{\\boldsymbol{x}}\_0^{(t)}$ and $\\hat{\\boldsymbol{x}}\_t$ is to relate their pixel variance with their L2 norm, we report the relative error of this approximation: $e=\\frac{\\sqrt{n\\text{Var}(\\boldsymbol{x})} - \\|\\boldsymbol{x}\\|}{ \\|\\boldsymbol{x}\\|} \times 100\\%$ in the tables below:
> >
> > | Denoising Timestep                                           | 951    | 901    | 851    | 801    | 751    | 701    | 651    | 601    | 551    | 501    |
> > | ------------------------------------------------------------ | ------ | ------ | ------ | ------ | ------ | ------ | ------ | ------ | ------ | ------ |
> > | Relative error for $\\hat{\\boldsymbol{x}}\_t$ (%) | 0.0019 | 0.0017 | 0.0010 | 0.0006 | 0.0038 | 0.0096 | 0.0196 | 0.0352 | 0.0585 | 0.0909 |
> > | Relative error for $\\hat{\\boldsymbol{x}}\_0^{(t)}$ (%) | 4.0269 | 3.0110 | 1.8091 | 1.1320 | 0.8772 | 0.8084 | 0.8015 | 0.8252 | 0.8448 | 0.8669 |
> >
> > | Denoising Timestep                                           | 451    | 401    | 351    | 301    | 251    | 201    | 151    | 101    | 51     | 1      |
> > | ------------------------------------------------------------ | ------ | ------ | ------ | ------ | ------ | ------ | ------ | ------ | ------ | ------ |
> > | Relative error for $\\hat{\\boldsymbol{x}}\_t$ (%) | 0.1337 | 0.1873 | 0.2506 | 0.3217 | 0.3973 | 0.4738 | 0.5472 | 0.6141 | 0.6725 | 0.7250 |
> > | Relative error for $\\hat{\\boldsymbol{x}}\_0^{(t)}$ (%) | 0.8783 | 0.8785 | 0.8670 | 0.8498 | 0.8285 | 0.8049 | 0.7807 | 0.7567 | 0.7321 | 0.7086 |
> >
> > The results demonstrate that throughout the entire generation process:
> >
> > - The **approximation error remains consistently small**, with its value being **below 1%** for most timesteps.
> > - The maximum relative error  **does not exceed 4.5%**.
> >
> > This shows that **assuming the zero mean of $\\hat{\\boldsymbol{x}}\_0^{(t)}$** and using its pixel variance as an approximation for its L2 norm **introduces minimal error**, which is acceptable for practical applications.
> >
> > The evolution of this relative error is also provided in Appendix A.14 of the revised manuscript.

---

> > ### Author Response · Authors · 2024-11-28
> > **Response to further comments (part 2)**
> >
> > > ### Pixel mean of $\\hat{\\boldsymbol{x}}\_t$, $\\hat{\\boldsymbol{x}}\_0^{(t)}$ and  $\\boldsymbol{\\epsilon}\_\\theta^{(t)}$  across timesteps
> >
> > **A4**
> > To address your concerns, we collect the average absolute pixel mean for the noisy data prediction: $\\mathbb{E}[|\\frac{1}{n}\\sum_{i=0}^n \\hat{\\boldsymbol{x}}\_t |]$, denoising observation: $\\mathbb{E}[|\\frac{1}{n}\\sum\_{i=0}^n \\hat{\\boldsymbol{x}}\_0^{(t)} |]$, and noise estimation: $\\mathbb{E}[|\\frac{1}{n}\\sum\_{i=0}^n \\boldsymbol{\\epsilon}\_\\theta^{(t)} |]$ across the generation process. The statistics were collected from 6,400 samples generated using a 100-step LDM on CelebA-HQ. It can be observed that the average **pixel mean for $\\hat{\\boldsymbol{x}}\_t$, $\\hat{\\boldsymbol{x}}\_0^{(t)}$ and  $\\boldsymbol{\\epsilon}\_\\theta^{(t)}$   exhibits only minor deviations from 0 across timesteps**.
> >
> >
> > | **Timestep**                                                 | 951    | 901    | 851    | 801    | 751    | 701    | 651    | 601    | 551    | 501    |
> > | ------------------------------------------------------------ | ------ | ------ | ------ | ------ | ------ | ------ | ------ | ------ | ------ | ------ |
> > | $\\mathbb{E}[\|\\frac{1}{n}\\Sigma\\hat{\\boldsymbol{x}}\_t\|]$ | 0.0075 | 0.0079 | 0.0087 | 0.0101 | 0.0122 | 0.0154 | 0.0196 | 0.0248 | 0.0311 | 0.0382 |
> > | $\\mathbb{E}[\|\\frac{1}{n}\\Sigma\\hat{\\boldsymbol{x}}\_0^{(t)}\|]$ | 0.1119 | 0.1093 | 0.1095 | 0.1099 | 0.1101 | 0.1109 | 0.1114 | 0.1118 | 0.1116 | 0.1119 |
> > | $\\mathbb{E}[\|\\frac{1}{n}\\Sigma\\boldsymbol{\\epsilon}\_\\theta^{(t)}\|]$ | 0.0068 | 0.0070 | 0.0070 | 0.0070 | 0.0069 | 0.0067 | 0.0064 | 0.0060 | 0.0056 | 0.0051 |
> >
> >
> > | **Timestep**                                                 | 451    | 401    | 351    | 301    | 251    | 201    | 151    | 101    | 51     | 1      |
> > | ------------------------------------------------------------ | ------ | ------ | ------ | ------ | ------ | ------ | ------ | ------ | ------ | ------ |
> > | $\\mathbb{E}[\|\\frac{1}{n}\\Sigma\\hat{\\boldsymbol{x}}\_t\|]$ | 0.0460 | 0.0543 | 0.0629 | 0.0716 | 0.0800 | 0.0880 | 0.0954 | 0.1020 | 0.1077 | 0.1126 |
> > | $\\mathbb{E}[\|\\frac{1}{n}\\Sigma\hat{\\boldsymbol{x}}\_0^{(t)}\|]$ | 0.1121 | 0.1123 | 0.1124 | 0.1125 | 0.1125 | 0.1126 | 0.1127 | 0.1128 | 0.1128 | 0.1128 |
> > | $\\mathbb{E}[\|\\frac{1}{n}\\Sigma\\boldsymbol{\\epsilon}\_\\theta^{(t)}\|]$ | 0.0045 | 0.0040 | 0.0036 | 0.0032 | 0.0028 | 0.0025 | 0.0022 | 0.0021 | 0.0020 | 0.0011 |
> >
> > The entire evolution of this relative error is provided in Appendix A.14 of the revised manuscript.

---

> > ### Author Response · Authors · 2024-11-28
> >
> > > Further clarification for W1.4
> >
> >  Updating soon.

---

### Official Review · Reviewer_Tp1G · 2024-10-27

**Soundness:** 3
**Presentation:** 2
**Contribution:** 2
**Rating:** 6
**Confidence:** 4

**Summary:**

From the manifold hypothesis, the paper proposes a method to reduce Exposure Bias by adjusting the variance of $x_t$ at each step to match the variance of $q_t$ made by the forward diffusion process. The paper demonstrates that this approach yields FID gains across various model and datasets.

**Strengths:**

They validated the method in various datasets and methods.

It is good to see that the method proposed from this work can be combined with the method from previous literature, i.e., DDIM-MCDO^\dagger.

**Weaknesses:**

**1. strong assumption**

Assuming that the manifold can be understood solely by considering variance is too strong an assumption. It might be better to tone it down to suggest that this approximation is sufficient for performance improvement.

**Questions:**

**Q1**: Using the statistics of the data directly in generation—could this be seen as an FID "hack"? For example, I’m curious how the FID would change if the mean and variance of the generated data w/o MCDO were adjusted to match \( q_0 \).

**Q2**: Does this method can improve other ODE solvers like DPM-solver++ [1] or PNDM [2]? I know that DDIM performs poorly when the NFE is below 50. I also want to see the results where NFE is around 50.

[1]: DPM-Solver++: Fast Solver for Guided Sampling of Diffusion Probabilistic Models ([Arxiv](https://arxiv.org/abs/2211.01095))
[2]: Pseudo Numerical Methods for Diffusion Models on Manifolds ([ICLR22](https://arxiv.org/abs/2202.09778))

**Q3**: (Minor) When comparing performance, I recommend plotting FID on the y-axis and NFE on the x-axis for clarity.

---

> ### Author Response · Authors · 2024-11-24
> **Response to Reviewer Tp1G (part 1)**
>
> We greatly appreciate your recognition of our method’s compatibility with previous work. Below, we provide answers to your concerns.
>
> >### **W1: Assumption in variance-manifold relation**
>
> **A1** Thank you for your suggestion. We agree that it is not sufficient to fully understand data manifold with the variance concept. However, we would like to clarify that **we do not claim that "manifold can be understood solely by considering variance."**  To better explain our method, we summarize the main ideas of our paper as follows:
>
> 1. Firstly, we introduce manifold assumptions extended or borrowed from previous works [1] [2].
> 2. Then, we establish a connection between the data-to-manifold distance and data pixel variance, by assuming: $\\mathbb{E}[\\boldsymbol{\\epsilon}\_\theta^t]=0, \\mathbb{E}[\\boldsymbol{x}\_t]=0 $ . These assumption are justified as follows:
>    1. The noise $\\boldsymbol{\\epsilon}\_t$ added to clean data during the training phase follows normal distribution: $\\boldsymbol{\\epsilon}\_t \\sim \\mathcal{N}(\\mathbf{0}, \\boldsymbol{I})$, which implies $\\mathbb{E}[\\boldsymbol{\\epsilon}\_t]=0$. Consequently, assuming $\\mathbb{E}[\\boldsymbol{\\epsilon}^t\_\\theta]=0$ is reasonable for well-trained diffusion models.
>    2. The predicted noisy data at timestep $t-1$ can be expressed as $\\hat{x}\_{t-1} = \\sqrt{\\bar{\\alpha}\_{t-1}} \\frac{x\_t - \\sqrt{1 - \\bar{\\alpha}\_t} \, \\boldsymbol{\\epsilon}^{(t)}\_\\theta\\left(x\_t\\right)}{\\sqrt{\\bar{\\alpha}\_t}},+ \\sqrt{1 - \\bar{\\alpha}\_{t-1} - \\sigma\_t^2} \\, \\boldsymbol{\\epsilon}\_\\theta^{(t)}\\left(\\mathbf{x}\_t\\right) + \\sigma\_t \\boldsymbol{\\epsilon}'\_t,\\quad \\boldsymbol{\\epsilon}'\_t \\sim \\mathcal{N}\\left(\\mathbf{0}, \\mathbf{I}\\right).$ Then, as long as $\\mathbb{E}[\\frac{1}{n}\\Sigma\_{i=0}^n\\boldsymbol{x}\_{t\_i}] =0$, one has $\\mathbb{E}[\\frac{1}{n}\\Sigma\_{i=0}^n\\boldsymbol{x}\_{{t-1}\_i}] =0$. Considering that the initial noise is sampled from normal distribution: $\\boldsymbol{x}\_T \\sim \\mathcal{N}(\\mathbf{0},\\boldsymbol{I})$, we have $\\mathbb{E}[\\frac{1}{n}\\Sigma\_{i=0}^n\\hat{\\boldsymbol{x}}\_{t-1}]=0, \\forall t \\in [1, T]$.
> 3. Leveraging the pixel variance-manifold relation, we achieve the following:
>    1. provide a new perspective on the effectiveness of prior work [3] from the viewpoint of the data manifold.
>    2. introduce a training-free and tuning-free method, whose effectiveness is demonstrated across various models and datasets.
>
> We hope this explanation addresses your concerns and clarifies any potential misunderstandings. Please do not hesitate to let us know if further clarification or additional details are required.
>
> >### **Q1: "FID Hack" & using $q_0$ to Implement MCDO**
>
> **A2** We do not consider the proposed method as a "hack" of the FID score. We explain our approach as follows:
>
> 1. The primary objective of our method is to improve model's performance under low NFE situaiton. By guiding the deviated data toward their manifold, we achieved in reducing the exposure bias during generation. To achieve this, **we leverage pixel variance estimation collected from generation process.**
> 2. **Since the FID score is calculated by measuring the distributional distance between two sets of Inception-v3 features, it fundamentally differs from the pixel variance or $L_2$-norm of an image.** For instance, modifying the pixel variance of an image does not necessarily improve its quality.
> 3. **In addition to the improvements in FID score, our approach consistently achieves better results across other metrics, such as CLIP score, IS score, and sFID.** This suggests that **the effectiveness of our method stems from its ability to reduce the prediction errors made by the models, rather than manipulating any metric.**
>
> **We agree with your opinion that using the statistics collected from the training data, $q_0$, could potentially lead to even better results**. Given that in this case, the model would have access to the real training data, directly modifying the generated data distribution to "hack" FID, without improving its prediction accuracy.
>
> [1] Chung et al. Improving diffusion models for inverse problems using manifold constraints. NeurIPS 2022.
>
> [2] He et al. Manifold preserving guided diffusion. ICML 2024.
>
> [3] Li et al. Alleviating exposure bias in diffusion models through sampling with shifted time steps. ICLR 2024.

---

> ### Author Response · Authors · 2024-11-24
> **Response to Reviewer Tp1G (part 2)**
>
> >### **Q2: Results with other solvers & results on DDIM with NFE = 50**
>
> **A3** Thank you for your suggestion. We summarize the results with DPM++ sampler as follows:
>
> | **Timesteps** | **Method**              | **FID**   | CLIP Score | **$t\_{thre}$** |
> | ----------------------- | ----------------------- | --------- | ---------- | -------------- |
> | 5                       | DPM-Solver++(2M)        | 19.16     | 31.52      | --             |
> | 5                       | DPM-Solver++(2M) + MCDO | **18.51** | **31.62**  | 100            |
> | 6                       | DPM-Solver++(2M)        | 17.05     | 31.65      | --             |
> | 6                       | DPM-Solver++(2M) + MCDO | **16.39** | **31.72**  | 100            |
> | 8                       | DPM-Solver++(2M)        | 15.57     | 31.74      | --             |
> | 8                       | DPM-Solver++(2M) + MCDO | **15.32** | **31.79**  | 100            |
> | 10                      | DPM-Solver++(2M)        | 15.10     | 31.74      | --             |
> | 10                      | DPM-Solver++(2M) + MCDO | **14.88** | **31.85**  | 100            |
>
> These resuls indicate that **our method enhances the performance of high-order solver: DPM-Solver++**.  Given that DPM-Solver++ inherently reduces prediction error and causes less exposure bias at low NFE, the improvements of our method are slightly smaller compared to first-order solvers (e.g. a 2.57 FID decrease for DDIM). The results are also presented in Appendix A.10 of the revised manuscript.
>
> We conducted experiments with DDIM using NFE = 50 on the SDXL dataset but did not observe significant improvements in the quantitative results with the current hyperparameters ($S=20$, $t_{\text{thre}}=100$). This suggests that, **at higher NFE values, where step sizes are smaller, diffusion models likely make fewer prediction errors, and consequently, the exposure bias problem discussed in the paper becomes less severe.**
>
> However, it is important to note that our method is specifically designed for accelerated sampling in diffusion models, with a primary focus on low NFE scenarios. While it is true that increasing NFE can improve the results to some extent, we believe that this does not diminish the core value of our approach. **The main benefit of our method lies in its ability to address exposure bias in scenarios with fewer sampling steps, which is the key challenge in accelerated sampling**.

---

> ### Comment · Reviewer_Tp1G · 2024-11-24
>
> Thank you for your hard work. Responding to six rebuttals must have been very challenging; I appreciate it.
>
> I had two concerns: one was the FID hack, and the other was the strong assumption.
>
> ---
>
> ### FID hack
>
> > Since the FID score is calculated by measuring the distributional distance between two sets of Inception-v3 features, it fundamentally differs from the pixel variance or L2-norm of an image.
>
> I know your point. However, I am still curious. In my experience, the FID score is greatly influenced by the color distribution of images. In the ablation study, you applied MCDO from $T$ down to $t$, but couldn't my experiment be considered as applying MCDO only at $t = 1$? I didn't think it was a baseless experiment. If I'm wrong, please correct me.
>
> > In addition to the improvements in FID score, our approach consistently achieves better results across other metrics, such as CLIP score, IS score, and sFID.
>
> I agree.
>
> > We agree with your opinion that using the statistics collected from the training data could potentially lead to even better results.
>
> I had misunderstood the method. I'm sorry, and thank you for clarifying.
>
> ---
>
> ### Strong Assumption
>
> > Firstly, we introduce manifold assumptions extended or borrowed from previous works [1][2].
>
> **I kindly disagree with this statement because [1] and [2] do not use mean and variance information to approximate the manifold**. MCG utilizes the property that the input gradient of the diffusion model $\nabla_{x_t} \epsilon(\cdot)$ is aligned with the manifold. This is also the case in DPS [4]. MPGD approximates the manifold using a VAE. That is, while they begin their theoretical development by asserting from the Gaussian annulus theorem that $x_t$ has a low-dimensional manifold structure, they do not claim that this can be approximated using mean and variance.
>
> Personally, the reason I cannot readily give a positive evaluation of this work is that **I find it hard to accept that resolving exposure bias through mean and variance should be considered a Manifold Constraint**. Generally, the tangent vectors of the latent space manifold have clean, interpretable signals, as seen in the guidance vectors of DPS and MCG. This is even more evident in [5, Figure 6]. However, since the Gaussian annulus through mean and variance does not possess such manifold structure, the claim that improvements are due to the diffusion model manifold constraint sounds somewhat like overclaiming to me. I question whether the Manifold Constraint is an appropriate mathematical framework to excellently explain your method.
>
> Or... it could just be me thinking this way. I would not consider this point as crucial one for evaluation.
>
> [4] Chung et al. Diffusion Posterior Sampling for General Noisy Inverse Problems
>
> [5] Park et al. Understanding the Latent Space of Diffusion Models through the Lens of Riemannian Geometry
>
> ---
>
> To this end, some of my concerns regarding the FID hack have been resolved. However, it’s still difficult to be fully convinced because I haven’t seen results where the mean/variance regularization is applied only to the t=1 (or t=0). As for the strong assumption, I find it difficult to agree with the authors. Therefore, I will maintain my score for now. If the FID experiment yields positive results, I am considering raising my score to a 6.

---

> > ### Author Response · Authors · 2024-11-24
> >
> > Dear Reviewer Tp1G,
> >
> > Thank you for your thoughtful insights and for engaging deeply with the proposed method. We are grateful for the opportunity to clarify and further elaborate.
> >
> > >Applying MCDO only at $t=1$
> >
> > We are delighted to see your interest in the proposed method and sincerely apologize for initially misunderstanding your suggested experiment. To address your valuable insight, we will conduct an experiment where MCDO is applied exclusively at the final timestep.
> >
> > It is reasonable that modifying pixel variance at the final stage will influence image contrast. However, we are uncertain whether this approach has the capacity to effectively recover fine details, as such details are often generated during earlier stages of the denoising process.
> >
> > Thank you again for your invaluable feedback.

---

> ### Comment · Reviewer_Tp1G · 2024-11-24
>
> Thank you for your hard work! I really appreciate it.
>
> By the way, what do you think about the second point, strong assumption? I want to hear your (possibly informal) opinion about this.

---

### Official Review · Reviewer_fuNA · 2024-11-02

**Soundness:** 3
**Presentation:** 3
**Contribution:** 3
**Rating:** 6
**Confidence:** 3

**Summary:**

This paper presents an approach to enhance the efficiency and accuracy of diffusion models by addressing the issue of exposure bias in accelerated sampling. Leveraging a manifold hypothesis, the authors introduce a manifold constraint that reduces error accumulation during sampling without requiring additional training or extensive tuning.

**Strengths:**

1. The use of the manifold constraint is an interesting idea, which addresses exposure bias without adding additional training costs during sampling.
2. The use of well-chosen visualizations enhances the readability of the method section, conveying key information and clarifying the approach.
3. The paper is technically clear and well-organized, with the proposed method thoroughly explained.

**Weaknesses:**

1. The term "denoising observations" requires a clear and precise definition, as its current use lacks specificity. A more rigorous description would help readers to understand and improve the technical clarity of the paper.
2. The pre-computation process still requires a full denoising sequence (e.g., 1000 steps), which incurs substantial computational cost, especially when applying the proposed method to new datasets or domains. It is suggested that potential strategies for reducing this computational cost be discussed or that an analysis of the trade-offs between the number of steps in pre-computation and the method's performance be provided.
3. The number of samples and their diversity will influence the resulting approach $v_t$. However, the experiments simply set the sample number to 20 without discussing the diversity of prompts or other characteristics of the samples. Including these details would be valuable for readers to have a better understanding, and provide insights for the community. It is recommended to conduct an ablation study on the impact of sample number and diversity on the performance of the proposed method, or to provide more details on how you selected the 20 samples used in the experiments.

**Questions:**

Please refer to the weaknesses.

---

> ### Author Response · Authors · 2024-11-24
> **Response to Reviewer fuNA**
>
> We thank reviewer fuNA for the acknowledgement of our contributions and the helpful comments. Below, we provided a point-to-point response to all comments.
>
> >### **W1: Explanation for denoising observation**
>
> **A1** We appreciate it that you point this out. In this paper, we leverage the term *denoising observation* from paper DDIM [1], which proposes the widly used accelerated sampling technique.
>
> >### **W2: Discussion on the Pre-computation of the Full Denoising Sequence**
>
> **A2** Thank you for highlighting the trade-off between the number of steps in pre-computation and the performance of the method. In the paper, we use a long denoising sequence (1,000 steps for SDXL experiments on the MS-COCO dataset) to better capture the statistics required by the proposed method. However, **denoising trajectories with fewer timesteps are also feasible**. For instance, in experiments on the CelebA-HQ dataset, we collect statistics using denoising sequences of 500 timesteps, and **the proposed method still demonstrates noticeable improvement with statistics derived from a shorter sampling trajectory**.
>
> Nevertheless, we believe that, in general, increasing the number of sampling steps (and thus reducing the sampling intervals) leads to more accurate estimation of the pixel variance of $\hat{x}_0^{(t)}$.
>
> >### **W3: Diversity of Reference Samples and Details in the Collection**
>
> **A3** To ensure a sufficient level of diversity in the samples collected for $v_t$ estimation:
> 1. **For text-to-image generation with MCDO, we randomly select 20 distinct prompts from the dataset** to guide the generation of reference samples (one sample per prompt).
> 2. **In class-conditional generation on ImageNet, 64 randomly sampled class IDs were used** to guide the generation of 64 images (one sample per class).
>
> We appreciate your insights into the trade-offs between performance and the number of reference samples. For ease of implementation, we use $S=20$ for text-to-image generation with SDXL, and $S=64$ for the other experiments. **While we agree that the $v_t$ estimation could potentially benefit from an increased number of reference samples, considering the improvements already achieved by MCDO in these experiments, we believe that the current choice of hyperparameters is practical.** Therefore, we have decided not to further tune them for potentially better quantitative results.
>
> [1] Song et al, Denoising Diffusion Implicit Models, ICLR 2021.

---

> ### Comment · Reviewer_fuNA · 2024-12-02
>
> I appreciate the authors' efforts in rebuttal. I decide to keep my positive rating. Thank you.

---

> > ### Author Response · Authors · 2024-12-02
> >
> > Dear Reviewer fuNA,
> >
> > Thank you for your helpful comments. We greatly appreciate your recognition of our efforts and your engagement with our work. Your support is invaluable to us.
> >
> > Best regards,
> >
> > The authors

---

### Official Review · Reviewer_oFBR · 2024-11-03

**Soundness:** 3
**Presentation:** 3
**Contribution:** 2
**Rating:** 6
**Confidence:** 4

**Summary:**

This paper views the issue of the quality degradation during accelerating diffusion model’s inference on the perspective of the explore bias. The authors point out that explore bias is an important reason that contributes to the loss of image quality when reducing inference steps. The noisy prediction will direct to inaccurate manifolds and thus the errors would be accumulated and amplified. A manifold constraint is proposed to curate this bias and thus lead to better image quality when significantly reducing inference steps.

**Strengths:**

1. The proposed method is well supported by a series of proofs with some assumptions.
2. The quantitative results look very promising, and largely outperforms the alternative.E.g.,  in Table 3, MCDO with 4 steps has better results than DDIM with 5 steps.
3. The method is well motivated and the writing logically reasonable and flows well.

**Weaknesses:**

1. Assumption 3 is a very important part in deriving the manifold constraint. I think it is too strong. I understand it's motivated by Equation (17), however \hat{x}_0 could have a different distribution of x_0 which is inaccessible during inference. Could you elaborate more?
2. There are some details not explained well for some key equations/explanations. I added those in the questions below.

**Questions:**

1. In table4, why do fewer steps have lower FID?
2. Can you add more details of how to get equation 11 and 16 in the appendix?
3. I didn’t understand why \epslon_t is equal to \sqrt{n} when n is large in L261? Can you explain?
4. Can you elaborate on L271? In my understanding, in fig3, var(x_t) decreases faster as reducing steps, thus potentially making d(.) larger than r_t in later steps. But figure 3 is only on one sample, how to generalize this observation?
5. The qualitative examples look a bit over-saturated, it would be helpful if you can also show an oracle results (e.g., 1000 steps) on the side for comparison.

---

> ### Author Response · Authors · 2024-11-24
> **Response to Reviewer oFBR (part 1)**
>
> Thank you for your insightful feedback.Our answers to each question are shown below:
>
> >### **W1 Explanation of Assumption 3**
>
> **A1** Thanks for raising this question about Assumption 3. We agree that for most of timesteps, especially those in the early denoising stage, $\hat{x}_0$ have a very different distribution from that of $x_0$. However, **this difference is a fundamental aspect of our method’s design**, which is detailed in Algorithm 1.
>
> The objective of MCDO is to dynamically adjust $\hat{x}_0$, bringing it closer to its corresponding distribution based on the current timestep $t$, rather than adhering to the fixed distribution of $x_0$. For example, if the distribution of $x_0$ were used to correct that of $\hat{x}_0^{(t)}$ at the initial timestep (t=T),  the entire process would be adversely affected.
>
> >### **Q1 Fewer steps have lower FID on ImageNet**
>
> **A2** Thank you for pointing this out. We also observed this phenomenon during our experiments and, as a result, re-evaluated all methods using a new environment. The outcomes of this re-evaluation were consistent with the results presented in the paper.
>
> As shown in the visualization of the generated images provided in Appendix A.8, **while fewer steps yield lower FID values, the corresponding image quality is noticeably worse, which aligns with general intuition.**
>
> Since FID is calculated based on the distance between estimated distributions of two Inception-v3 features, it may not always perfectly reflect image quality in all scenarios. We believe this could explain the observed behavior on this dataset.
>
> >### **Q2 Explanation of Equation 11 and Equation 16**
>
> **A3** Equation 11 is based on the triangle inequality. As Equation 16 is derived by plugging $\| \sqrt{\bar{\alpha}_t} x_0 + \sqrt{1-\bar{\alpha}_t} \epsilon_t \| \approx \sqrt{\| \sqrt{\bar{\alpha}_t} x_0 \|^2 + \| \sqrt{1-\bar{\alpha}_t} \boldsymbol{\epsilon}_t \|^2}$ into $d(\hat{x}_t,1, \mathcal{M}_t) \gtrapprox \left| \| \sqrt{\bar{\alpha}_t} x_0 + \sqrt{1-\bar{\alpha}_t}\boldsymbol{\epsilon}_t \| - \sqrt{n\text{Var}(\hat{x}_t)} \right|$. Typos in Equation16 are corrected in the revised manuscript.
>
> >### **Q3 Explanation of approximation in L261**
>
> **A4** To better explain L261, we would like to refer you to the Equation 14 in L258-259. Equation 14 says that, for random data sampled from n-dimension normal distribution: $\epsilon \sim \mathcal{N}(0,1,\mathbb{R}^n)$, the expectation of its norm satisfy: $|\\mathbb{E}[\\|\\boldsymbol{\\epsilon}\\|] - \\sqrt{n} | \\leq \\frac{1}{\\sqrt{n}}$ [1]. As $n$, the latent space dimension in diffusion models is usually large in diffusion models (e.g. $n=12288$ for LDM-4 trained on CelebA-HQ), we have: $\\frac{1}{\\sqrt{n}} \\rightarrow 0$. Therefore, $| \\mathbb{E}[\\|\\boldsymbol{\\epsilon}\\|] - \\sqrt{n} | \\| \\rightarrow 0$. That is to say, with high probability, we have $\\|\\boldsymbol{\\epsilon}\\| \\approx \\sqrt{n}$.
>
> Alternatively, with large $n$, one can directly consider this: $\\mathbb{E}(\\|\\boldsymbol{\\epsilon}\\|^2) = \\mathbb{E}(\\Sigma\_{i=0}^{n}\\boldsymbol{\\epsilon}\_i^2)= \\Sigma\_{i=0}^{n}\\mathbb{E}(\\boldsymbol{\\epsilon}\_i^2) = \\Sigma\_{i=0}^{n}\\left(\\mathbb{E}(\\boldsymbol{\\epsilon}\_i)^2+\\text{Var}\\left(\\boldsymbol{\\epsilon}\_i\\right)\\right)= \\Sigma\_{i=0}^{n}\\left(\\text{Var}\\left(\\boldsymbol{\\epsilon}\_i\\right)\\right) =n$. Which leads to $\\mathbb{E}[\\|\\boldsymbol{\\epsilon}\\|] \\approx \\sqrt{n}$.
>
> [1] Sven-Ake Wegner. Lecture notes on high-dimensional data, 2024.

---

> ### Author Response · Authors · 2024-11-24
> **Response to Reviewer oFBR (part 2)**
>
> >### **Q4: Explanation of L271 and generalization of observation presented in figure 3**
>
> **A5** Thanks for raising this question. In fact, **the objective of Equation 16 is to elucidate the relation between** $d(x_t,1,\mathcal{M}_t)$ **and the deviation of** $\text{Var}(x_t)$. **As this perspective can provide a manifold perspective to understand the effectiveness of previous published work TS-DDIM [2]**, which is based on assumption of using pixel variance of a single image to estimate distribution variance.
>
> **We appreciate your insightful observation that a decrease in pixel variance could potentially make $d(x_t, 1, \mathcal{M}_t)$ larger than $r_t$.** However, we do not claim that the pixel variance of $x_t$ will necessarily decrease with fewer NFEs in all cases, as this would be too strong of an assumption, even though such observation might holds true statistically. Again, thank you for pointing this out.
>
> >### **Q5 Saturated examples**
>
> **A6**  Thanks for mentioning this. We would like to answer this question in detail below:
> 1. As the sampling steps decreases, blurriness are observed in many baseline samples. Therefore, it is reasonable for solutions which improve image quality make the generated results look "relatively saturated". Which is also observed in experiments in previous published work [3].
> 2. We contribute the higher saturation to our method's effectiveness in reducing models' prediciton error, **producing image with finer quality (sharper textures and richer structural details), as can be observed in all visualizations presented in paper.**
> 3. To provide a fair illustration of the saturation level, we have included qualitative results generated with 1,000 steps in Appendix A.13 of the revised manuscript. **These results demonstrate that our method achieves a saturation level similar to that of high-quality images generated with 1,000 steps**, when compared to the images produced without our method.
>
> [2] Li et al. Alleviating Exposure Bias in Diffusion Models through Sampling with Shifted Time Steps. ICLR 2024.
> [3] Chen et al. On the Trajectory Regularity of ODE-based Diffusion Sampling, ICML 2024

---

> > ### Comment · Reviewer_oFBR · 2024-11-25
> > **Increase score**
> >
> > I appreciate the prompt answers, which addressed all my questions. I increased my score to 6 based on my understanding of the novalty/contribution. I think this is a valid work, pushing the boundary to exteme cases like single step generation or compare with other methods with more steps (similar to Appendix A.13) would better demonstrate the effectiveness of the proposed method. A.13 does show that the saturation level is closer to 1000steps, but the numbers in table 2-5 are still much worse compared with more steps. Though it is understandable, showing more improvement along this direction would make the contribution stronger. Thanks.

---

> > > ### Author Response · Authors · 2024-11-26
> > >
> > > Dear Reviewer oFBR,
> > >
> > > Thank you very much for your valuable suggestions and for engaging in the discussion.
> > >
> > > We greatly appreciate your recognition of the novelty and contributions of our work. We agree that further improvements in this direction could provide additional insights, and we will certainly explore this avenue in future work.
> > >
> > > Once again, thank you for your constructive comments and your engagement with our work.
> > >
> > > Best regards,
> > >
> > > The authors

---

### Official Review · Reviewer_JP98 · 2024-11-03

**Soundness:** 3
**Presentation:** 3
**Contribution:** 2
**Rating:** 6
**Confidence:** 4

**Summary:**

- This paper proposes a method for improving the performance in accelerated diffusion sampling algorithms
- The paper identifies the exposure bias in accelerated diffusion sampling
- The method applies manifold constraint for reducing the exposure bias that occurs in accelerated sampling.
-  The paper presents evaluations of the methods showing improvements over the baseline.
- A discussion on the geometric view of the exposure bias is presented.

**Strengths:**

- The paper is well written
- The method is evaluated on multiple diffusion models trained on different datasets
- The method is simple yet effective. It does not require any further training and the additional computations are marginal.
- The approach shows an improvement over the baselines in most cases

**Weaknesses:**

- The derivation of section 4.2 is relatively weak, many assumptions and loose steps need to be either refined or omitted.
  - In section 4.2, you assume that $E[\epsilon_\theta^t]=0$, but this is not always the case.
  - In Equation (12) the authors utilize the fact that $\frac{1}{n}\sum_{i=1}^n \hat{x}_i \approx 0$, which does not always hold.
  - Equation (15) does not hold, an expectation value is required for it to be true
- The evaluations include only comparison to DDIM, even though there are many accelerated samplers that achieve much better results [1,2,3], adding them to the tables is very important for evaluating the method.

[1] GENIE: Higher-Order Denoising Diffusion Solvers

[2] DPM-Solver: A Fast ODE Solver for Diffusion Probabilistic Model Sampling in Around 10 Steps

[3] DPM-Solver++: Fast Solver for Guided Sampling of Diffusion Probabilistic Models.

**Questions:**

- Figure 3: the x-axis title is not clear, what do you mean by denoising steps if the sampling steps are given in the legend? and in general the figure needs to be explained properly
- In section 4.2, you assume that $E[\epsilon_\theta^t]=0$, but this is not always the case.
- Equation 15 does not make any sense without expectation value.

---

> ### Author Response · Authors · 2024-11-24
> **Response to Reviewer JP98**
>
> Thank you very much for your helpful reviews and feedback, we now reply to your concerns:
>
> >### **W1&Q2&Q3: Assumptions on the mean value of noise estimation and noisy data, and refinement for Eq. 15**
>
> **A1** Thank you for pointing this out. We explain each as follows.
>
> 1. In the proposed method, the assumption that the expectation of the pixel sum of the noise estimation and the noisy data is zero **serves as tool to relate the pixel variance of the noisy data and its deviation to the manifold.**
> 2. Relationship between these two concepts provides a novel insight into the methodology of TS-DDIM, and forms the basis for developing our proposed method.
> 3. **We made these assumptions based on statistic facts listed as follows**:
>    1. Denoting the noise added to clean data during training phase as $\\boldsymbol{\\epsilon}_t$. As it follows normal distribution: $\\boldsymbol{\\epsilon}_t \\sim \\mathcal{N}(\\mathbf{0}, \\boldsymbol{I})$, we have $\\mathbb{E}[\\boldsymbol{\\epsilon}_t]=0$. Therefore, when diffusion models are well-trained, assuming $\\mathbb{E}[\\boldsymbol{\\epsilon}^{t}\_\theta]=0$ is reasonable.
>    2. Given that the predicted noisy data of timestep $t-1$ can be expressed as $\\hat{x}\_{t-1} = \\sqrt{\\bar{\\alpha}\_{t-1}} \\frac{x\_t - \\sqrt{1 - \\bar{\\alpha}\_t} \, \\boldsymbol{\\epsilon}^{(t)}\_\\theta\\left(x\_t\\right)}{\\sqrt{\\bar{\\alpha}\_t}},+ \\sqrt{1 - \\bar{\\alpha}\_{t-1} - \\sigma\_t^2} \\, \\boldsymbol{\\epsilon}\_\\theta^{(t)}\\left(\\mathbf{x}\_t\\right) + \\sigma\_t \\boldsymbol{\\epsilon}'\_t,\\quad \\boldsymbol{\\epsilon}'\_t \\sim \\mathcal{N}\\left(\\mathbf{0}, \\mathbf{I}\\right).$ Then, as long as $\\mathbb{E}[\\frac{1}{n}\\Sigma_{i=0}^n\\boldsymbol{x}\_{t\_i}] =0$, one has $\\mathbb{E}[\\frac{1}{n}\\Sigma\_{i=0}^n\\boldsymbol{x}\_{{t-1}\_i}] =0$. Considering that the initial noise is sampled from normal distribution: $\\boldsymbol{x}\_T \\sim \\mathcal{N}(\\mathbf{0},\\boldsymbol{I})$, we have $\\mathbb{E}[\\frac{1}{n}\\Sigma\_{i=0}^n\\hat{\\boldsymbol{x}}\_{{t-1}\_i}]=0, \\forall t \\in [1, T]$.
> 4. In addition, **extensive experiments demonstrate the effectiveness of our approach across various model and datasets**. Therefore, we believe these assumptions are acceptable.
> 5. Nevertheless, we agree that these assumptions may not hold for every case. In some application, there might exist cases where the mean of latent does not approaches zero, we leave this as further investigation in future work.
> 6. Thanks for your suggestion to improve Equation 15. We correct the problem you mentioned in the revised manuscript.
>
> >### **W2: Evaluation on other accelerated samplers**
>
> **A2** We appreciate your suggestion to include evaluations with other samplers for a more comprehensive comparison. **Following your feedback, we have incorporated evaluations with the DPM++2M scheduler on the MS-COCO dataset using SDXL**. The experiments followed the same settings as in our original submission. Due to limited computational resources and the rebuttal timeline, we evaluated performance using 10k generated images. The results, now included in the Appendix A.10 of the revised manuscript, are summarized below:
>
> | **Timestep** | **Method** | **FID**   | **CLIP Score** | **$t\_{thre}$** |
> |  ---------- |  ----------|  ---------- |  ---------- |  ----------|
> | 5                       | DPM-Solver++(2M)        | 19.16     | 31.52      | --             |
> | 5                       | DPM-Solver++(2M) + MCDO | **18.51** | **31.62**  | 100            |
> | 6                       | DPM-Solver++(2M)        | 17.05     | 31.65      | --             |
> | 6                       | DPM-Solver++(2M) + MCDO | **16.39** | **31.72**  | 100            |
> | 8                       | DPM-Solver++(2M)        | 15.57     | 31.74      | --             |
> | 8                       | DPM-Solver++(2M) + MCDO | **15.32** | **31.79**  | 100            |
> | 10                      | DPM-Solver++(2M)        | 15.10     | 31.74      | --             |
> | 10                      | DPM-Solver++(2M) + MCDO | **14.88** | **31.85**  | 100           |
>
> These resuls indicate that our method **enhances the performance of high-order solvers** like DPM-Solver++.
>
> >### **Q1 Explanation of x-axis title in Figure 3**
>
> **A3** Thank you for highlighting this point. The timesteps on the x-axis correspond to the timesteps $t$, one of the input to the noise estimation network: $\epsilon_\theta(x_t,t,y)$. In contrast, the legends represent the number of inference steps used during sampling. Based on your feedback, we have revised and improved Figure 3 in the updated manuscript.

---

> > ### Comment · Reviewer_JP98 · 2024-11-27
> >
> > Thank you for your response.
> >
> > - Regarding the assumptions: While I agree that the assumptions are reasonable and work well in practice, my concern lies in the derivation of the analytical bounds. Specifically, omitting certain terms based on their negligible impact in practice may undermine the accuracy of the analysis. For the results to be both correct and accurate, these terms should either be explicitly considered or quantitatively justified.
> >
> > - On the method's improvement: I appreciate that your approach improves DPM-Solver++, although the improvement is marginal compared to the improvement you got using DDIM. This observation supports your claims, and I have increased my score accordingly. However, I recommend standardizing the evaluation to align more closely with the literature (my other suggestions in the review). This would provide a more robust and comparable assessment of your method's effectiveness. Please consider incorporating this suggestion in your revised version.

---

> > > ### Author Response · Authors · 2024-11-27
> > >
> > > Dear Reviewer,
> > >
> > > Thank you very much for your thoughtful feedback and for engaging in the discussion to improve our work. We sincerely appreciate your recognition of our method’s simplicity and effectiveness, as well as your valuable suggestions for further improvement. We will carefully consider these recommendations in the revised version.
> > >
> > > Best regards,
> > >
> > > The authors

---

### Official Review · Reviewer_rs44 · 2024-11-04

**Soundness:** 3
**Presentation:** 3
**Contribution:** 3
**Rating:** 6
**Confidence:** 3

**Summary:**

To narrow the gap between the training of the sampling phase of diffusion models, the authors analyze the diffusion processes from the view of the manifold. They propose to compute the statics of all intermediate diffusion variables and calibrate the sampling process based on the computed statics (variance, mean, \etc). Experiments show that the proposed method can reduce the sampling steps while maintain the generation quality.

**Strengths:**

1. The proposed method improves sample quality without adding substantial computational overhead.
2.Unlike some prior methods, this approach does not require model retraining or intensive hyperparameter tuning.
3. The manifold constraint method shows improved performance across various high-resolution datasets, achieving better FID scores with fewer sampling steps.
4. The paper provides both theoretical analysis and empirical evidence to support its method, including experiments on multiple tasks like image and video generation.

**Weaknesses:**

1. The proposed method needs to be verified on more diffusion schedulers, such as DPMSolver, PNDM.

2. Some ODE-based diffusion models such as rectified flow and consistency models can reduce the sampling steps to two or even one. The proposed method focuses on accelerating the sampling process but is not compared with these fast-sampling diffusion models.  The authors are encouraged to apply their methods to more recent diffusion models (\ie SD3, FLUX) to show their priority and general ability.

**Questions:**

See the weakness above

---

> ### Author Response · Authors · 2024-11-24
> **Response to Reviewer rs44**
>
> Thank you for your insightful feedback and for highlighting the need for comparisons with high-order diffusion samplers and recent diffusion models. We recognize the importance of demonstrating the scalability and general applicability of our proposed method. Below, we address each concern in detail.
>
> >### **W1**: Verification on more diffusion schedulers, such as DPM solver
>
> **A1** We conducted additional experiments combining our method with the DPM++2M scheduler on the MS-COCO dataset using SDXL. The experiments followed the same settings as in our original submission (image resolution, classifier-free guidance scale). Due to limited computational resources and the rebuttal timeline, we evaluated performance using 10,000 generated images. The results, now included in the Appendix A.10 of the revised manuscript, are summarized below:
>
> | **Number of Timesteps** | **Method**              | **FID**   | CLIP Score | **$t_{thre}$** |
> | ----------------------- | ----------------------- | --------- | ---------- | -------------- |
> | 5                       | DPM-Solver++(2M)        | 19.16     | 31.52      | --             |
> | 5                       | DPM-Solver++(2M) + MCDO | **18.51** | **31.62**  | 100            |
> | 6                       | DPM-Solver++(2M)        | 17.05     | 31.65      | --             |
> | 6                       | DPM-Solver++(2M) + MCDO | **16.39** | **31.72**  | 100            |
> | 8                       | DPM-Solver++(2M)        | 15.57     | 31.74      | --             |
> | 8                       | DPM-Solver++(2M) + MCDO | **15.32** | **31.79**  | 100            |
> | 10                      | DPM-Solver++(2M)        | 15.10     | 31.74      | --             |
> | 10                      | DPM-Solver++(2M) + MCDO | **14.88** | **31.85**  | 100           |
>
> These resuls indicate that our method **enhances the performance of high-order solvers** like DPM-Solver++.  While the improvements are slightly smaller compared to first-order solvers (e.g. a 2.57 FID decrease for DDIM)—likely because DPM-Solver++ inherently reduces prediction error and causes less exposure bias at low NFE.
>
> Nonetheless, the consistent performance gains highlight the versatility of our method as a **plug-and-play module** compatible with a wide range of solvers.
>
> >### **W2**: Results on recent diffusion models
>
> **A2** We applied our method on **Stable Diffusion 3** ( SD3 ) [1], a recent model based on flow matching. The quantitative results, obtained by generating 10,000 images at \(1024 \times 1024\) resolution, are summarized below:
>
> | **Model**          | **Method**  | **Steps** | **FID↓**  | **CLIP Score↑** | **$t_{thre}$** |
> | ------------------ | ----------- | --------- | --------- | --------------- | -------------- |
> | Stable Diffusion 3 | DDIM        | 6         | 33.31     | 30.56           | --             |
> | Stable Diffusion 3 | DDIM - MCDO | 6         | **27.57** | **31.18**       | 0              |
> | Stable Diffusion 3 | DDIM        | 9         | 22.71     | 31.44           | --             |
> | Stable Diffusion 3 | DDIM-MCDO   | 9         | **22.26** | **31.70**       | 0              |
>
> The results demonstrate **consistent improvements in both FID and CLIP scores** when integrating our method with SD3. These findings highlight the scalability and compatibility of our approach with cutting-edge diffusion models.
>
> The results are now included in Appendix A.11 of the revised manuscript.
>
> [1] Esser et al. Scaling Rectified Flow Transformers for High-Resolution Image Synthesis. ICML 2024 Oral.

---

### Meta-Review · Area_Chair_cJkf · 2024-12-19

**Metareview:**

This paper proposes a novel method to address exposure bias in accelerated diffusion sampling using a manifold constraint, demonstrating significant improvements in FID and CLIP scores across various datasets and diffusion models. Reviewers praised the method’s simplicity, training-free implementation, and consistent performance gains, particularly with low NFE scenarios. However, some concerns were raised regarding the theoretical assumptions, limited comparisons with high-order solvers, and occasional visual artifacts, suggesting further refinements in the theoretical framework and broader empirical evaluations would enhance its impact.

**Additional Comments On Reviewer Discussion:**

During the rebuttal period, reviewers raised concerns about the strong assumptions in the theoretical framework, limited evaluations with high-order solvers, and the potential for FID improvements being overly influenced by color distribution adjustments. The authors addressed these by conducting additional experiments with DPM-Solver++, evaluating under varying CFG scales, and clarifying the theoretical basis, providing empirical evidence of their method's robustness across metrics like FD and KD. They also included further visualizations and ablation studies to showcase qualitative and quantitative improvements, resolving most concerns and demonstrating the method’s generalizability and practicality.

---

### Decision · Program_Chairs · 2025-01-22

Accept (Poster)